# Measuring Scientific Capabilities of Language Models with a Systems Biology Dry Lab

**Haonan Duan**[*]
University of Toronto

**Stephen Zhewen Lu**[*]
SickKids

**Caitlin F. Harrigan**
University of Toronto

**Nishkrit Desai**
Axiom

**Jiarui Lu**
Mila

**Michał Koziarski**
SickKids

**Leonardo Cotta**
Vector Institute

**Chris J. Maddison**
University of Toronto

## Abstract

Designing experiments and result interpretations are core scientific competencies, particularly in biology, where researchers perturb complex systems to uncover the underlying systems. Recent efforts to evaluate the scientific capabilities of large language models (LLMs) fail to test these competencies because wet-lab experimentation is prohibitively expensive: in expertise, time and equipment. We introduce SciGym, a first-in-class benchmark that assesses LLMs' iterative experiment design and analysis abilities in open-ended scientific discovery tasks. SciGym overcomes the challenge of wet-lab costs by running a dry lab of biological systems. These models, encoded in Systems Biology Markup Language, are efficient for generating simulated data, making them ideal testbeds for experimentation on realistically complex systems. We evaluated six frontier LLMs on 137 small systems, and released a total of 350 systems at `https://huggingface.co/datasets/h4duan/scigym-sbml`. Our evaluation shows that while more capable models demonstrated superior performance, all models' performance declined significantly as system complexity increased, suggesting substantial room for improvement in the scientific capabilities of LLM agents.

## 1  Introduction

Scientific experimentation is the primary tool that researchers in the natural sciences use to gain insight about our world's physical and biological systems. A researcher can test a hypothesis by systematically perturbing systems and observing effects. They then interpret their results and identify the next best experiment to perform, closing the scientific discovery loop. Thus, when assessing large language models (LLMs) capabilities as scientists, it is essential to have evaluation frameworks effectively testing these skills. However, in this context, a fundamental challenge emerges: *How can we generate experimental data in the loop to evaluate an LLM's scientific iteration?*

For each experiment proposed by an LLM, we need to obtain data corresponding to how the system responds to the suggested perturbations. In other domains, the success of end-to-end benchmarks has hinged on our ability to quickly and automatically assess LLM actions. For example, SWE-bench tests an LLM's ability to resolve real-world GitHub issues, as evaluated by a suite of unit tests Jimenez et al. (2024). Importantly, coding is done in formal programming languages that can be executed and analyzed cheaply. Unfortunately, this is not attainable with a traditional laboratory setup in which experiments are expensive and laborious to perform. Substantial progress has been

---

[*]Equal contribution.

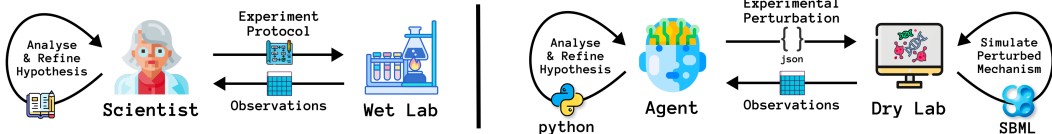

Figure 1: **SCIGYM simulates scientific discovery.** Real-world scientists (left) iteratively refine their hypotheses by designing experiments, collecting lab data, and analyzing results. In SCIGYM (right), agents request experimental actions, receive the observation data simulated from the perturbed reference system, and then perform analyses in a Python shell to refine their hypotheses.

made in robotics and control for self-driving labs, however these technologies are not yet mature enough to easily be used to assess LLM performance at scale Abolhasani & Kumacheva (2023).

A promising approach comes from systems biology, where formal mathematical models of various biological processes have been developed. BioModels (Malik-Sheriff et al., 2020) is a public repository that hosts manually-curated models from published literature in variety of fields —such as cell signaling, metabolic pathways, gene regulatory networks, and epidemiological models of infectious disease. These models are stored in Systems Biology Markup Language (SBML), a standard exchange format which provides a machine-readable representation supported by numerous software tools for simulation and analysis (SBML.org; Dubitzky et al., 2013; Medley et al., 2018; Hoops et al., 2006). Crucially, SBMLs can facilitate a "dry lab" in the context of a benchmark: when an LLM requests an experiment, we can create an SBML file describing the proposed perturbation, and return the corresponding simulated data.

Based on this insight, we introduce SCIGYM, an agentic benchmark that evaluates LLMs' end-to-end scientific discovery abilities. We leverage 350 SBML models from BioModels (Malik-Sheriff et al., 2020) ranging in complexity from simple linear pathways with a handful of species and reactions to sophisticated networks containing hundreds of molecular components and interactions. We release two benchmark splits: *small*, which contains 137 models with fewer than 10 reactions each, and *large*, which contains the remaining 213 models with up to 400 reactions each. Our framework operates as follows: the agent is tasked with discovering a reference system described by a biology model by analyzing data simulated from the SBML. The agent can perturb the simulated system and write Python code to analyze the resulting data. Performance is assessed by measuring correctness in topology of the graph of the true system, recovery of the reactions in the system, and percent error in data generated by a model proposed by the agent. To succeed in our tasks, the agent must excel in both experimental design and data analysis in an open-ended manner, mirroring the core competencies of human scientists.

In addition to our specific benchmarking settings, SCIGYMis highly extensible, allowing researchers to add various configurations that test different tasks, abilities, and biological systems. Although these SBML models do not fully capture all physiological variables, these carefully curated dynamic systems represent a treasure trove of biochemical insights derived from expert knowledge. By harnessing these models to simulate from, we posit that faithfully recovering the underlying systems which generate the trajectories of species concentration over time is a challenge which is suitably complex to evaluate an agents's scientific reasoning. We foresee this is a capability which will translate to real-world performance when it comes to inferring mechanisms from experimental data.

We evaluated six LLMs from three frontier model families (Gemini, Claude, GPT-4) on SCIGYM-small. We found that more capable models generally outperform their smaller counterparts, with Gemini-2.5-Pro leading our benchmark followed by Claude-Sonnet. However, we also identified consistent limitations across all models: 1) performance decreases as the underlying biological system becomes more complex; 2) proposed mechanisms often overfit to experimental data without generalizing to unseen initial conditions; and 3) models struggle to identify subtle relationships, particularly those involving modifiers.

## 2 Background on formal biology models and SBML

Mathematical modeling of biochemical reaction networks, such as metabolic, gene regulatory, and protein-signaling networks is one of the central tasks in systems biology (Sauro, 2020). Motivated

by the need to share models, researchers have developed standard machine-readable formats for describing chemical reaction networks. Among these, the Systems Biology Markup Language (SBML) has become a de facto standard for formal specifications of dynamic network models via an XML-based standard. Going forward, we use `texttt` when referring to specific SBML tag types.

SBML adopts terminologies from biochemistry: `reaction` is the central object for describing processes that change the quantities of `species` (entities such as small molecules, proteins, *etc.*) in a model. The `kineticLaw` of a `reaction` specifies in MathML the speed with which this process happens. Systems described by SBMLs can be simulated as systems of ordinary differential equations, as we discuss below, or other frameworks like discrete stochastic systems. The `reaction` tag can be understood in terms of the participating `species` found in the `listOfReactants`, `listOfProducts`, and possibly `listOfModifiers`. A `reaction` is a directed relationship: the quantities of reactants decrease as they are consumed, while the quantities of products increase as they are generated due to the reaction proceeding. Modifiers affect reaction rates, and their quantities remain unchanged through the reactions. A reaction can also define the relevant `kineticLaw` and `listOfParameters` to characterize the rate of reactions. SBML can include other optional components, such as `rule` and `event` tags. We refer readers to Appendix D.1 major SBML components and the SBML Level 3 language specification for more details (Hucka et al., 2019).

**Reduced SBML Representation.** For our purposes, an SBML is a 4-tuple $(\mathcal{S}, \Theta, \mathcal{R}, \mathcal{T}) \in \mathfrak{S}$ consisting of its `listOfSpecies` $\mathcal{S} := \{S_j\}_{j=1}^n$, `listOfParameters` $\Theta$, `listOfReactions` $\mathcal{R} := \{R_i\}_{i=1}^m$, and all other tags $\mathcal{T}$. Each reaction $R_i := (\mathbb{R}_i, \mathbb{P}_i, \mathbb{M}_i, \theta_i, r_i, \mathbb{T}_i)$ consists of its `listOfReactants` $\mathbb{R}_i$, `listOfProducts` $\mathbb{P}_i$, `listOfModifiers` $\mathbb{M}_i$, `listOfParameters` $\theta_i$, a `kineticLaw` represented as a function $r_i : \mathfrak{S} \to \mathbb{R}^+$, and all other tags $\mathbb{T}_i$.

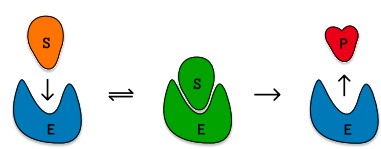

Figure 2: **Biological process in The Michaelis–Menten enzymatic process.** The substrate **S** binds to the enzyme **E** (left) which catalyzes the formation of the product **P** (right).

**Example system.** The Michaelis–Menten (Michaelis et al., 2011) enzymatic process describes a system that produces product **P** from the substrate **S**, catalyzed by the enzyme **E**. The reaction is illustrated in Figure 2 and represented via the chemical equation: $E + S \underset{k_{\text{off}}}{\overset{k_{\text{on}}}{\rightleftharpoons}} ES \xrightarrow{k_{\text{cat}}} E + P$. The SBML model to describe this system consists of the set of species $\mathcal{S} = \{\mathbf{S}, \mathbf{E}, \mathbf{ES}, \mathbf{P}\}$, two reactions $R_1, R_2$, and parameters $\Theta = \{k_{\text{off}}, k_{\text{on}}, k_{\text{cat}}\}$. The first reaction describes the reversible formation of the enzyme-substrate complex with rate $r_1(\mathcal{S}, \Theta, \mathcal{R}, \mathcal{T}) = v k_{\text{on}}[\mathbf{E}][\mathbf{S}] - v k_{\text{off}}[\mathbf{ES}]$, where $v > 0$ is the volume of the compartment where the reaction occurs, specified in $\mathcal{T}$. The second reaction represents the conversion of the substrate to the product with rate $r_2(\mathcal{S}, \Theta, \mathcal{R}, \mathcal{T}) = v k_{\text{cat}}[\mathbf{ES}]$. The square brackets $[\cdot]$ map a species to its time-varying concentration. We omit the time index to avoid notational clutter.

To translate this into an ODE, we sum the rates implied by the reactions for each species. For $S_j$ we have $v \cdot d[S_j]/dt = \sum_{i=1}^m s_{ij} \cdot r_i(\mathcal{S}, \Theta, \mathcal{R}, \mathcal{T})$. $s_{ij}$ are stoichiometric coefficients (specified in $\mathbb{R}_i$ or $\mathbb{P}_i$) that describe how much of each species is consumed or produced in the reaction. They are signed according to the role of $S_j$ in $R_i$: concentration decreases and $s_{ij} < 0$ if $S_j$ is a reactant in reaction $R_i$, concentration increases and $s_{ij} > 0$ if $S_j$ is a product, and $s_{ij} = 0$ otherwise. We include a full worked example of this enzyme system's SBML and ODE in Appendix D.2, and a worked example of an SBML with a modifier in Appendix D.3.

**Simulation.** Numerous simulation softwares have been developed to efficiently generate time-series data from SBML models, with the most popular ones being libRoadRunner (Somogyi et al., 2015), and COPASI (Hoops et al., 2006). These simulators are typically based on ordinary differential equation (ODE) solvers. They are highly efficient and can handle various complexities in SBML specifications, such as stiff equations arising from different reaction timescales and algebraic constraints defined by rules. In this work, we use Tellurium (Medley et al., 2018), a python-based library using libRoadRunner as the backend simulation engine.

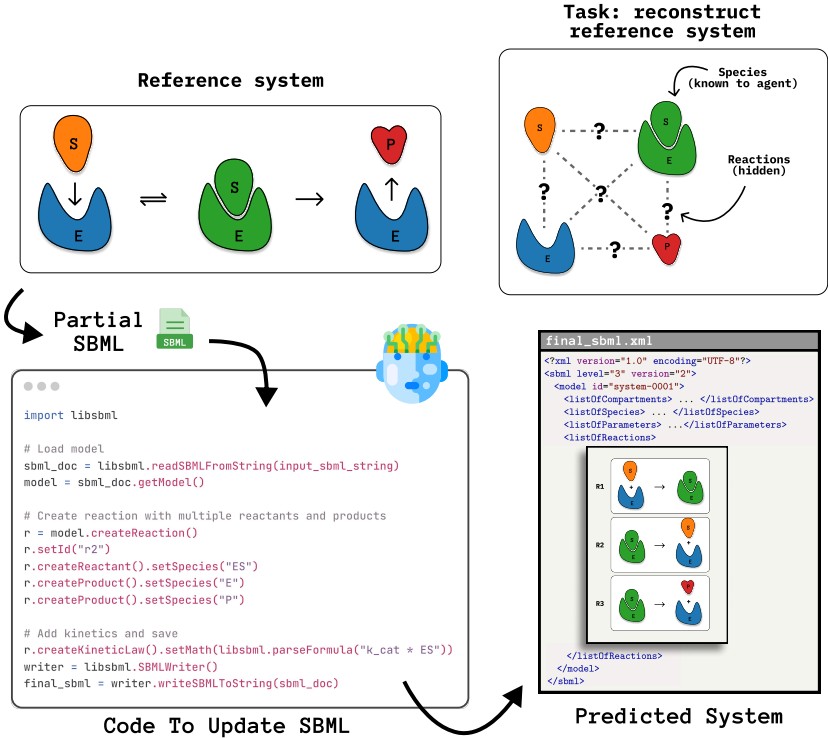

Figure 3: **Open-ended scientific iteration** Agents must discover all components of missing reactions and can update the SBML models using Python code (left). SBML models are encoded as structured XML files, with reactions consisting of five components: reactants, products, modifiers, kinetic laws and parameters (right).

# 3 SCIGYM Benchmark

## 3.1 Data Curation

The full curation pipeline is shown in Figure S1, and we describe briefly the major steps below.

**Filtering**. Starting from the curated collection of 1096 peer-reviewed models in the BioModels repository (Malik-Sheriff et al., 2020), we first removed models with no reactions or no species. Next, we removed models containing `rule` and `event` tags. We found that rules could leak information about held-out reactions, artificially inflating scores. On the other hand, agents often got stuck trying to trigger events, which can require complex combinations of interventions that our experimental interface may not support. We classified the models in our benchmark by their curated gene ontology (Ashburner et al., 2000) (GO) terms (Figure S4). Finally, we removed models that failed to parse with libsbml or failed to simulate using Tellurium. This resulted in 350 models in our benchmark. We provide a detailed breakdown of removed instances in Appendix C.1.

**Preprocessing**. In order to prevent memorization of SBML documents by the agent, we preprocessed each of the filtered models by stripping optional metadata fields, shuffling core components, and anonymizing identifiers to a unique 4-character alphanumeric string (see Appendix C.3). This preprocessing step is how we derive an initial partial SBML which the agent has access to, from the reference system which is kept hidden from the agent.

**Determining Simulation Duration**. We found that a large proportion of the models in the BioModels database did not provide an appropriate simulation timescale in order to observe the characteristic behavior of the mechanism. Thus, we systematically analyzed each model in our benchmark and chose a simulation duration that either enabled the system to reach a steady state or a maximum duration of 10000 seconds (see Appendix C.2).

## 3.2 Framework

**Task.** As illustrated in Figure 1, SBML models provide a general framework to simulate the scientific discovery process: an SBML model is used to generate time-series data; the agent is tasked with discovering the underlying reference system by designing experiments and analyzing their results. One can design tasks of varying difficulty by modulating the agent's access to the system (what species can be observed and what perturbations are permitted). We decided to focus on a fully-observed reaction discovery task, but SCIGYM is easily extended to any task or degree of observability that SBML simulators can support. We allowed the agent to fully observe the concentration time series of all `species` and asked it to discover the missing `reaction` tags connecting `species` in the system. This task mirrors fundamental research problems in biology, such as inferring gene regulatory networks from Perturb-seq experiments (Dixit et al., 2016) or reconstructing cellular signaling pathways from spatial transcriptomics data (Papin et al., 2005).

Starting with a reference SBML $m_{\text{ref}} = (\mathcal{S}, \Theta, \mathcal{R}, \mathcal{T})$, the agent is given a partial SBML $m_{\text{hyp}} = (\mathcal{S}, \Theta', \emptyset, \mathcal{T})$ where the reactions $\mathcal{R}$ have been removed completely and the parameter set $\Theta'$ contains all parameters except those that exclusively appear in a reaction's $\theta_i$. Additionally, we remove `initialAssignment`, `functionDefinition`, and `constraint` tags that reference the removed reactions to prevent information leakage. The agent is given all units, compartments, and species from the original model. The agent's task is to recover the missing reactions $\mathcal{R}$ in $m_{\text{ref}}$.

**Agent.** We implemented a ReAct-style agent (Yao et al., 2023)using a Thoughts-Actions-Observations framework due to its simplicity for comparing performance across different models. The agent is required to articulate its reasoning process before taking actions, structuring its response in markdown format with `## Thoughts` and `## Action` sections. The environment's outputs (code execution and experiment results) are then added as `## Observation` sections. In the prompt, we inform the agent that its goal is to discover the underlying biological mechanisms. To maintain an open-ended task that better mirrors actual scientific discovery processes, we deliberately avoid specifying exact evaluation metrics to the agent. The system prompt is provided in Section A.1.

**Action Space.** In each iteration, the agent can choose from three actions: writing code, conducting experiments, or submitting the model. The agent can choose to write code or conduct experiments or both within a single iteration. However, once the agent chooses to submit the model, no further experiments or code execution is permitted. Each action is signaled with a corresponding markdown subsection, with specific response formats detailed in the system prompt.

**Coding.** We provide a Python execution environment that enables the agent to analyze experimental results. When the agent chooses to write code, the environment executes it and returns stdout output or stderr messages for any errors. The agent is instructed in the system prompts to use print statements for important outputs that should be communicated back to its reasoning process. The environment maintains state as global variables in

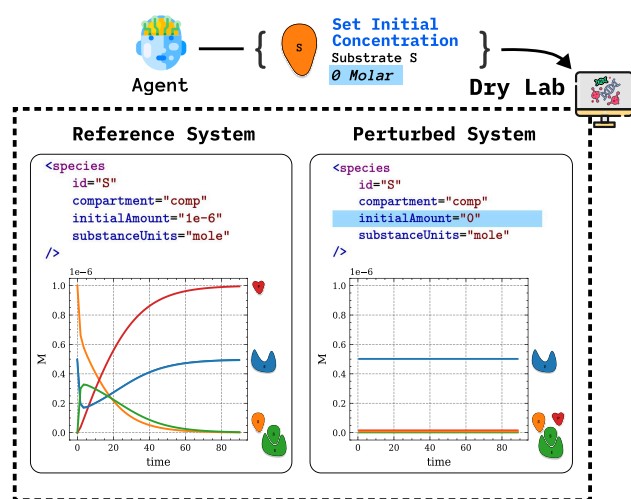

Figure 4: **SBML models serve as data simulators for various experimental perturbations.** The agent proposes perturbations to the system, such as changing its initial conditions and can receive the observed data of the perturbed system.

its Python shell between iterations, but the agent must explicitly choose which variables to store for future access. This design allows the agent to reuse some important data structures especially large objects like pandas DataFrames. The agent has access to a specified set of libraries and customized functions. A customized tool documentation is provided with docstrings (example in Section A.2). In

Table 1: **Pro models outperform their mini counterparts in SCIGYM, with Gemini dominating in both size categories.** We report the average Simulation Trajectory Error (STE) and Reaction Matching Score (RMS) across all benchmark instances. Bold values indicate best performance.

| Model | STE ↓ | RMS ↑ | | | | | |
| | | With Modifiers | | | Without Modifiers | | |
| | | Precision | Recall | F1 | Precision | Recall | F1 |
|---|---|---|---|---|---|---|---|
| Gemini-2.5-Flash | 0.4181 | 0.1527 | 0.1071 | 0.1217 | 0.2399 | 0.1839 | 0.2005 |
| GPT-4.1-mini | 0.6007 | 0.1516 | 0.1253 | 0.1320 | 0.2530 | 0.2313 | 0.2322 |
| Claude-3.5-Haiku | 0.6281 | 0.0858 | 0.0421 | 0.0530 | 0.1454 | 0.0805 | 0.0987 |
| Gemini-2.5-Pro | **0.3212** | **0.2138** | 0.1664 | **0.1817** | **0.3781** | **0.3219** | **0.3383** |
| GPT-4.1 | 0.4611 | 0.2067 | 0.1597 | 0.1740 | 0.3517 | 0.2888 | 0.3038 |
| Claude-3.7-Sonnet | 0.3615 | 0.1780 | **0.1698** | 0.1688 | 0.3160 | 0.3170 | 0.3047 |

our experiments, we provide a wrapper around the Tellurium simulator as the customized functions to help them simulate the proposed mechanisms.

**Experiments.** The agent is provided with a list of permitted experimental perturbations, detailed in an experiment manual (example in Section A.3). This documentation explains the effects of available perturbations and specifies the JSON format for requesting each type. In our experiments, we allow agents to use one simple type of perturbations: changing the initial concentrations of a species, with a detailed discussion in Section 5. When the agent requests an experiment, our environment applies the specified perturbation to the SBML model (using libSBML libraries), simulates the modified system, and returns time-series data to the agent. The resulting time-series data is stored in a dictionary where the key is the iteration number and the value is the pandas DataFrame. Each dataset remains accessible throughout the session, allowing the agent to reference previous experimental results. The environment also provides a summary of each dataset in the observation output, giving the agent an overview of the experimental results to inform its subsequent actions.

**Model Submission.** The agent may submit its final model and conclude the discovery process at any point, subject to a maximum iteration limit (20 in our benchmark). To submit a model, the agent must provide a code block that defines the final model in a variable named `finalsbml`. The agent has direct access to the initial incomplete SBML structure `inputsbml` and can reference it in their code. This approach encourages the agent to use libSBML libraries for model manipulation rather than manual SBML construction. If the submitted model is invalid or cannot be simulated, we allow for 3 additional debugging iterations before evaluating against the incomplete SBML structure.

### 3.3 Evaluation Metrics

We developed three complementary metrics to evaluate how well an agent recovers the underlying SBML model. Our evaluation is designed to balance structural and dynamical similarity. However, it should be noted that these metrics do not capture all aspects of SBML equivalence. For example, we do not explicitly evaluate whether the proposed kinetic laws match the ground truth formulations. For formal definitions of these metrics, see Appendix B.2.

**Network Topology Score (NTS).** To assess success in recovering the correct system topology, we measure pairwise interactions between all species in the system and calculate F1 score. If identical relationships between the same species appear in multiple reactions, we count them only once.

**Reaction Matching Score (RMS).** We evaluate the agent's ability to recover the structures of ground truth reactions. We consider two reactions "matched" if they have identical sets of reactants and products. In the second, more stringent version, we additionally require matching modifiers.

**Simulation Trajectory Error (STE).** This metric evaluates how accurately the predicted system matches the dynamic behavior of the true system by comparing time-series trajectories of each species. We employ the Symmetric Mean Absolute Percentage Error (SMAPE) (Makridakis, 1993), which normalizes errors relative to magnitude.

Table 2: **All models struggle significantly more with identifying modifier relationships compared to reactant-product.** We report network topolgy performance (NTS precision and recall) across different types of edges in reaction networks.

| Metric | Relationship | Gemi-M | Gemi-P | GPT-M | GPT-P | Clau-M | Clau-P |
|--------|-------------|--------|--------|-------|-------|--------|--------|
| **Pre** | Reactant-Product | **0.453** | **0.448** | **0.415** | **0.371** | **0.385** | **0.442** |
| | Reactant-Modifier | 0.005 | 0.075 | 0.010 | 0.000 | 0.037 | 0.068 |
| | Modifier-Product | 0.000 | 0.106 | 0.014 | 0.000 | 0.054 | 0.119 |
| **Rec** | Reactant-Product | **0.362** | **0.346** | **0.340** | **0.305** | **0.214** | **0.435** |
| | Reactant-Modifier | 0.001 | 0.032 | 0.002 | 0.000 | 0.016 | 0.042 |
| | Modifier-Product | 0.000 | 0.055 | 0.007 | 0.000 | 0.021 | 0.088 |

## 4  Related Work

**AI Scientist Benchmarks.** Most existing benchmarks have either focused on evaluating a single skill in the scientific discovery pipeline (Laurent et al., 2024; Narayanan et al., 2024) or the one-shot ability of LLMs to solve scientific tasks (Newsham et al., 2025). To the best of our knowledge, no previous work evaluating the scientific method end-to-end has effectively challenged LLMs to perform open-ended biological discovery research using experimental perturbations. DiscoveryWorld (Jansen et al., 2024) evaluates protein outlier detection and chemical optimization in a fixed, observed dataset. That is, unlike SCIGYM, DiscoveryWorld does not evaluate the LLM's ability to discover the underlying mechanisms of action in a system —it only asks the agent to reproduce the observed data. Moreover, the evaluation pipeline used in BioDiscoveryAgent (Roohani et al., 2024) only deals with genetic perturbations from a fixed, predefined dataset, rather than a general model that can evaluate open-ended hypothesis from the agent. Finally, there exist other benchmarks exploring the discovery of equations (Shojaee et al., 2025; Ma et al., 2024; Romera-Paredes et al., 2024) and chemicals (Sprueill et al., 2024a) with LLMs and simulated data. This line of work does not allow the agent to perturb the system and feedback is given through fixed reward functions, a fundamentally different pipeline from the open-ended setting we explore in SCIGYM.

**Causal Discovery Benchmarks.** Causal discovery seeks to identify cause-effect relationships between variables from observational or, when possible, interventional data (Tian & Pearl, 2013). Many recent efforts have applied LLMs to causal discovery tasks with varying degrees of success (Long et al., 2023; Jiralerspong et al., 2024). This has motivated the development of causal reasoning benchmarks across a wide-range of scientific domains including gene regulatory networks (Chevalley et al., 2023), clinical disease modeling (Abdulaal et al., 2024), and causal question answering for coding and math (Wang, 2024). SCIGYM differs from existing causality benchmarks in two main ways: 1) SCIGYM requires active planning and design of intervention experiments rather than passive analysis of a static dataset or context; 2) SCIGYM demands a more comprehensive skill-set to succeed, requiring agents to combine domain knowledge, data analysis skills, and coding ability to effectively discover biological mechanisms —not just their structural relationships.

**Simulations in AI for Science.** When the cost of obtaining experimental data in the real world is prohibitively expensive, simulations offer a cheaper alternative, albeit at the risk of a simulation-to-reality gap. In the biochemical sciences, simulation tools such as molecular dynamics (Hollingsworth & Dror, 2018) and quantum computational chemistry (McArdle et al., 2020) have provided valuable datasets leading to significant advances in drug discovery (Kitchen et al., 2004) and molecular modeling (Ramakrishnan et al., 2014). Furthermore, many scientific discovery benchmarks incorporate simulations into their workflow to evaluate AI systems without the constraints of physical experimentation. ChemReasoner (Sprueill et al., 2024b) provides quantum-chemical feedback to discover active catalysts while Ma et al. (2024) use differentiable simulation to fit the coordinates of LLM-proposed molecules. Despite these advances, most simulations in AI benchmarks fail to capture the iterative, open-ended nature of real-world scientific discovery. Our benchmark addresses this gap by utilizing structured SBML documents that are manually curated with experimental support to create the drylab-in-the-loop.

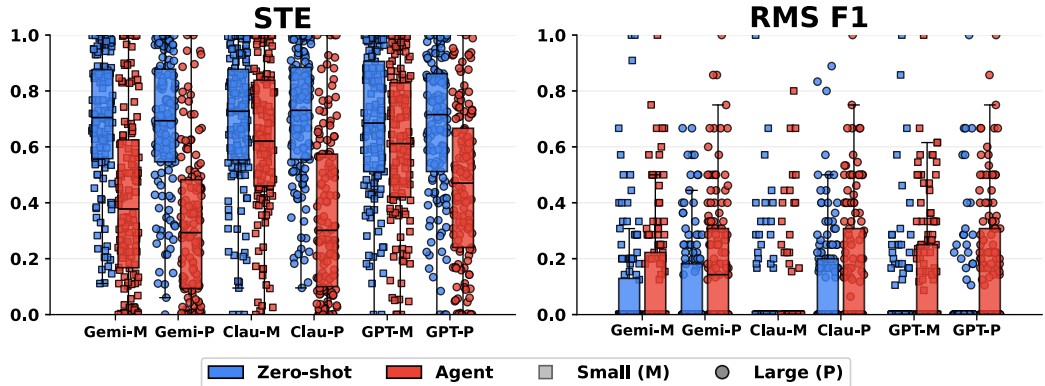

Figure 5: **All agents outperform their respective zero-shot baselines.** We compare each agent's performance with zero-shot prompting. Each marker represents performance on one system, with circles representing pro models and squares representing mini models. Blue shows zero-shot performance while red shows agent performance.

# 5 Experiments

We evaluated six models from three frontier model families, each represented by their professional and mini variants: Anthropic's Claude-3.7-Sonnet-20250219 (P) and Claude-3.5-Haiku-20241022 (M); Google's Gemini-2.5-Pro-Preview-03-25 (P) and Gemini-2.5-Flash-Preview-04-17 (M); and OpenAI's GPT-4.1-2025-04-14 (P) and GPT-4.1-Mini-2025-04-14 (M). These variants are categorized based on their API pricing tiers. Due to computational resource constraints, we restricted our evaluation to SCIGYM-small, with a total of 137 tasks.

**Perturbations.** For our main experiments, we permitted one type of experimental perturbation: changing the initial concentration of a specified species to a designated amount. We selected this approach as it represents a common and cost-effective perturbation in biological research. For example, researchers might alter initial protein concentrations to observe downstream effects on a signaling pathway. This perturbation type, when strategically designed, provides substantial information about species relationships, reaction types, kinetic laws, and reaction rates. In an ablation study, we evaluated a second perturbation type: species knockout, which completely removes a species from the system. We used this condition to assess whether models could improve performance when given more powerful experimental tools. The results are shown in Appendix E.1. This style of perturbation has two main limitations: 1) they may not always be biologically plausible. For example, scientists may not have the equipments to change initial concentrations of the requested species to the specified amount. 2) our perturbations cannot guarantee complete system recovery of every SBML model (Villaverde et al., 2016). However, this mirrors real-world biological research where scientists often work with limited experimental tools.

## 5.1 Results

**Do models learn from experiments?** As an initial validation, we implemented a baseline direct prompting of the models. For fair comparison, we employed similar ReAct prompts across conditions, excluding only the experiment and tools components. We also permitted three additional debugging rounds in the baseline condition. Results shown in Figure 5 demonstrate that all models achieved lower STE and higher RMS when using the SCIGYM framework compared to zero-shot prompting. This confirms that models are effectively extracting and incorporating information from ex-

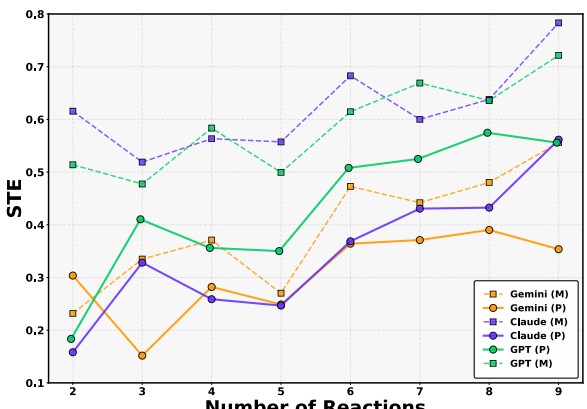

Figure 6: **Agents' performance decline as the underlying system's complexity grows.**

perimental results during the scientific discovery process.

**Do pro models demonstrate superior scientific discovery capabilities?** Table 1 presents the average performance of all six models across all systems. Detailed results per system can be found in Section F. Our results show that across all model families, pro variants consistently outperform their mini counterparts, with both lower STE errors and higher RMS scores. This performance difference suggests that success in the scientific discovery process benefits from the enhanced capabilities of models. Among all models evaluated, Gemini-Pro achieved the best overall performance. Gemini-Mini also demonstrated superior performance among the mini models.

**How does model performance scale with the complexity of the underlying mechanism?** We plot the performance of each model on instances aggregated by the number of reactions and species in Figure 6. We find that as system complexity grows, the simulation error consistently increases across all models tested. For example, both Claude-Sonnet and GPT-4.1 show error increases from close to 0.1 to 0.55 when the number of reactions increases from 2 to 10. This demonstrates that agents struggle increasingly with larger and more complex biological systems.

**How does performance vary across different types of species relationships?** We analyzed the models' ability to discover different types of relationships in biological systems. We categorized relationships based on the roles of connected species: reactant/product, reactant/modifier, and product/modifier connections. For each category, we computed average precision, recall, and F1 scores of NTS. Table 2 shows that all models performed substantially better at discovering reactant-product relationships compared to relationships involving modifiers. Claude-Sonnet achieved the highest F1 score across all relationship types, yet its score for reactant-product relationships was more than 5 times higher than for modifier-related connections. This performance gap is expected, as identifying modifiers requires more targeted experiments to test how specific species affect reaction rates rather than simply observing which species are consumed or produced.

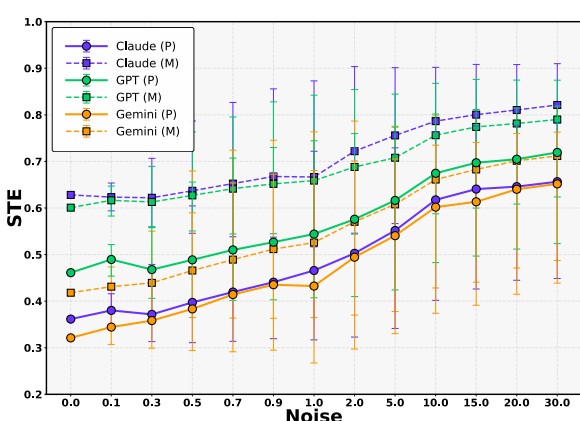

Figure 7: **Agents' proposed systems are sensitive to perturbing initial conditions.**

**How robust are the proposed systems?** We conducted analyses to determine whether agents' proposed SBML models generalize to different initial conditions or overfit to the specific experimental data. We added Gaussian noise (mean zero, variance proportional to the original concentration) to the initial concentrations. We then compared the STE between proposed models and the true system under these perturbed conditions. Figure 7 shows that STE increases consistently with noise levels across all models. This degradation in performance under perturbations indicates that agents' proposed mechanisms overfit to the specific experimental data to some degree rather than capturing the underlying biological system's fundamental properties.

## 6 Conclusion and Limitations

We introduced SCIGYM, a benchmark that evaluates an LLM's scientific discovery ability through a complete discovery cycle, with emphasis on experiment design and analysis. Our work addressed the challenge of collecting experimental data by using formal biology models constructed by biologists to create a dry lab in the loop. These models represent established knowledge of biological systems with precise mathematical formulations of reactions, species interactions, and kinetics. In creating SCIGYM we identified the following limitations:

**Simulation to Reality Gap.** Although the SBMLs in SCIGYM are peer-reviewed models of real-world biological mechanisms, many simplifications are made in order to transform these into systems

that are possible to represent with an ODE. Therefore, conclusions derived from SCIGYM may be systematically biased in subtle ways and not fully generalize to real-world settings. There is also **Lack of realistic noise.** The ODE-based SBMLs we selected for SCIGYM gave us a realistic level of complexity, but they do not contain any noise in the experimental simulations. As such, we are not able to measure the effect of noise level on LLM's ability to recover the underlying system. In a similar vein, our simulations always provide comprehensive time-series data with many sampled timepoints, and we did not study how measurement sparsity impacts performance.

**Simplification of SBML Representations.** In the curation pipeline of SCIGYM, we filter out SBML models with more complex components like `rules` and `events`. This restricts the set of biological systems we can evaluate and possible experiment perturbations we can design. SCIGYM does not make full use of the richness of the SBML standard. For example, we are not specifically interrogating the `kineticLaw` equations when comparing `reactions`. Thus, there is significant scope to increase the richness and impact of SCIGYM.

Despite these limitations, we believe that SCIGYM realizes an interesting new direction for AI scientist benchmarks. To our knowledge, SCIGYM is the first benchmark to evaluate LLMs on the full cycle of scientific experimentation, and makes possible the study of agentic scientific decision-making. SCIGYM is also a framework which can improve over time. As systems biologists continue to make discoveries in their domain and add to the BioModels database, SCIGYM will also become more robust and potentially expand to cover more, different, experimental modalities. For experimental protocols that can be precisely specified and simulated, SCIGYM can easily be extended.

Our benchmarking results on six frontier LLMs are immediately relevant to LLM-based autonomous experiment planning in self-driving labs, where selecting the right AI scientist to put in-the-loop has the potential to enhance search efficiency, lower operational costs, and accelerate scientific discovery. Looking forward, our framework produces artifacts: logs from best performer agents could be used to improve future agent's performance. Examples of successful strategies may aid reasoning by chain-of-thought prompting Wei et al. (2023) or be useful data to improve agents via finetuning Llama Team (2024). We hope that our work introduces a valuable testbed that can continue to evolve to better capture the complexities of real scientific discovery.

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

# Contents of Appendices

# A Agent Framework

## A.1 System Prompts

```
You are a biologist investigating a biological system. Your goal is
    to discover the biological mechanisms missing from your model by
    designing experiments and analyzing results. You must ultimately
    express your findings as a complete SBML model that accurately
    represents the biological system. Your final model will be
    evaluated based on how accurately it represents the true
    biological system.

Your final model will be evaluated by its similarity with the actual
    system under different perturbations, so discovering the true
    underlying mechanisms rather than overfitting to observed data is
     crucial.

# Action

Each time, you can choose one of the three actions:

1. Request Experiments: You can request experiments to gather data
    from the true biological system you are studying. You can also
    choose to perturb the system and see how the system responds.
    This will help you better understand the mechanism of the system.

2. Write Code: You have access to a Python environment to run
    analysis. You can use several scientific computing libraries and
    customized functions. Your code is executed as provided, so
    ensure your syntax is correct.

3. Submit the model: You can choose to submit the model and end the
    process if you think your hypothesis completely explains the
    mechanism.

<!-- BEGIN EXPERIMENTAL_ACTIONS -->
<!-- END EXPERIMENTAL_ACTIONS -->

## Code Execution

For your code, print the results you want to see, and we will provide
     them for you. However, ensure your print content isn't too large
    , as large outputs will be truncated. For large variables like
    long arrays or dataframes, you can store them using the '
    shared_variables' and access them in future sessions:

* 'shared_variables.add(variable_name, val)': Store a variable for
    future access
* 'shared_variables.access(variable_name)': Retrieve a previously
    stored variable

### Libraries

You are allowed to import the following libaries in your code: 'numpy
    ', 'pandas', 'math', 'scipy', 'sklearn', 'libsbml'

### Global variable access

- input_sbml_string (str): Initial incomplete SBML model
```

```
- experiment_history (Dict[str, pd.DataFrame]): Time-series data for
    all experiments
- shared_variables: Storage for all variables you've added in
    previous code executions

**Important** You can access these variables directly in your code.
    You can assume they are global variables provided to you.

### Customized Functions

You can also call the follow functions in your code.

<!-- BEGIN CUSTOMIZED_FUNCTIONS -->
<!-- END CUSTOMIZED_FUNCTIONS -->

## Add reactions using libsbml

'''python
# Example of adding a reaction to an SBML model using libSBML
import libsbml

# Assuming we already have an SBML string loaded
sbml_doc = libsbml.readSBMLFromString(input_sbml_string)
model = sbml_doc.getModel()

# Create a new reaction
reaction = model.createReaction()
reaction.setId("reaction1")
reaction.setReversible(False)
reaction.setFast(False)  # Required in SBML Level 3

# Add a reactant
reactant = reaction.createReactant()
reactant.setSpecies("A")  # Species ID
reactant.setStoichiometry(1.0)
reactant.setConstant(False)  # Required in SBML Level 3

# Add a product
product = reaction.createProduct()
product.setSpecies("B")  # Species ID
product.setStoichiometry(1.0)
product.setConstant(True)  # Required in SBML Level 3

# Write the updated SBML
writer = libsbml.SBMLWriter()
updated_sbml = writer.writeSBMLToString(sbml_doc)
'''

# Submit the model

If you want to submit the model and end the process, put your final
    model as a string variable called 'final_sbml' in your python
    code. It is recommended using libsbml to modify '
    input_sbml_string' rather than write the entire xml on your own.

# Response Format
```

```
Your response should follow thought -action framework in markdown
    formats. You should have a thoughts section followed by an action
     section.

"""
## Thoughts
write down your thoughs here.

## Action

### Code
Include this if you want to write codes. Put your code in a python
    block. You can only include one code block in each response.
```python
import numpy as np
import pandas as pd
```

### Experiment
Include this if you want to request experiments. Put your experiment
    configuration in a json block. You can only include one json
    block in each response.
```json
{
    "action": "",
    "meta_data": {}
}
```

### Submit
Include this if you want to submit the model and end the process. Put
     your final model as a string variable called `final_sbml` in
     your python code.
```python
import libsbml
final_sbml=...
```
"""
```

## A.2   Tool Manual

```python
def simulate(sbml_string: str) -> pd.DataFrame:
    """
    Simulates an SBML model and returns time series data.

    You can use this function to run simulations on your hypothesis
     model and compare it with the data gathered from the experiments.

    Args:
        sbml_string: an SBML model in xml format

    Returns:
        - A pandas dataframe of time series data for the given sbml
     models (with columns 'Time' and the species ID.)
    """
```

## A.3 Experiment Manual

```
## Available Experiment Actions

### Observe
This experiment runs the system with default settings.

```json
{
    "action": "observe",
    "meta_data": {}
}
```

### change initial concentrations

This perturbation changes the initial concentrations of the given
    species. You cannot change the concentration of boundary and
    constant species.

```json
{
     "action": "change_initial_concentration",
     "meta_data": {
         "id_species1": 0.2, // Set the initial concentration of
    species id_species1 to 0.2.
         "id_species2": 0.5
         // Only include the id of the species you want to modify. Any
     species not listed will keep their default values
     }
}
```
```

# B Evaluation Metrics

In this section, we provide additional details on the evaluation metrics used in the paper.

## B.1 Network Topology Scores

For each SBML model, we construct a directed graph $G = (\mathcal{S}, \mathcal{E})$ where nodes $\mathcal{S}$ represent species and edges $\mathcal{E}$ represent direct relationships between species in reactions. An edge $(s_i, s_j) \in \mathcal{E}$ exists if species $s_i$ appears as a reactant and $s_j$ appears as a product in any reaction. This score focuses on the recovery of existence of pairwise interactions between species, without requiring all species in a reaction to be correctly identified.

## B.2 Reaction Recovery Scores

The language model is shown all the species in the SBML file and is explicitly tasked with recovering deleted reactions. Thus, we can evaluate its performance using standard classification metrics if we define a binary function $f : R \times R \to \{0, 1\}$ that indicates whether two reactions are equivalent. Formally, let $\mathcal{R}_{\text{true}} = \{R_1, R_2, \ldots, R_n\}$ denote the set of ground truth reactions and let $\mathcal{R}_{\text{pred}} = \{R'_1, R'_2, \ldots, R'_m\}$ denote the set of predicted reactions. Given an equivalence function $f$, we compute the true positive (TP) count, the false positive (FP) count, and the false negative (FN) count.

$$TP = \mathcal{R}_{\text{pred}} \cap \mathcal{R}_{\text{true}} = \{R' \in \mathcal{R}_{\text{pred}} | \exists R \in \mathcal{R}_{\text{true}} : f(R', R) = 1\}$$
$$FP = \mathcal{R}_{\text{pred}} - \mathcal{R}_{\text{true}} = \{R' \in \mathcal{R}_{\text{pred}} | \forall R \in \mathcal{R}_{\text{true}} : f(R', R) = 0\}$$
$$FN = \mathcal{R}_{\text{true}} - \mathcal{R}_{\text{pred}} = \{R' \in \mathcal{R}_{\text{pred}} | \forall R' \in \mathcal{R}_{\text{pred}} : f(R', R) = 0\}$$

Using these counts, we report the precision, recall, and F1 score as follows:

$$\text{Precision} = \frac{|TP|}{|TP| + |FP|} \qquad \text{Recall} = \frac{|TP|}{|TP| + |FN|} \qquad \text{F1} = 2 \cdot \frac{\text{Precision} \cdot \text{Recall}}{\text{Precision} + \text{Recall}}$$

Note how RMS differs from NTS: NTS focuses on species relationships at a coarser level, awards partial credit for partial relationships, and does not account for multiple relationships between species.

## B.3 Symmetric Mean Absolute Percentage Error (SMAPE)

For ground truth time-series $y$ and predicted series $\hat{y}$ of length $N$, SMAPE is calculated as $\text{SMAPE}(y, \hat{y}) = \frac{1}{N} \sum_{i=0}^{N-1} \frac{|y_i - \hat{y}_i|}{|y| + |\hat{y}|}$. We then average SMAPE across all species. We evaluate models under both original and perturbed initial conditions to test whether they capture underlying mechanistic principles or merely overfit specific experiment data.

# C  Benchmark Curation Details

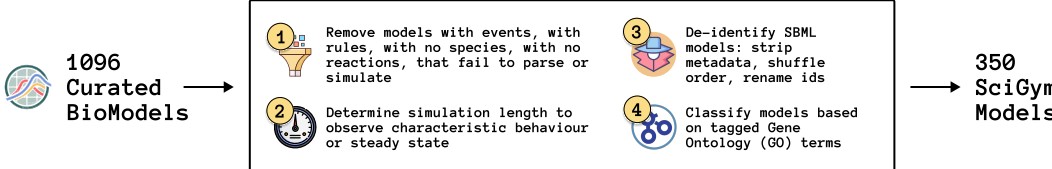

Figure S1: SCIGYM benchmark curation pipeline

## C.1  Filtering BioModels

Among the 1,096 manually curated systems on BioModels, we filter out models that:

1. Can not be parsed by `libsbml` (49)
2. Can not be simulated with `Tellurium` (19)
3. Have no reactions (106)
4. Have no species (5)
5. Have events (145)
6. Have rules (422)

This resulted in 350 SBML models out of the original 1,096 manually curated instances.

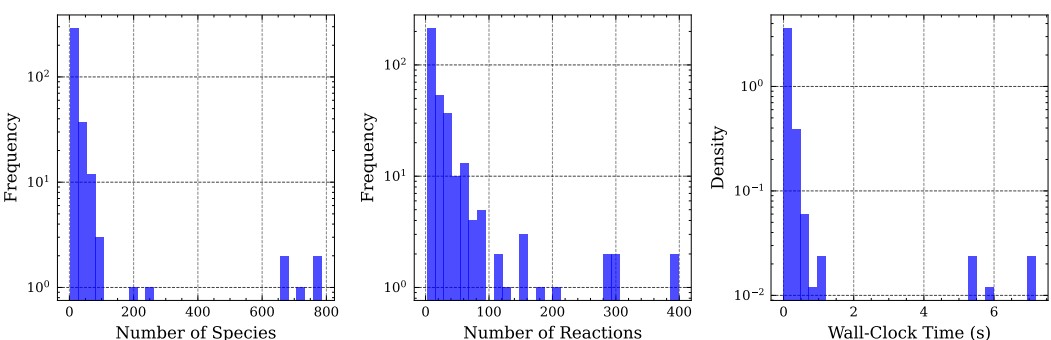

Figure S2: Number of species (left), reactions (middle), and simulation wall-clock time (right) for filtered BioModels.

## C.2  De-identifying SBML files

To prevent language models from memorizing the SBML files, we design a pre-processing pipeline that removes potentially compromising information from the models without affecting their functionality or simulation dynamics. This de-identification pipeline consists of three steps.

**1. Strip metadata.** We remove optional metadata fields from the SBML entities. These include *metaid*, *notes*, *annotations*, *model history*, *control vocabulary*, *dates*, *author information*, and *names**.

**2. Shuffle components.** We shuffle the order of parameters, reactions, species, and compartments.

**3. Renaming ids.** We rename the ids of all components in the model to a unique 4 character alphanumeric identifier.

---

*Species names are kept such that biological entities in the system remain identifiable.

## C.3 Determining Simulation Timescale

Each SBML instance in the BioModels database is accompanied by a SED-ML file that specifies the simulation setup in order to reproduce the curated model. However, 204 out of the 350 filtered models provided an auto-generated template SED-ML file with a default simulation duration of 10 seconds. We found this to be troublesome since some models require a longer simulation budget to observe interesting behavior or reach a steady state. Thus, we developed a pipeline to systematically compute a suitable simulation duration for every filtered BioModel. Our procedure consists of three stages.

**1. Steady-state analysis:** For each model, we first attempt to solve for the steady state of the ODE system (when every species has a rate of change less than $10^{-6}$) using the NLEQ2 algorithm. This step may fail for some models due to numerical issues or the absence of a steady state.

**2. Time-course simulation:** We then simulate the ODE system by integrating over time with the following specifications: We use the integrator defined in the original SED-ML file, defaulting to CVODE if missing. We use a fixed step size of $t = 0.05$ seconds. We simulate until at least one of three termination criteria is met:

- A steady state is reached
- The integrator fails (typically due to stiffness issues or numerical instabilities)
- The maximum simulation budget of 10,000 seconds is exhausted

**3. Final duration determination:** We select the final simulation duration by taking the maximum value among the time required to reach steady state (if successfully solved in step 1), the end time of the time-course simulation, and the original end time specified in the SED-ML file.

This approach ensures that sufficient simulation time is allocated for each model to either reach steady state or exhibit its characteristic dynamic behavior, while still respecting any intentionally specified simulation parameters in the original SED-ML file. We have plotted the simulation duration and maximum species rate of change at simulation endpoint in Figure S3.

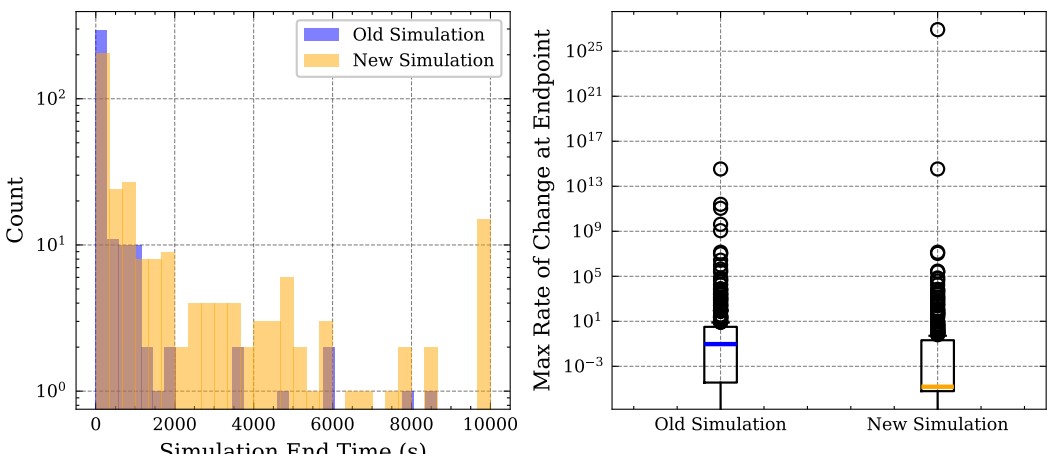

Figure S3: Simulation duration (left) and maximum species rate of change at endpoint (right) before and after processing.

## C.4 Creating Incomplete SBML Models

To create benchmark questions, the final step consists of removing reactions in the SBML files that the language model is tasked with recovering. Below, we describe this deletion operation in greater detail.

**Removing reactions**. To remove a reaction without leaving any traces, we edit the SBML file such that all references to the deleted reaction are also removed. This includes *initial assignments*, *function definitions*, *constraints* that use the deleted reaction's rate to define its own declaration.

**Removing orphaned parameters.** After removing the reactions for each question, we make a final pass through the SBML file and remove orphaned parameters that are no longer referenced. This step ensures that we aren't leaking any unnecessary information to the language model.

To ensure that masked models remain valid SBML files, we only remove core SBML objects[†] that contain nested references to the deleted reaction. For example, if we find a constraint with a clause referencing a deleted reaction, we remove the entire constraint instead of removing only the problematic clause. This guarantees that components are never functionally modified in a way that might mutate their biological significance.

### C.5    Classifying Curated Models

To gain further insight into the types of biochemical networks that our benchmark contains, we perform an analysis of the gene ontology (Ashburner et al., 2000) (GO) terms that are tagged by each SBML model that pass the filtration step. More precisely, for each GO term, we identify its ancestors and attempt to map it onto one of the 318 grandchildren of the `biological_process` root term. Altogether, we were able to map 601 of the 765 filtered models to target GO terms. We provide a breakdown of these terms in Figure S4.

## 350 SciGym Models

☐ unclassified (68)

**cellular process (133)**

signal transduction (62)

metabolic process (56)

| cell cycle (4) | cell development (4) |
| cell death (3) | other (4) |

**bio regulation (67)**

process regulation (44)

quality regulation (13)

molecular function regulation (10)

**stimulus response (33)**

response to biotic stimulus (18)

chemical response (6)

immune response (4)

other (5)

**multicellular process (14)**

developmental process (6)

immune system process (7)

interspecies interaction (5)

homeostatic process (5)

other (3)

Figure S4: **SCIGYM covers a wide range of biology processes**. Classification of SCIGYM models by Gene Ontology (GO) terms.

---

[†]We identify first level objects as the list of core SBML objects in the `libsbml` package.

# D  Additional Details on SBML

## D.1  Major SBML Components

Table S1: SBML Element Definitions, reproduced from Hucka et al. (2019) Level 3 Version 2 Core specification

| SBML Element | Description |
| --- | --- |
| Function | A named mathematical function that may be used throughout the rest of a model. |
| Unit | A named definition of a new unit of measurement. Named units can be used in the expression of quantities in a model. |
| Compartment | A well-stirred container of finite size where species may be located. Compartments may or may not represent actual physical structures. |
| Species | A pool of entities of the same kind located in a compartment and participating in reactions (processes). In biochemical network models, common examples of species include ions, proteins and other molecules; however, in practice, an SBML species can be any kind of entity that makes sense in the context of a given model. |
| Parameter | A quantity with a symbolic name. In SBML, the term parameter is used in a generic sense to refer to named quantities regardless of whether they are constants or variables in a model. SBML Level 3 provides the ability to define parameters that are global to a model as well as parameters that are local to a single reaction. |
| Initial Assignment | A mathematical expression used to determine the initial conditions of a model. This type of object can only be used to define how the value of a symbol can be calculated from other values and symbols at the start of simulated time. |
| Rule | A mathematical expression added to the set of equations constructed based on the reactions defined in a model. Rules can be used to define how a symbol's value can be calculated from other symbols, or used to define the rate of change of a symbol. The set of rules in a model can be used with the reaction rate equations to determine the behavior of the model with respect to time. Rules constrain the model for the entire duration of simulated time. |
| Constraint | A means of detecting out-of-bounds conditions during a dynamical simulation and optionally issuing diagnostic messages. Constraints are defined by an arbitrary mathematical expression computing a true/false value from model symbols. An SBML constraint applies at all instants of simulated time; however, the set of constraints in model should not be used to determine the behavior of the model with respect to time. |
| Reaction | A statement describing some transformation, transport or binding process that can change the amount of one or more species. For example, a reaction may describe how certain entities (reactants) are transformed into certain other entities (products). Reactions have associated kinetic rate expressions describing how quickly they take place. |
| Event | A statement describing an instantaneous, discontinuous change in one or more symbols of any type (species, compartment, parameter, etc.) when a triggering condition is satisfied. |

## D.2  Example SBML Document of Enzyme Process

We provide in Listing 1 the full SBML document of the enzymatic process introduced as an example in Section 2 and used throughout the text.

Listing 1: SBML file for Enzymatic Process

```
1  <?xml version="1.0" encoding="UTF-8"?>
2  <sbml xmlns="http://www.sbml.org/sbml/level3/version2/core" level="3" version="2">
3    <model timeUnits="second" extentUnits="mole">
4      <listOfUnitDefinitions>
5        <unitDefinition id="per_second">
6          <listOfUnits>
7            <unit kind="second" exponent="-1" scale="0" multiplier="1" />
8          </listOfUnits>
```

```xml
        </unitDefinition>
        <unitDefinition id="litre_per_mole_sec">
          <listOfUnits>
            <unit kind="mole" exponent="-1" scale="0" multiplier="1" />
            <unit kind="litre" exponent="1" scale="0" multiplier="1" />
            <unit kind="second" exponent="-1" scale="0" multiplier="1" />
          </listOfUnits>
        </unitDefinition>
      </listOfUnitDefinitions>
      <listOfCompartments>
        <compartment id="comp" spatialDimensions="3" size="1e-14" units="litre"
            constant="true" />
      </listOfCompartments>
      <listOfSpecies>
        <species id="E" compartment="comp" initialAmount="5e-21" substanceUnits="mole"
          hasOnlySubstanceUnits="false" boundaryCondition="false" constant="false" />
        <species id="S" compartment="comp" initialAmount="1e-20" substanceUnits="mole"
          hasOnlySubstanceUnits="false" boundaryCondition="false" constant="false" />
        <species id="P" compartment="comp" initialAmount="0" substanceUnits="mole"
          hasOnlySubstanceUnits="false" boundaryCondition="false" constant="false" />
        <species id="ES" compartment="comp" initialAmount="0" substanceUnits="mole"
          hasOnlySubstanceUnits="false" boundaryCondition="false" constant="false" />
      </listOfSpecies>
      <listOfParameters>
        <parameter id="veq_koff" value="0.2" units="per_second" constant="true" />
        <parameter id="veq_kon" value="1e6" units="litre_per_mole_sec" constant="true"
            />
        <parameter id="vcat_kcat" value="0.1" units="per_second" constant="true" />
      </listOfParameters>
      <listOfReactions>
        <reaction id="veq" reversible="true">
          <listOfReactants>
            <speciesReference species="E" stoichiometry="1" constant="true" />
            <speciesReference species="S" stoichiometry="1" constant="true" />
          </listOfReactants>
          <listOfProducts>
            <speciesReference species="ES" stoichiometry="1" constant="true" />
          </listOfProducts>
          <kineticLaw>
            <math xmlns="http://www.w3.org/1998/Math/MathML">
              <apply>
                <times />
                <ci> comp </ci>
                <apply>
                  <minus />
                  <apply>
                    <times />
                    <ci> veq_kon </ci>
                    <ci> E </ci>
                    <ci> S </ci>
                  </apply>
                  <apply>
                    <times />
                    <ci> veq_koff </ci>
                    <ci> ES </ci>
                  </apply>
                </apply>
              </apply>
            </math>
          </kineticLaw>
        </reaction>
        <reaction id="vcat" reversible="false">
          <listOfReactants>
            <speciesReference species="ES" stoichiometry="1" constant="true" />
          </listOfReactants>
```

```
72          <listOfProducts>
73            <speciesReference species="E" stoichiometry="1" constant="true" />
74            <speciesReference species="P" stoichiometry="1" constant="true" />
75          </listOfProducts>
76          <kineticLaw>
77            <math xmlns="http://www.w3.org/1998/Math/MathML">
78              <apply>
79                <times />
80                <ci> comp </ci>
81                <ci> vcat_kcat </ci>
82                <ci> ES </ci>
83              </apply>
84            </math>
85          </kineticLaw>
86        </reaction>
87      </listOfReactions>
88    </model>
89  </sbml>
```

To translate this SBML model into an ODE, we sum the rates of change implied by the reactions for each species. The `kineticLaw` specifies the rate for each reaction as a function of the participating species. To calculate the differential equation for each species, we multiply its signed stoichiometric coefficient by the rate of change specified in the `kineticLaw`. The sign of the stoichiometric coefficient is -1 if the species is a reactant in the reaction, +1 if it is a product in the reaction, and 0 otherwise.

For this example, we have

$$v\frac{d[E]}{dt} = -(vk_{\mathrm{on}}[\mathbf{E}][\mathbf{S}] - vk_{\mathrm{off}}[\mathbf{ES}]) + (vk_{\mathrm{cat}}[\mathbf{ES}]) \tag{1}$$

$$v\frac{d[S]}{dt} = -(vk_{\mathrm{on}}[\mathbf{E}][\mathbf{S}] - vk_{\mathrm{off}}[\mathbf{ES}]) \tag{2}$$

$$v\frac{d[ES]}{dt} = (vk_{\mathrm{on}}[\mathbf{E}][\mathbf{S}] - vk_{\mathrm{off}}[\mathbf{ES}]) - (vk_{\mathrm{cat}}[\mathbf{ES}]) \tag{3}$$

$$v\frac{d[P]}{dt} = (vk_{\mathrm{cat}}[\mathbf{ES}]) \tag{4}$$

where $v > 0$ is the volume of compartment `comp`.

### D.3   Example SBML Document with Modifier

We provide in Listing 2 a full SBML document that includes a modifier. Modifiers change kinetic laws but are not consumed or produced in the reaction.

Listing 2: SBML example with a modifier

```
1  <?xml version="1.0" encoding="UTF-8"?>
2  <sbml xmlns="http://www.sbml.org/sbml/level3/version2/core" level="3" version="2">
3    <model id="catalyzed_reaction_model">
4      <listOfCompartments>
5        <compartment id="v" spatialDimensions="3" size="1" constant="true"/>
6      </listOfCompartments>
7      <listOfSpecies>
8        <species id="S1" compartment="v" initialAmount="10" hasOnlySubstanceUnits="
             true" boundaryCondition="false" constant="false"/>
9        <species id="S2" compartment="v" initialAmount="0" hasOnlySubstanceUnits="true
             " boundaryCondition="false" constant="false"/>
10       <species id="M" compartment="v" initialAmount="5" hasOnlySubstanceUnits="true"
             boundaryCondition="false" constant="false"/>
11     </listOfSpecies>
12     <listOfParameters>
13       <parameter id="k1" value="0.1" constant="true"/>
14     </listOfParameters>
```

```
15    <listOfReactions>
16      <reaction id="R1" reversible="false">
17        <listOfReactants>
18          <speciesReference species="S1" stoichiometry="1" constant="true"/>
19        </listOfReactants>
20        <listOfProducts>
21          <speciesReference species="S2" stoichiometry="1" constant="true"/>
22        </listOfProducts>
23        <listOfModifiers>
24          <modifierSpeciesReference species="M"/>
25        </listOfModifiers>
26        <kineticLaw>
27          <math xmlns="http://www.w3.org/1998/Math/MathML">
28            <apply>
29              <times/>
30              <ci> v </ci>
31              <ci> k1 </ci>
32              <ci> S1 </ci>
33              <ci> M </ci>
34            </apply>
35          </math>
36        </kineticLaw>
37      </reaction>
38    </listOfReactions>
39  </model>
40 </sbml>
```

To translate this SBML model into an ODE, we sum the rates of change implied by the reactions for each species. The kineticLaw specifies the rate for each reaction as a function of the participating species. To calculate the differential equation for each species, we multiply its signed stoichiometric coefficient by the rate of change specified in the kineticLaw. The sign of the stoichiometric coefficient is -1 if the species is a reactant in the reaction, +1 if it is a product in the reaction, and 0 otherwise.

For this example, we have

$$v\frac{d[\mathbf{S1}]}{dt} = -vk_1[\mathbf{S1}][\mathbf{M}] \tag{5}$$

$$v\frac{d[\mathbf{S2}]}{dt} = vk_1[\mathbf{S1}][\mathbf{M}] \tag{6}$$

$$v\frac{d[\mathbf{M}]}{dt} = 0 \tag{7}$$

where $v > 0$ is the volume of compartment.

# E    Additional Experiments

## E.1    Other types of experiment perturbations

**How do models perform with other types of experimental perturbations?**    We explored an additional experimental capability by enabling agents to perform *species knockouts*, which completely removes a specified species from the system by eliminating all reactions where it participates and setting its initial concentration to zero. Such knockouts are typically extremely expensive or impossible in general biological systems. We also acknowledge this abstraction is our custom design rather than a standard SBML operation. Due to the computational constraints, we only did this ablation study on two Gemini models. We observed modest performance improvements as shown in Table S2. Future work should focus on designing biologically meaningful perturbations and exploring whether more realistic experimental tools can substantially improve discovery performance.

Table S2: **Adding knockout perturbations improve model performance modestly.** We compared Gemini models' performance with and without access to our designed knockout operations. C indicates changing initial concentrations, and K indicates knockout experiments. Models generally demonstrated modest performance improvements when given access to knockout capabilities.

| Model | Experiment | STE $\downarrow$ | RMS $\uparrow$ | |
| --- | --- | --- | --- | --- |
| | | | with mod | w/o mod |
| **Gemi-M** | C | **0.4181** | 0.1217 | 0.2005 |
| | C + K | 0.4276 | **0.1417** | **0.2454** |
| **Gemi-P** | C | 0.3212 | 0.1817 | 0.3383 |
| | C + K | **0.3003** | **0.1923** | **0.3527** |

## E.2    Does SciGym require reasoning like a biologist does?

### E.2.1    Experiments

SciGym uses models studied by actual biologists to represent real biological processes, so reasoning like a biologist would be essential to recover these models in SciGym. We designed a controlled experiment to test whether SciGym requires biological knowledge by removing domain context while keeping identical mathematical structure. Results show significant performance drops when biological reasoning is removed, confirming our hypothesis. Furthermore, SciGym explicitly tests the core reasoning skills required for scientists: hypothesis formation, experiment design and data analysis. These skills are essential for scientific discovery and go beyond what's required for standard ODE discovery tasks.

If our benchmark only tests general ODE discovery, then domain context shouldn't matter, which means that models should perform equally well regardless of how we describe the task. To test this, we conducted a controlled experiment where we kept the exact same underlying SBML mathematical system but only changed the context in the system prompt. By describing the studied system as other subjects (such as Epidemiology and Ecology) instead of molecular biology, we prevented the model from accessing its knowledge of common biological mechanisms and known gene pathways. More specifically, we have changed the first paragraph of the system prompt:

- For ecology: You are an ecological researcher at an environmental institute that has developed a new modeling platform for complex ecosystem dynamics.
- For Epidemiology: You are an epidemiologist at a public health research institute that has developed a new modeling platform for complex disease transmission systems.

This allows us to isolate to what extent biological reasoning provides a genuine advantage in SciGym. If biological reasoning doesn't matter, performance should be identical across all framings since the mathematical structure remains unchanged. We tested this hypothesis using Gemini-2.5-Flash on all instances:

The result above in Table S3 shows that when the agent does not use biology reasoning (row 2 and 3), their performance drops significantly ($p < 0.05$ across all metrics). This confirms our hypothesis that biology reasoning is essential in our benchmark.

Table S3: Model performance degrades with biological domain context. Results show significantly worse performance when the system is framed as Biology compared to domain-agnostic framings (Ecology, Epidemiology), demonstrating that biological reasoning requirements increase task difficulty beyond pure mathematical discovery.

| Domain Framing | STE | RMS (w/ modifier) | RMS (w/o modifier) |
|---|---|---|---|
| Biology (Original) | 0.4181 | 0.1217 | 0.2005 |
| Ecology | 0.4382** | 0.0623** | 0.1023** |
| Epidemiology | 0.4417** | 0.0761** | 0.1241** |

### E.2.2  Discussions

As we have mentioned, SBML models in SciGym are designed by the systems biologists to represent the real biology process. While it might theoretically be possible to discover these systems through brute-force mathematical exploration, our benchmark limits agents to 20 interaction rounds. In order to succeed within a limited interaction round, we think agents must leverage biological priors like mass action kinetics (rather than testing arbitrary functional forms), common regulatory patterns (feedback loops, competitive inhibition), and systematic perturbation strategies to efficiently navigate the vast space of hypotheses. Without this biological knowledge, agents would likely exhaust their interaction budget on implausible hypotheses, failing to discover the correct mechanisms in time.

Most importantly, SciGym test skills essential for scientific discovery. In the end, we acknowledge that Scigym involves abstraction, but this is a necessary tradeoff in almost all benchmarks. Creating end-to-end scientific discovery tasks requires abstraction from real-world complexities as actual biological experiments involve extensive human labor, specialized equipment, and months of work. However, we believe we've made the best possible tradeoff (to date) by preserving the core reasoning skills required for scientists: experimental design, hypothesis formation, mechanistic thinking, and iterative model refinement. These skills are essential for biological discovery and go beyond what's required for standard ODE discovery tasks: by requiring an agent to write code, make choices on how to analyze data, and select which experiments to execute, and in what order. Unlike other abstract-ODE benchmarks (LLM-SR) which focus on equation recovery, we focus our evaluation primarily on discovering causal relationships, and SciGym represents the largest dataset to date of such evaluations, making our benchmark a uniquely valuable test of scientific reasoning capability.

### E.3  Do Models have sufficient SBML knowledge for SciGym?

Knowing the syntax of SBML is a requirement to succeed in SciGym, but this challenge exists in virtually every benchmark. It's impossible to completely isolate abstract capabilities from concrete syntactic or domain-specific knowledge. We also designed SciGym to minimize the SBML technical burden. We filtered out advanced SBML features and retained only basic, standardized components (removing complex elements like `<events>` and `<rules>`). Our task primarily requires knowing how to add reactions, a simple operation that we also demonstrate with examples in the system prompt. We hope that this ensures that models can focus on the scientific reasoning aspects rather than getting bogged down in technical SBML complexities.

### E.3.1  Experiment Design

Therefore, we designed a simple zero-shot task to test whether the evaluated LLMs have the necessary SBML knowledge to succeed. In this experiment, the agent is given an incomplete SBML model and its corresponding complete SBML model. The agent is tasked to write a Python script using the libsbml package to add missing components to the incomplete model such that it matches the complete model. The incomplete and complete models are provided to the agent, both in its LLM context and in the scope of the Python script it submits. If errors occur during parsing or execution of the agent's script, we allow up to 4 submission attempts.

Upon successful execution of the agent's script, we extract the modified incomplete SBML model and systematically evaluate its equivalence to the complete version. We consider two models equivalent if they contain the exact same set of species, reactions, kinetic laws, parameters, values, and initial concentrations (this equivalence is sufficient to obtain perfect scores on the *NTS, RMS, STE* metrics). To prevent cheating, we enforce the usage of the libsbml package to interact with SBML variables,

and prohibit direct string manipulation of the SBML input variables. We randomly picked 50 models from the small SciGym split and evaluated the same six models in our paper.

### E.3.2 Experiment Results

| Model | % Equiv | Avg. Iter. | Fail. Mode(s) |
|---|---|---|---|
| claude-3-5-haiku-20241022 | 80 | 2.04 | wrong ID (2), equation (8) |
| claude-3-7-sonnet-20250219 | 100 | 1.16 | N/A |
| gemini-2.5-flash-preview-05-20 | 98 | 1.16 | failed (1) |
| gemini-2.5-pro-preview-03-25 | 100 | 1.71 | N/A |
| gpt-4.1-2025-04-14 | 64 | 2.17 | wrong ID (9), failed (9) |
| gpt-4.1-mini-2025-04-14 | 66 | 2.44 | wrong eq. (1), failed (16) |
| o3-2025-04-16 | 96 | 1.47 | wrong ID (1), failed (1) |
| o4-mini-2025-04-16 | 44 | 2.73 | failed (28) |

Notes on the failure modes:

- Wrong ID → There was a mismatch in the IDs of at least one component
- Wrong equation → There was a mismatch in the kinetic law equations
- Failed → LLM submitted more than 4 responses that failed to parse or run

### E.3.3 Key Findings

- From the results, we first observe that most models possess sufficient SBML knowledge to succeed in our task. If models lacked SBML knowledge, they should achieve near-zero success rates. Given that we used the same incomplete SBML models as those in our SciGym benchmark, these models demonstrate they can effectively manipulate SBML via libsbml for SciGym tasks.
- More importantly, all frontier models (Sonnet, Gemini, o3) achieve near-perfect performance on this task. This implies that the performance bottleneck for frontier LLMs on SciGym stems primarily from scientific discovery aptitude rather than coding ability or SBML language mastery.

End-to-end scientific discovery inherently requires a combination of technical and reasoning skills. We agree that coding is essential to succeed in our benchmark, but this reflects the reality of modern scientific research. Scientists routinely need to write code for data analysis, implement statistical tests, and use domain-specific software tools. The coding requirement forces models to translate scientific hypotheses into testable implementations—a crucial skill in computational biology and scientific research more broadly. Rather than being a confound, this translation step from conceptual understanding to executable code is an essential component of the scientific discovery process we aim to evaluate. Creating a benchmark that isolates pure scientific reasoning while eliminating all technical skills would be less representative of real scientific work.

## F   Detailed Results by BioMD

Table S4: Detailed results for benchmark BIOMD0000000039 across all models.

| Model | STE ↓ | RMS (with modifiers) ↑ | | | RMS (without modifiers) ↑ | | |
|---|---|---|---|---|---|---|---|
| | | Precision | Recall | F1 | Precision | Recall | F1 |
| Gemini-2.5-Flash | 0.2213 | 0.1667 | 0.1429 | 0.1538 | 0.3333 | 0.3333 | 0.3333 |
| Gemini-2.5-Pro | 0.1022 | 0.2500 | 0.1429 | 0.1818 | 0.5000 | 0.3333 | 0.4000 |
| Gpt-4.1-mini | 0.5452 | 0.0000 | 0.0000 | 0.0000 | 0.2500 | 0.1667 | 0.2000 |
| Gpt-4.1 | 0.5729 | 0.2000 | 0.1429 | 0.1667 | 0.2000 | 0.1667 | 0.1818 |
| Claude-3.5-Haiku | 0.1629 | 0.0000 | 0.0000 | 0.0000 | 0.0000 | 0.0000 | 0.0000 |
| Claude-3.7-Sonnet | 0.1298 | 0.0000 | 0.0000 | 0.0000 | 0.2500 | 0.1667 | 0.2000 |

Table S5: Detailed results for benchmark BIOMD0000000039 across all models.

| Model | STE ↓ | RMS (with modifiers) ↑ | | | RMS (without modifiers) ↑ | | |
|---|---|---|---|---|---|---|---|
| | | Precision | Recall | F1 | Precision | Recall | F1 |
| Gemini-2.5-Flash | 0.2213 | 0.1667 | 0.1429 | 0.1538 | 0.3333 | 0.3333 | 0.3333 |
| Gemini-2.5-Pro | 0.1022 | 0.2500 | 0.1429 | 0.1818 | 0.5000 | 0.3333 | 0.4000 |
| Gpt-4.1-mini | 0.5452 | 0.0000 | 0.0000 | 0.0000 | 0.2500 | 0.1667 | 0.2000 |
| Gpt-4.1 | 0.5729 | 0.2000 | 0.1429 | 0.1667 | 0.2000 | 0.1667 | 0.1818 |
| Claude-3.5-Haiku | 0.1629 | 0.0000 | 0.0000 | 0.0000 | 0.0000 | 0.0000 | 0.0000 |
| Claude-3.7-Sonnet | 0.1298 | 0.0000 | 0.0000 | 0.0000 | 0.2500 | 0.1667 | 0.2000 |

Table S6: Detailed results for benchmark BIOMD0000000041 across all models.

| Model | STE ↓ | RMS (with modifiers) ↑ | | | RMS (without modifiers) ↑ | | |
|---|---|---|---|---|---|---|---|
| | | Precision | Recall | F1 | Precision | Recall | F1 |
| Gemini-2.5-Flash | 0.6552 | 0.0000 | 0.0000 | 0.0000 | 0.0000 | 0.0000 | 0.0000 |
| Gemini-2.5-Pro | 0.3252 | 0.0000 | 0.0000 | 0.0000 | 0.0000 | 0.0000 | 0.0000 |
| Gpt-4.1-mini | 0.8718 | 0.1429 | 0.1111 | 0.1250 | 0.1429 | 0.1111 | 0.1250 |
| Gpt-4.1 | 0.7187 | 0.0000 | 0.0000 | 0.0000 | 0.0000 | 0.0000 | 0.0000 |
| Claude-3.5-Haiku | 0.8415 | 0.0000 | 0.0000 | 0.0000 | 0.0000 | 0.0000 | 0.0000 |
| Claude-3.7-Sonnet | 0.3036 | 0.0455 | 0.1111 | 0.0645 | 0.0455 | 0.1111 | 0.0645 |

Table S7: Detailed results for benchmark BIOMD0000000043 across all models.

| Model | STE ↓ | RMS (with modifiers) ↑ | | | RMS (without modifiers) ↑ | | |
|---|---|---|---|---|---|---|---|
| | | Precision | Recall | F1 | Precision | Recall | F1 |
| Gemini-2.5-Flash | 0.9965 | 0.4000 | 0.2857 | 0.3333 | 0.6000 | 0.5000 | 0.5455 |
| Gemini-2.5-Pro | 0.5637 | 0.1667 | 0.1429 | 0.1538 | 0.1667 | 0.1667 | 0.1667 |
| Gpt-4.1-mini | 0.7097 | 0.0000 | 0.0000 | 0.0000 | 0.0000 | 0.0000 | 0.0000 |
| Gpt-4.1 | 0.7679 | 0.0000 | 0.0000 | 0.0000 | 0.1667 | 0.1667 | 0.1667 |
| Claude-3.5-Haiku | 0.8383 | 0.0000 | 0.0000 | 0.0000 | 0.3333 | 0.1667 | 0.2222 |
| Claude-3.7-Sonnet | 0.5688 | 0.1000 | 0.1429 | 0.1176 | 0.2000 | 0.3333 | 0.2500 |

Table S8: Detailed results for benchmark BIOMD0000000044 across all models.

| Model | STE ↓ | RMS (with modifiers) ↑ | | | RMS (without modifiers) ↑ | | |
|---|---|---|---|---|---|---|---|
| | | Precision | Recall | F1 | Precision | Recall | F1 |
| Gemini-2.5-Flash | 0.4545 | 0.3333 | 0.2500 | 0.2857 | 0.3333 | 0.3333 | 0.3333 |
| Gemini-2.5-Pro | 0.3169 | 0.0000 | 0.0000 | 0.0000 | 0.1429 | 0.1667 | 0.1538 |
| Gpt-4.1-mini | 0.4833 | 0.0000 | 0.0000 | 0.0000 | 0.0000 | 0.0000 | 0.0000 |
| Gpt-4.1 | 0.4163 | 0.3333 | 0.3750 | 0.3529 | 0.3333 | 0.5000 | 0.4000 |
| Claude-3.5-Haiku | 0.4833 | 0.0000 | 0.0000 | 0.0000 | 0.0000 | 0.0000 | 0.0000 |
| Claude-3.7-Sonnet | 0.2929 | 0.1667 | 0.1250 | 0.1429 | 0.1667 | 0.1667 | 0.1667 |

Table S9: Detailed results for benchmark BIOMD0000000045 across all models.

| Model | STE ↓ | RMS (with modifiers) ↑ | | | RMS (without modifiers) ↑ | | |
|---|---|---|---|---|---|---|---|
| | | Precision | Recall | F1 | Precision | Recall | F1 |
| Gemini-2.5-Flash | 0.9667 | 0.2500 | 0.3333 | 0.2857 | 0.2500 | 0.3333 | 0.2857 |
| Gemini-2.5-Pro | 0.6670 | 0.2000 | 0.1667 | 0.1818 | 0.2000 | 0.1667 | 0.1818 |
| Gpt-4.1-mini | 0.8494 | 0.5000 | 0.3333 | 0.4000 | 0.5000 | 0.3333 | 0.4000 |
| Gpt-4.1 | 0.7270 | 0.2222 | 0.3333 | 0.2667 | 0.2222 | 0.3333 | 0.2667 |
| Claude-3.5-Haiku | 0.8563 | 0.3333 | 0.1667 | 0.2222 | 0.3333 | 0.1667 | 0.2222 |
| Claude-3.7-Sonnet | 0.9199 | 0.1667 | 0.1667 | 0.1667 | 0.1667 | 0.1667 | 0.1667 |

Table S10: Detailed results for benchmark BIOMD0000000066 across all models.

| Model | STE ↓ | RMS (with modifiers) ↑ | | | RMS (without modifiers) ↑ | | |
|---|---|---|---|---|---|---|---|
| | | Precision | Recall | F1 | Precision | Recall | F1 |
| Gemini-2.5-Flash | 0.5611 | 0.0000 | 0.0000 | 0.0000 | 0.0000 | 0.0000 | 0.0000 |
| Gemini-2.5-Pro | 0.3937 | 0.2857 | 0.2857 | 0.2857 | 0.2857 | 0.2857 | 0.2857 |
| Gpt-4.1-mini | 0.8539 | 0.0000 | 0.0000 | 0.0000 | 0.0000 | 0.0000 | 0.0000 |
| Gpt-4.1 | 0.6358 | 0.3333 | 0.1429 | 0.2000 | 0.3333 | 0.1429 | 0.2000 |
| Claude-3.5-Haiku | 0.8511 | 0.0000 | 0.0000 | 0.0000 | 0.0000 | 0.0000 | 0.0000 |
| Claude-3.7-Sonnet | 0.6764 | 0.1250 | 0.1429 | 0.1333 | 0.1250 | 0.1429 | 0.1333 |

Table S11: Detailed results for benchmark BIOMD0000000068 across all models.

| Model | STE ↓ | RMS (with modifiers) ↑ | | | RMS (without modifiers) ↑ | | |
|---|---|---|---|---|---|---|---|
| | | Precision | Recall | F1 | Precision | Recall | F1 |
| Gemini-2.5-Flash | 0.0040 | 0.0000 | 0.0000 | 0.0000 | 0.0000 | 0.0000 | 0.0000 |
| Gemini-2.5-Pro | 0.0001 | 0.0000 | 0.0000 | 0.0000 | 0.0000 | 0.0000 | 0.0000 |
| Gpt-4.1-mini | 0.1896 | 0.0000 | 0.0000 | 0.0000 | 0.0000 | 0.0000 | 0.0000 |
| Gpt-4.1 | 0.0005 | 0.0000 | 0.0000 | 0.0000 | 0.0000 | 0.0000 | 0.0000 |
| Claude-3.5-Haiku | 0.0257 | 0.0000 | 0.0000 | 0.0000 | 0.0000 | 0.0000 | 0.0000 |
| Claude-3.7-Sonnet | 0.0002 | 0.0000 | 0.0000 | 0.0000 | 0.0000 | 0.0000 | 0.0000 |

Table S12: Detailed results for benchmark BIOMD0000000072 across all models.

| Model | STE ↓ | RMS (with modifiers) ↑ | | | RMS (without modifiers) ↑ | | |
|---|---|---|---|---|---|---|---|
| | | Precision | Recall | F1 | Precision | Recall | F1 |
| Gemini-2.5-Flash | 0.3495 | 0.0000 | 0.0000 | 0.0000 | 0.0000 | 0.0000 | 0.0000 |
| Gemini-2.5-Pro | 0.3401 | 0.2000 | 0.1429 | 0.1667 | 0.2000 | 0.1429 | 0.1667 |
| Gpt-4.1-mini | 0.7310 | 0.0000 | 0.0000 | 0.0000 | 0.0000 | 0.0000 | 0.0000 |
| Gpt-4.1 | 0.4918 | 0.0000 | 0.0000 | 0.0000 | 0.0000 | 0.0000 | 0.0000 |
| Claude-3.5-Haiku | 0.4244 | 0.0000 | 0.0000 | 0.0000 | 0.0000 | 0.0000 | 0.0000 |
| Claude-3.7-Sonnet | 0.6369 | 0.2500 | 0.2857 | 0.2667 | 0.2500 | 0.2857 | 0.2667 |

Table S13: Detailed results for benchmark BIOMD0000000076 across all models.

| Model | STE ↓ | RMS (with modifiers) ↑ | | | RMS (without modifiers) ↑ | | |
|---|---|---|---|---|---|---|---|
| | | Precision | Recall | F1 | Precision | Recall | F1 |
| Gemini-2.5-Flash | 0.0035 | 0.0000 | 0.0000 | 0.0000 | 0.0000 | 0.0000 | 0.0000 |
| Gemini-2.5-Pro | 0.0001 | 0.0000 | 0.0000 | 0.0000 | 0.0000 | 0.0000 | 0.0000 |
| Gpt-4.1-mini | 0.1815 | 0.0000 | 0.0000 | 0.0000 | 0.0000 | 0.0000 | 0.0000 |
| Gpt-4.1 | 0.0005 | 0.0000 | 0.0000 | 0.0000 | 0.0000 | 0.0000 | 0.0000 |
| Claude-3.5-Haiku | 0.0136 | 0.0000 | 0.0000 | 0.0000 | 0.0000 | 0.0000 | 0.0000 |
| Claude-3.7-Sonnet | 0.0003 | 0.5000 | 0.5000 | 0.5000 | 0.5000 | 0.5000 | 0.5000 |

Table S14: Detailed results for benchmark BIOMD0000000079 across all models.

| Model | STE ↓ | RMS (with modifiers) ↑ | | | RMS (without modifiers) ↑ | | |
|---|---|---|---|---|---|---|---|
| | | Precision | Recall | F1 | Precision | Recall | F1 |
| Gemini-2.5-Flash | 0.2022 | 0.0000 | 0.0000 | 0.0000 | 0.2500 | 0.1667 | 0.2000 |
| Gemini-2.5-Pro | 0.2869 | 0.4286 | 0.5000 | 0.4615 | 0.5714 | 0.6667 | 0.6154 |
| Gpt-4.1-mini | 0.3794 | 0.2500 | 0.1667 | 0.2000 | 0.2500 | 0.1667 | 0.2000 |
| Gpt-4.1 | 0.5643 | 0.2857 | 0.3333 | 0.3077 | 0.4286 | 0.5000 | 0.4615 |
| Claude-3.5-Haiku | 0.2413 | 0.0000 | 0.0000 | 0.0000 | 0.0000 | 0.0000 | 0.0000 |
| Claude-3.7-Sonnet | 0.3171 | 0.3333 | 0.5000 | 0.4000 | 0.5556 | 0.8333 | 0.6667 |

Table S15: Detailed results for benchmark BIOMD0000000080 across all models.

| Model | STE ↓ | RMS (with modifiers) ↑ | | | RMS (without modifiers) ↑ | | |
|---|---|---|---|---|---|---|---|
| | | Precision | Recall | F1 | Precision | Recall | F1 |
| Gemini-2.5-Flash | 0.2300 | 0.0000 | 0.0000 | 0.0000 | 0.0000 | 0.0000 | 0.0000 |
| Gemini-2.5-Pro | 0.3577 | 0.0000 | 0.0000 | 0.0000 | 0.0000 | 0.0000 | 0.0000 |
| Gpt-4.1-mini | 0.6403 | 0.0000 | 0.0000 | 0.0000 | 0.0000 | 0.0000 | 0.0000 |
| Gpt-4.1 | 0.5796 | 0.0000 | 0.0000 | 0.0000 | 0.0000 | 0.0000 | 0.0000 |
| Claude-3.5-Haiku | 0.5873 | 0.0000 | 0.0000 | 0.0000 | 0.0000 | 0.0000 | 0.0000 |
| Claude-3.7-Sonnet | 0.1367 | 0.0000 | 0.0000 | 0.0000 | 0.0000 | 0.0000 | 0.0000 |

Table S16: Detailed results for benchmark BIOMD0000000082 across all models.

| Model | STE ↓ | RMS (with modifiers) ↑ | | | RMS (without modifiers) ↑ | | |
|---|---|---|---|---|---|---|---|
| | | Precision | Recall | F1 | Precision | Recall | F1 |
| Gemini-2.5-Flash | 0.2705 | 0.0000 | 0.0000 | 0.0000 | 0.0000 | 0.0000 | 0.0000 |
| Gemini-2.5-Pro | 0.2033 | 0.0000 | 0.0000 | 0.0000 | 0.0000 | 0.0000 | 0.0000 |
| Gpt-4.1-mini | 0.5927 | 0.0000 | 0.0000 | 0.0000 | 0.0000 | 0.0000 | 0.0000 |
| Gpt-4.1 | 0.4703 | 0.0000 | 0.0000 | 0.0000 | 0.0000 | 0.0000 | 0.0000 |
| Claude-3.5-Haiku | 0.6056 | 0.0000 | 0.0000 | 0.0000 | 0.0000 | 0.0000 | 0.0000 |
| Claude-3.7-Sonnet | 0.5290 | 0.0000 | 0.0000 | 0.0000 | 0.0000 | 0.0000 | 0.0000 |

Table S17: Detailed results for benchmark BIOMD0000000084 across all models.

| Model | STE ↓ | RMS (with modifiers) ↑ | | | RMS (without modifiers) ↑ | | |
|---|---|---|---|---|---|---|---|
| | | Precision | Recall | F1 | Precision | Recall | F1 |
| Gemini-2.5-Flash | 0.2929 | 0.0000 | 0.0000 | 0.0000 | 0.0000 | 0.0000 | 0.0000 |
| Gemini-2.5-Pro | 0.2624 | 0.2500 | 0.1250 | 0.1667 | 1.0000 | 0.5000 | 0.6667 |
| Gpt-4.1-mini | 0.6967 | 0.0000 | 0.0000 | 0.0000 | 0.0000 | 0.0000 | 0.0000 |
| Gpt-4.1 | 0.5450 | 0.0909 | 0.1250 | 0.1053 | 0.0909 | 0.1250 | 0.1053 |
| Claude-3.5-Haiku | 0.8307 | 0.3333 | 0.1250 | 0.1818 | 1.0000 | 0.3750 | 0.5455 |
| Claude-3.7-Sonnet | 0.2615 | 0.2500 | 0.1250 | 0.1667 | 1.0000 | 0.5000 | 0.6667 |

Table S18: Detailed results for benchmark BIOMD0000000092 across all models.

| Model | STE ↓ | RMS (with modifiers) ↑ | | | RMS (without modifiers) ↑ | | |
|---|---|---|---|---|---|---|---|
| | | Precision | Recall | F1 | Precision | Recall | F1 |
| Gemini-2.5-Flash | 0.3167 | 0.0000 | 0.0000 | 0.0000 | 0.0000 | 0.0000 | 0.0000 |
| Gemini-2.5-Pro | 0.0021 | 0.5000 | 0.3333 | 0.4000 | 0.5000 | 0.3333 | 0.4000 |
| Gpt-4.1-mini | 0.9322 | 0.0000 | 0.0000 | 0.0000 | 0.0000 | 0.0000 | 0.0000 |
| Gpt-4.1 | 0.1430 | 0.0000 | 0.0000 | 0.0000 | 0.0000 | 0.0000 | 0.0000 |
| Claude-3.5-Haiku | 0.5402 | 0.5000 | 0.3333 | 0.4000 | 0.5000 | 0.3333 | 0.4000 |
| Claude-3.7-Sonnet | 0.0846 | 0.0000 | 0.0000 | 0.0000 | 0.0000 | 0.0000 | 0.0000 |

Table S19: Detailed results for benchmark BIOMD0000000098 across all models.

| Model | STE ↓ | RMS (with modifiers) ↑ | | | RMS (without modifiers) ↑ | | |
|---|---|---|---|---|---|---|---|
| | | Precision | Recall | F1 | Precision | Recall | F1 |
| Gemini-2.5-Flash | 0.7426 | 1.0000 | 0.5000 | 0.6667 | 1.0000 | 0.5000 | 0.6667 |
| Gemini-2.5-Pro | 0.4478 | 0.3333 | 0.2500 | 0.2857 | 0.3333 | 0.2500 | 0.2857 |
| Gpt-4.1-mini | 0.1116 | 0.6667 | 0.5000 | 0.5714 | 0.6667 | 0.5000 | 0.5714 |
| Gpt-4.1 | 0.4069 | 1.0000 | 0.5000 | 0.6667 | 1.0000 | 0.5000 | 0.6667 |
| Claude-3.5-Haiku | 0.3561 | 1.0000 | 0.2500 | 0.4000 | 1.0000 | 0.2500 | 0.4000 |
| Claude-3.7-Sonnet | 0.2657 | 0.1667 | 0.2500 | 0.2000 | 0.1667 | 0.2500 | 0.2000 |

Table S20: Detailed results for benchmark BIOMD0000000156 across all models.

| Model | STE ↓ | RMS (with modifiers) ↑ | | | RMS (without modifiers) ↑ | | |
|---|---|---|---|---|---|---|---|
| | | Precision | Recall | F1 | Precision | Recall | F1 |
| Gemini-2.5-Flash | 0.2671 | 0.0000 | 0.0000 | 0.0000 | 0.0000 | 0.0000 | 0.0000 |
| Gemini-2.5-Pro | 0.6261 | 0.3333 | 0.2000 | 0.2500 | 0.3333 | 0.2000 | 0.2500 |
| Gpt-4.1-mini | 0.2769 | 0.3333 | 0.2000 | 0.2500 | 0.3333 | 0.2000 | 0.2500 |
| Gpt-4.1 | 0.3085 | 0.1667 | 0.2000 | 0.1818 | 0.3333 | 0.4000 | 0.3636 |
| Claude-3.5-Haiku | 0.4243 | 0.0000 | 0.0000 | 0.0000 | 0.0000 | 0.0000 | 0.0000 |
| Claude-3.7-Sonnet | 0.3023 | 0.3333 | 0.4000 | 0.3636 | 0.6667 | 0.8000 | 0.7273 |

Table S21: Detailed results for benchmark BIOMD0000000157 across all models.

| Model | STE ↓ | RMS (with modifiers) ↑ | | | RMS (without modifiers) ↑ | | |
|---|---|---|---|---|---|---|---|
| | | Precision | Recall | F1 | Precision | Recall | F1 |
| Gemini-2.5-Flash | 0.5773 | 0.0000 | 0.0000 | 0.0000 | 0.0000 | 0.0000 | 0.0000 |
| Gemini-2.5-Pro | 0.5918 | 0.3333 | 0.3333 | 0.3333 | 0.3333 | 0.4000 | 0.3636 |
| Gpt-4.1-mini | 0.5829 | 0.3333 | 0.3333 | 0.3333 | 0.3333 | 0.4000 | 0.3636 |
| Gpt-4.1 | 0.3561 | 0.2500 | 0.1667 | 0.2000 | 0.2500 | 0.2000 | 0.2222 |
| Claude-3.5-Haiku | 0.4965 | 0.0000 | 0.0000 | 0.0000 | 0.0000 | 0.0000 | 0.0000 |
| Claude-3.7-Sonnet | 0.4263 | 0.5000 | 0.5000 | 0.5000 | 0.6667 | 0.8000 | 0.7273 |

Table S22: Detailed results for benchmark BIOMD0000000159 across all models.

| Model | STE ↓ | RMS (with modifiers) ↑ | | | RMS (without modifiers) ↑ | | |
|---|---|---|---|---|---|---|---|
| | | Precision | Recall | F1 | Precision | Recall | F1 |
| Gemini-2.5-Flash | 1.0000 | 0.2500 | 0.1667 | 0.2000 | 0.2500 | 0.2000 | 0.2222 |
| Gemini-2.5-Pro | 0.2588 | 0.7500 | 0.5000 | 0.6000 | 0.7500 | 0.6000 | 0.6667 |
| Gpt-4.1-mini | 0.9990 | 0.5714 | 0.6667 | 0.6154 | 0.5714 | 0.8000 | 0.6667 |
| Gpt-4.1 | 0.4786 | 0.1667 | 0.1667 | 0.1667 | 0.3333 | 0.4000 | 0.3636 |
| Claude-3.5-Haiku | 0.9990 | 0.3333 | 0.1667 | 0.2222 | 0.3333 | 0.2000 | 0.2500 |
| Claude-3.7-Sonnet | 0.2440 | 0.6667 | 0.6667 | 0.6667 | 0.6667 | 0.8000 | 0.7273 |

Table S23: Detailed results for benchmark BIOMD0000000184 across all models.

| Model | STE ↓ | RMS (with modifiers) ↑ | | | RMS (without modifiers) ↑ | | |
|---|---|---|---|---|---|---|---|
| | | Precision | Recall | F1 | Precision | Recall | F1 |
| Gemini-2.5-Flash | 0.1836 | 0.5000 | 0.1429 | 0.2222 | 0.5000 | 0.1667 | 0.2500 |
| Gemini-2.5-Pro | 0.1107 | 0.6667 | 0.2857 | 0.4000 | 0.6667 | 0.3333 | 0.4444 |
| Gpt-4.1-mini | 0.3354 | 0.0000 | 0.0000 | 0.0000 | 0.0000 | 0.0000 | 0.0000 |
| Gpt-4.1 | 0.1673 | 0.6000 | 0.4286 | 0.5000 | 0.6000 | 0.5000 | 0.5455 |
| Claude-3.5-Haiku | 0.3201 | 0.0000 | 0.0000 | 0.0000 | 0.0000 | 0.0000 | 0.0000 |
| Claude-3.7-Sonnet | 0.4057 | 0.6000 | 0.4286 | 0.5000 | 0.6000 | 0.5000 | 0.5455 |

Table S24: Detailed results for benchmark BIOMD0000000191 across all models.

| Model | STE ↓ | RMS (with modifiers) ↑ | | | RMS (without modifiers) ↑ | | |
|---|---|---|---|---|---|---|---|
| | | Precision | Recall | F1 | Precision | Recall | F1 |
| Gemini-2.5-Flash | 0.0275 | 0.4000 | 0.4000 | 0.4000 | 0.4000 | 0.5000 | 0.4444 |
| Gemini-2.5-Pro | 0.0163 | 0.5000 | 0.4000 | 0.4444 | 0.5000 | 0.5000 | 0.5000 |
| Gpt-4.1-mini | 0.4203 | 0.2500 | 0.2000 | 0.2222 | 0.5000 | 0.5000 | 0.5000 |
| Gpt-4.1 | 0.0055 | 0.5000 | 0.4000 | 0.4444 | 0.5000 | 0.5000 | 0.5000 |
| Claude-3.5-Haiku | 0.0438 | 0.0000 | 0.0000 | 0.0000 | 0.0000 | 0.0000 | 0.0000 |
| Claude-3.7-Sonnet | 0.4257 | 0.5000 | 0.4000 | 0.4444 | 0.5000 | 0.5000 | 0.5000 |

Table S25: Detailed results for benchmark BIOMD0000000192 across all models.

| Model | STE ↓ | RMS (with modifiers) ↑ | | | RMS (without modifiers) ↑ | | |
|---|---|---|---|---|---|---|---|
| | | Precision | Recall | F1 | Precision | Recall | F1 |
| Gemini-2.5-Flash | 0.4250 | 0.0000 | 0.0000 | 0.0000 | 0.0000 | 0.0000 | 0.0000 |
| Gemini-2.5-Pro | 0.1003 | 0.0000 | 0.0000 | 0.0000 | 0.0000 | 0.0000 | 0.0000 |
| Gpt-4.1-mini | 0.6365 | 0.0000 | 0.0000 | 0.0000 | 0.0000 | 0.0000 | 0.0000 |
| Gpt-4.1 | 0.3599 | 0.0000 | 0.0000 | 0.0000 | 0.0000 | 0.0000 | 0.0000 |
| Claude-3.5-Haiku | 1.0000 | 0.0000 | 0.0000 | 0.0000 | 0.0000 | 0.0000 | 0.0000 |
| Claude-3.7-Sonnet | 0.2660 | 0.0000 | 0.0000 | 0.0000 | 0.0000 | 0.0000 | 0.0000 |

Table S26: Detailed results for benchmark BIOMD0000000224 across all models.

| Model | STE ↓ | RMS (with modifiers) ↑ | | | RMS (without modifiers) ↑ | | |
|---|---|---|---|---|---|---|---|
| | | Precision | Recall | F1 | Precision | Recall | F1 |
| Gemini-2.5-Flash | 0.6407 | 0.2000 | 0.1667 | 0.1818 | 0.6000 | 0.5000 | 0.5455 |
| Gemini-2.5-Pro | 0.3212 | 0.2000 | 0.1667 | 0.1818 | 0.6000 | 0.5000 | 0.5455 |
| Gpt-4.1-mini | 0.4353 | 0.0000 | 0.0000 | 0.0000 | 0.6667 | 0.3333 | 0.4444 |
| Gpt-4.1 | 0.3359 | 0.0000 | 0.0000 | 0.0000 | 0.3333 | 0.1667 | 0.2222 |
| Claude-3.5-Haiku | 0.7506 | 0.0000 | 0.0000 | 0.0000 | 0.0000 | 0.0000 | 0.0000 |
| Claude-3.7-Sonnet | 0.4060 | 0.1667 | 0.1667 | 0.1667 | 0.5000 | 0.5000 | 0.5000 |

Table S27: Detailed results for benchmark BIOMD0000000231 across all models.

| Model | STE ↓ | RMS (with modifiers) ↑ | | | RMS (without modifiers) ↑ | | |
|---|---|---|---|---|---|---|---|
| | | Precision | Recall | F1 | Precision | Recall | F1 |
| Gemini-2.5-Flash | 0.0231 | 0.0000 | 0.0000 | 0.0000 | 0.0000 | 0.0000 | 0.0000 |
| Gemini-2.5-Pro | 0.1208 | 0.0000 | 0.0000 | 0.0000 | 0.0000 | 0.0000 | 0.0000 |
| Gpt-4.1-mini | 0.5512 | 0.0000 | 0.0000 | 0.0000 | 0.0000 | 0.0000 | 0.0000 |
| Gpt-4.1 | 0.3539 | 0.0000 | 0.0000 | 0.0000 | 0.0000 | 0.0000 | 0.0000 |
| Claude-3.5-Haiku | 0.5060 | 0.0000 | 0.0000 | 0.0000 | 0.0000 | 0.0000 | 0.0000 |
| Claude-3.7-Sonnet | 0.0371 | 0.0000 | 0.0000 | 0.0000 | 0.0000 | 0.0000 | 0.0000 |

Table S28: Detailed results for benchmark BIOMD0000000233 across all models.

| Model | STE ↓ | RMS (with modifiers) ↑ | | | RMS (without modifiers) ↑ | | |
|---|---|---|---|---|---|---|---|
| | | Precision | Recall | F1 | Precision | Recall | F1 |
| Gemini-2.5-Flash | 0.2611 | 0.0000 | 0.0000 | 0.0000 | 0.0000 | 0.0000 | 0.0000 |
| Gemini-2.5-Pro | 0.2732 | 0.3333 | 0.2500 | 0.2857 | 0.3333 | 0.2500 | 0.2857 |
| Gpt-4.1-mini | 0.2065 | 0.0000 | 0.0000 | 0.0000 | 0.0000 | 0.0000 | 0.0000 |
| Gpt-4.1 | 0.2401 | 0.0000 | 0.0000 | 0.0000 | 0.0000 | 0.0000 | 0.0000 |
| Claude-3.5-Haiku | 0.4110 | 0.0000 | 0.0000 | 0.0000 | 0.0000 | 0.0000 | 0.0000 |
| Claude-3.7-Sonnet | 0.3013 | 0.0000 | 0.0000 | 0.0000 | 0.0000 | 0.0000 | 0.0000 |

Table S29: Detailed results for benchmark BIOMD0000000258 across all models.

| Model | STE ↓ | RMS (with modifiers) ↑ | | | RMS (without modifiers) ↑ | | |
|---|---|---|---|---|---|---|---|
| | | Precision | Recall | F1 | Precision | Recall | F1 |
| Gemini-2.5-Flash | 0.1344 | 0.3333 | 0.2500 | 0.2857 | 0.6667 | 0.5000 | 0.5714 |
| Gemini-2.5-Pro | 0.2931 | 0.0000 | 0.0000 | 0.0000 | 0.0000 | 0.0000 | 0.0000 |
| Gpt-4.1-mini | 0.4833 | 0.3333 | 0.2500 | 0.2857 | 0.3333 | 0.2500 | 0.2857 |
| Gpt-4.1 | 0.3291 | 0.5000 | 0.2500 | 0.3333 | 1.0000 | 0.5000 | 0.6667 |
| Claude-3.5-Haiku | 0.5966 | 0.0000 | 0.0000 | 0.0000 | 0.0000 | 0.0000 | 0.0000 |
| Claude-3.7-Sonnet | 0.0885 | 0.3333 | 0.2500 | 0.2857 | 1.0000 | 0.7500 | 0.8571 |

Table S30: Detailed results for benchmark BIOMD0000000282 across all models.

| Model | STE ↓ | RMS (with modifiers) ↑ | | | RMS (without modifiers) ↑ | | |
|---|---|---|---|---|---|---|---|
| | | Precision | Recall | F1 | Precision | Recall | F1 |
| Gemini-2.5-Flash | 0.0918 | 0.0000 | 0.0000 | 0.0000 | 0.0000 | 0.0000 | 0.0000 |
| Gemini-2.5-Pro | 0.1365 | 0.0000 | 0.0000 | 0.0000 | 0.0000 | 0.0000 | 0.0000 |
| Gpt-4.1-mini | 0.3644 | 0.0000 | 0.0000 | 0.0000 | 0.0000 | 0.0000 | 0.0000 |
| Gpt-4.1 | 0.3056 | 0.5000 | 0.3333 | 0.4000 | 0.5000 | 0.3333 | 0.4000 |
| Claude-3.5-Haiku | 0.3980 | 0.0000 | 0.0000 | 0.0000 | 0.0000 | 0.0000 | 0.0000 |
| Claude-3.7-Sonnet | 0.2512 | 0.0000 | 0.0000 | 0.0000 | 0.0000 | 0.0000 | 0.0000 |

Table S31: Detailed results for benchmark BIOMD0000000283 across all models.

| Model | STE ↓ | RMS (with modifiers) ↑ | | | RMS (without modifiers) ↑ | | |
|---|---|---|---|---|---|---|---|
| | | Precision | Recall | F1 | Precision | Recall | F1 |
| Gemini-2.5-Flash | 0.8293 | 0.5000 | 0.5000 | 0.5000 | 0.5000 | 0.5000 | 0.5000 |
| Gemini-2.5-Pro | 0.2757 | 0.0000 | 0.0000 | 0.0000 | 0.0000 | 0.0000 | 0.0000 |
| Gpt-4.1-mini | 0.5541 | 0.0000 | 0.0000 | 0.0000 | 0.0000 | 0.0000 | 0.0000 |
| Gpt-4.1 | 0.5206 | 0.0000 | 0.0000 | 0.0000 | 0.0000 | 0.0000 | 0.0000 |
| Claude-3.5-Haiku | 0.5216 | 0.0000 | 0.0000 | 0.0000 | 0.0000 | 0.0000 | 0.0000 |
| Claude-3.7-Sonnet | 0.2165 | 0.0000 | 0.0000 | 0.0000 | 0.0000 | 0.0000 | 0.0000 |

Table S32: Detailed results for benchmark BIOMD0000000319 across all models.

| Model | STE ↓ | RMS (with modifiers) ↑ | | | RMS (without modifiers) ↑ | | |
|---|---|---|---|---|---|---|---|
| | | Precision | Recall | F1 | Precision | Recall | F1 |
| Gemini-2.5-Flash | 0.2432 | 0.0000 | 0.0000 | 0.0000 | 0.0000 | 0.0000 | 0.0000 |
| Gemini-2.5-Pro | 0.0467 | 0.5000 | 0.5000 | 0.5000 | 0.5000 | 0.5000 | 0.5000 |
| Gpt-4.1-mini | 0.5681 | 0.3750 | 0.7500 | 0.5000 | 0.3750 | 0.7500 | 0.5000 |
| Gpt-4.1 | 0.3117 | 0.6000 | 0.7500 | 0.6667 | 0.6000 | 0.7500 | 0.6667 |
| Claude-3.5-Haiku | 0.3284 | 0.5000 | 0.2500 | 0.3333 | 0.5000 | 0.2500 | 0.3333 |
| Claude-3.7-Sonnet | 0.0851 | 0.2500 | 0.2500 | 0.2500 | 0.2500 | 0.2500 | 0.2500 |

Table S33: Detailed results for benchmark BIOMD0000000322 across all models.

| Model | STE ↓ | RMS (with modifiers) ↑ | | | RMS (without modifiers) ↑ | | |
| | | Precision | Recall | F1 | Precision | Recall | F1 |
|---|---|---|---|---|---|---|---|
| Gemini-2.5-Flash | 0.5064 | 0.0000 | 0.0000 | 0.0000 | 0.0000 | 0.0000 | 0.0000 |
| Gemini-2.5-Pro | 0.5171 | 0.3333 | 0.2222 | 0.2667 | 0.3333 | 0.2222 | 0.2667 |
| Gpt-4.1-mini | 0.4708 | 0.1429 | 0.1111 | 0.1250 | 0.1429 | 0.1111 | 0.1250 |
| Gpt-4.1 | 0.4354 | 0.0000 | 0.0000 | 0.0000 | 0.0000 | 0.0000 | 0.0000 |
| Claude-3.5-Haiku | 0.8012 | 0.3333 | 0.1111 | 0.1667 | 0.3333 | 0.1111 | 0.1667 |
| Claude-3.7-Sonnet | 0.5632 | 0.0000 | 0.0000 | 0.0000 | 0.0000 | 0.0000 | 0.0000 |

Table S34: Detailed results for benchmark BIOMD0000000323 across all models.

| Model | STE ↓ | RMS (with modifiers) ↑ | | | RMS (without modifiers) ↑ | | |
| | | Precision | Recall | F1 | Precision | Recall | F1 |
|---|---|---|---|---|---|---|---|
| Gemini-2.5-Flash | 0.2559 | 0.0000 | 0.0000 | 0.0000 | 0.0000 | 0.0000 | 0.0000 |
| Gemini-2.5-Pro | 0.2966 | 0.5000 | 0.5000 | 0.5000 | 0.5000 | 0.5000 | 0.5000 |
| Gpt-4.1-mini | 0.5472 | 0.0000 | 0.0000 | 0.0000 | 0.0000 | 0.0000 | 0.0000 |
| Gpt-4.1 | 0.5136 | 0.5000 | 0.5000 | 0.5000 | 0.5000 | 0.5000 | 0.5000 |
| Claude-3.5-Haiku | 0.7152 | 0.0000 | 0.0000 | 0.0000 | 0.0000 | 0.0000 | 0.0000 |
| Claude-3.7-Sonnet | 0.1352 | 0.5000 | 0.5000 | 0.5000 | 0.5000 | 0.5000 | 0.5000 |

Table S35: Detailed results for benchmark BIOMD0000000329 across all models.

| Model | STE ↓ | RMS (with modifiers) ↑ | | | RMS (without modifiers) ↑ | | |
| | | Precision | Recall | F1 | Precision | Recall | F1 |
|---|---|---|---|---|---|---|---|
| Gemini-2.5-Flash | 0.3470 | 0.5000 | 0.2500 | 0.3333 | 0.5000 | 0.3333 | 0.4000 |
| Gemini-2.5-Pro | 0.8958 | 0.1667 | 0.1250 | 0.1429 | 0.6667 | 0.6667 | 0.6667 |
| Gpt-4.1-mini | 0.7202 | 0.2000 | 0.1250 | 0.1538 | 0.6000 | 0.5000 | 0.5455 |
| Gpt-4.1 | 0.7084 | 0.4286 | 0.3750 | 0.4000 | 0.5714 | 0.6667 | 0.6154 |
| Claude-3.5-Haiku | 0.9347 | 0.0000 | 0.0000 | 0.0000 | 0.0000 | 0.0000 | 0.0000 |
| Claude-3.7-Sonnet | 0.5742 | 0.5714 | 0.5000 | 0.5333 | 1.0000 | 1.0000 | 1.0000 |

Table S36: Detailed results for benchmark BIOMD0000000357 across all models.

| Model | STE ↓ | RMS (with modifiers) ↑ | | | RMS (without modifiers) ↑ | | |
| | | Precision | Recall | F1 | Precision | Recall | F1 |
|---|---|---|---|---|---|---|---|
| Gemini-2.5-Flash | 0.8159 | 0.0000 | 0.0000 | 0.0000 | 0.0000 | 0.0000 | 0.0000 |
| Gemini-2.5-Pro | 0.4504 | 0.1250 | 0.1250 | 0.1250 | 0.1250 | 0.1250 | 0.1250 |
| Gpt-4.1-mini | 0.7683 | 0.0000 | 0.0000 | 0.0000 | 0.0000 | 0.0000 | 0.0000 |
| Gpt-4.1 | 0.7003 | 0.0000 | 0.0000 | 0.0000 | 0.0000 | 0.0000 | 0.0000 |
| Claude-3.5-Haiku | 0.8223 | 0.0000 | 0.0000 | 0.0000 | 0.0000 | 0.0000 | 0.0000 |
| Claude-3.7-Sonnet | 0.6243 | 0.0000 | 0.0000 | 0.0000 | 0.0000 | 0.0000 | 0.0000 |

Table S37: Detailed results for benchmark BIOMD0000000359 across all models.

| Model | STE ↓ | RMS (with modifiers) ↑ | | | RMS (without modifiers) ↑ | | |
|---|---|---|---|---|---|---|---|
| | | Precision | Recall | F1 | Precision | Recall | F1 |
| Gemini-2.5-Flash | 0.8930 | 0.1250 | 0.1250 | 0.1250 | 0.1250 | 0.1250 | 0.1250 |
| Gemini-2.5-Pro | 0.4009 | 0.1429 | 0.1250 | 0.1333 | 0.1429 | 0.1250 | 0.1333 |
| Gpt-4.1-mini | 0.7740 | 0.0000 | 0.0000 | 0.0000 | 0.0000 | 0.0000 | 0.0000 |
| Gpt-4.1 | 0.8763 | 0.0000 | 0.0000 | 0.0000 | 0.0000 | 0.0000 | 0.0000 |
| Claude-3.5-Haiku | 0.7591 | 0.0000 | 0.0000 | 0.0000 | 0.0000 | 0.0000 | 0.0000 |
| Claude-3.7-Sonnet | 0.7117 | 0.1250 | 0.1250 | 0.1250 | 0.1250 | 0.1250 | 0.1250 |

Table S38: Detailed results for benchmark BIOMD0000000360 across all models.

| Model | STE ↓ | RMS (with modifiers) ↑ | | | RMS (without modifiers) ↑ | | |
|---|---|---|---|---|---|---|---|
| | | Precision | Recall | F1 | Precision | Recall | F1 |
| Gemini-2.5-Flash | 0.6096 | 0.0000 | 0.0000 | 0.0000 | 0.0000 | 0.0000 | 0.0000 |
| Gemini-2.5-Pro | 0.5411 | 0.0000 | 0.0000 | 0.0000 | 0.0000 | 0.0000 | 0.0000 |
| Gpt-4.1-mini | 0.8952 | 0.0000 | 0.0000 | 0.0000 | 0.0000 | 0.0000 | 0.0000 |
| Gpt-4.1 | 0.7970 | 0.0000 | 0.0000 | 0.0000 | 0.0000 | 0.0000 | 0.0000 |
| Claude-3.5-Haiku | 0.8390 | 0.0000 | 0.0000 | 0.0000 | 0.0000 | 0.0000 | 0.0000 |
| Claude-3.7-Sonnet | 0.5745 | 0.0000 | 0.0000 | 0.0000 | 0.0000 | 0.0000 | 0.0000 |

Table S39: Detailed results for benchmark BIOMD0000000361 across all models.

| Model | STE ↓ | RMS (with modifiers) ↑ | | | RMS (without modifiers) ↑ | | |
|---|---|---|---|---|---|---|---|
| | | Precision | Recall | F1 | Precision | Recall | F1 |
| Gemini-2.5-Flash | 0.5061 | 0.1667 | 0.2000 | 0.1818 | 0.1667 | 0.2000 | 0.1818 |
| Gemini-2.5-Pro | 0.4931 | 0.1667 | 0.2000 | 0.1818 | 0.1667 | 0.2000 | 0.1818 |
| Gpt-4.1-mini | 0.9735 | 0.0556 | 0.2000 | 0.0870 | 0.0556 | 0.2000 | 0.0870 |
| Gpt-4.1 | 0.5194 | 0.2500 | 0.2000 | 0.2222 | 0.2500 | 0.2000 | 0.2222 |
| Claude-3.5-Haiku | 0.9661 | 0.0000 | 0.0000 | 0.0000 | 0.0000 | 0.0000 | 0.0000 |
| Claude-3.7-Sonnet | 0.5663 | 0.2500 | 0.4000 | 0.3077 | 0.2500 | 0.4000 | 0.3077 |

Table S40: Detailed results for benchmark BIOMD0000000363 across all models.

| Model | STE ↓ | RMS (with modifiers) ↑ | | | RMS (without modifiers) ↑ | | |
|---|---|---|---|---|---|---|---|
| | | Precision | Recall | F1 | Precision | Recall | F1 |
| Gemini-2.5-Flash | 0.2280 | 0.5000 | 0.5000 | 0.5000 | 0.5000 | 0.5000 | 0.5000 |
| Gemini-2.5-Pro | 0.0018 | 1.0000 | 0.7500 | 0.8571 | 1.0000 | 0.7500 | 0.8571 |
| Gpt-4.1-mini | 0.7069 | 0.0000 | 0.0000 | 0.0000 | 0.0000 | 0.0000 | 0.0000 |
| Gpt-4.1 | 0.1685 | 1.0000 | 0.7500 | 0.8571 | 1.0000 | 0.7500 | 0.8571 |
| Claude-3.5-Haiku | 0.8193 | 0.3333 | 0.2500 | 0.2857 | 0.3333 | 0.2500 | 0.2857 |
| Claude-3.7-Sonnet | 0.0678 | 0.7500 | 0.7500 | 0.7500 | 0.7500 | 0.7500 | 0.7500 |

Table S41: Detailed results for benchmark BIOMD0000000405 across all models.

| Model | STE ↓ | RMS (with modifiers) ↑ | | | RMS (without modifiers) ↑ | | |
|---|---|---|---|---|---|---|---|
| | | Precision | Recall | F1 | Precision | Recall | F1 |
| Gemini-2.5-Flash | 0.0563 | 0.0000 | 0.0000 | 0.0000 | 0.0000 | 0.0000 | 0.0000 |
| Gemini-2.5-Pro | 1.0000 | 0.0000 | 0.0000 | 0.0000 | 0.0000 | 0.0000 | 0.0000 |
| Gpt-4.1-mini | 0.4653 | 0.0000 | 0.0000 | 0.0000 | 0.0000 | 0.0000 | 0.0000 |
| Gpt-4.1 | 0.3312 | 0.0000 | 0.0000 | 0.0000 | 0.0000 | 0.0000 | 0.0000 |
| Claude-3.5-Haiku | 0.6334 | 0.0000 | 0.0000 | 0.0000 | 0.0000 | 0.0000 | 0.0000 |
| Claude-3.7-Sonnet | 0.4975 | 0.0000 | 0.0000 | 0.0000 | 0.0000 | 0.0000 | 0.0000 |

Table S42: Detailed results for benchmark BIOMD0000000413 across all models.

| Model | STE ↓ | RMS (with modifiers) ↑ | | | RMS (without modifiers) ↑ | | |
|---|---|---|---|---|---|---|---|
| | | Precision | Recall | F1 | Precision | Recall | F1 |
| Gemini-2.5-Flash | 0.1108 | 0.5000 | 0.2222 | 0.3077 | 0.5000 | 0.2222 | 0.3077 |
| Gemini-2.5-Pro | 0.0726 | 0.2000 | 0.1111 | 0.1429 | 0.2000 | 0.1111 | 0.1429 |
| Gpt-4.1-mini | 0.4716 | 0.0000 | 0.0000 | 0.0000 | 0.0000 | 0.0000 | 0.0000 |
| Gpt-4.1 | 0.0924 | 0.8000 | 0.4444 | 0.5714 | 0.8000 | 0.4444 | 0.5714 |
| Claude-3.5-Haiku | 0.5709 | 0.0000 | 0.0000 | 0.0000 | 0.0000 | 0.0000 | 0.0000 |
| Claude-3.7-Sonnet | 0.0824 | 0.3333 | 0.2222 | 0.2667 | 0.4000 | 0.2222 | 0.2857 |

Table S43: Detailed results for benchmark BIOMD0000000435 across all models.

| Model | STE ↓ | RMS (with modifiers) ↑ | | | RMS (without modifiers) ↑ | | |
|---|---|---|---|---|---|---|---|
| | | Precision | Recall | F1 | Precision | Recall | F1 |
| Gemini-2.5-Flash | 0.3934 | 0.0000 | 0.0000 | 0.0000 | 0.6667 | 0.2500 | 0.3636 |
| Gemini-2.5-Pro | 0.5931 | 0.0000 | 0.0000 | 0.0000 | 0.0000 | 0.0000 | 0.0000 |
| Gpt-4.1-mini | 0.9993 | 0.0000 | 0.0000 | 0.0000 | 0.0000 | 0.0000 | 0.0000 |
| Gpt-4.1 | 0.7326 | 0.0000 | 0.0000 | 0.0000 | 0.0000 | 0.0000 | 0.0000 |
| Claude-3.5-Haiku | 0.4821 | 0.0000 | 0.0000 | 0.0000 | 0.0000 | 0.0000 | 0.0000 |
| Claude-3.7-Sonnet | 0.6931 | 0.0000 | 0.0000 | 0.0000 | 1.0000 | 1.0000 | 1.0000 |

Table S44: Detailed results for benchmark BIOMD0000000438 across all models.

| Model | STE ↓ | RMS (with modifiers) ↑ | | | RMS (without modifiers) ↑ | | |
|---|---|---|---|---|---|---|---|
| | | Precision | Recall | F1 | Precision | Recall | F1 |
| Gemini-2.5-Flash | 0.6830 | 0.0000 | 0.0000 | 0.0000 | 0.0000 | 0.0000 | 0.0000 |
| Gemini-2.5-Pro | 0.4543 | 0.0000 | 0.0000 | 0.0000 | 0.2000 | 0.1429 | 0.1667 |
| Gpt-4.1-mini | 0.8708 | 0.0000 | 0.0000 | 0.0000 | 0.0000 | 0.0000 | 0.0000 |
| Gpt-4.1 | 0.6301 | 0.0000 | 0.0000 | 0.0000 | 0.0000 | 0.0000 | 0.0000 |
| Claude-3.5-Haiku | 0.8708 | 0.0000 | 0.0000 | 0.0000 | 0.0000 | 0.0000 | 0.0000 |
| Claude-3.7-Sonnet | 0.7962 | 0.0000 | 0.0000 | 0.0000 | 0.0000 | 0.0000 | 0.0000 |

Table S45: Detailed results for benchmark BIOMD0000000454 across all models.

| Model | STE ↓ | RMS (with modifiers) ↑ | | | RMS (without modifiers) ↑ | | |
|---|---|---|---|---|---|---|---|
| | | Precision | Recall | F1 | Precision | Recall | F1 |
| Gemini-2.5-Flash | 0.0161 | 0.0000 | 0.0000 | 0.0000 | 0.3333 | 0.2500 | 0.2857 |
| Gemini-2.5-Pro | 0.0020 | 0.0000 | 0.0000 | 0.0000 | 0.2500 | 0.2500 | 0.2500 |
| Gpt-4.1-mini | 0.2175 | 0.0000 | 0.0000 | 0.0000 | 0.0000 | 0.0000 | 0.0000 |
| Gpt-4.1 | 0.3276 | 0.0000 | 0.0000 | 0.0000 | 0.0000 | 0.0000 | 0.0000 |
| Claude-3.5-Haiku | 0.1641 | 0.0000 | 0.0000 | 0.0000 | 0.0000 | 0.0000 | 0.0000 |
| Claude-3.7-Sonnet | 0.0190 | 0.0000 | 0.0000 | 0.0000 | 0.0000 | 0.0000 | 0.0000 |

Table S46: Detailed results for benchmark BIOMD0000000455 across all models.

| Model | STE ↓ | RMS (with modifiers) ↑ | | | RMS (without modifiers) ↑ | | |
|---|---|---|---|---|---|---|---|
| | | Precision | Recall | F1 | Precision | Recall | F1 |
| Gemini-2.5-Flash | 0.0056 | 0.0000 | 0.0000 | 0.0000 | 0.0000 | 0.0000 | 0.0000 |
| Gemini-2.5-Pro | 0.0005 | 0.0000 | 0.0000 | 0.0000 | 0.0000 | 0.0000 | 0.0000 |
| Gpt-4.1-mini | 0.0000 | 0.0000 | 0.0000 | 0.0000 | 0.0000 | 0.0000 | 0.0000 |
| Gpt-4.1 | 0.1924 | 0.0000 | 0.0000 | 0.0000 | 0.0000 | 0.0000 | 0.0000 |
| Claude-3.5-Haiku | 0.1399 | 0.0000 | 0.0000 | 0.0000 | 0.0000 | 0.0000 | 0.0000 |
| Claude-3.7-Sonnet | 0.0004 | 0.0000 | 0.0000 | 0.0000 | 0.0000 | 0.0000 | 0.0000 |

Table S47: Detailed results for benchmark BIOMD0000000456 across all models.

| Model | STE ↓ | RMS (with modifiers) ↑ | | | RMS (without modifiers) ↑ | | |
|---|---|---|---|---|---|---|---|
| | | Precision | Recall | F1 | Precision | Recall | F1 |
| Gemini-2.5-Flash | 0.0001 | 0.0000 | 0.0000 | 0.0000 | 0.1429 | 0.1667 | 0.1538 |
| Gemini-2.5-Pro | 0.0033 | 0.0000 | 0.0000 | 0.0000 | 0.0000 | 0.0000 | 0.0000 |
| Gpt-4.1-mini | 0.1217 | 0.0000 | 0.0000 | 0.0000 | 0.0000 | 0.0000 | 0.0000 |
| Gpt-4.1 | 0.1114 | 0.0000 | 0.0000 | 0.0000 | 0.0000 | 0.0000 | 0.0000 |
| Claude-3.5-Haiku | 0.0934 | 0.0000 | 0.0000 | 0.0000 | 0.0000 | 0.0000 | 0.0000 |
| Claude-3.7-Sonnet | 0.0013 | 0.0000 | 0.0000 | 0.0000 | 0.0000 | 0.0000 | 0.0000 |

Table S48: Detailed results for benchmark BIOMD0000000458 across all models.

| Model | STE ↓ | RMS (with modifiers) ↑ | | | RMS (without modifiers) ↑ | | |
|---|---|---|---|---|---|---|---|
| | | Precision | Recall | F1 | Precision | Recall | F1 |
| Gemini-2.5-Flash | 0.1334 | 0.0000 | 0.0000 | 0.0000 | 0.0000 | 0.0000 | 0.0000 |
| Gemini-2.5-Pro | 0.1455 | 0.0000 | 0.0000 | 0.0000 | 0.5000 | 0.3333 | 0.4000 |
| Gpt-4.1-mini | 0.2124 | 0.0000 | 0.0000 | 0.0000 | 0.0000 | 0.0000 | 0.0000 |
| Gpt-4.1 | 0.2314 | 0.0000 | 0.0000 | 0.0000 | 0.0000 | 0.0000 | 0.0000 |
| Claude-3.5-Haiku | 0.2269 | 0.0000 | 0.0000 | 0.0000 | 0.5000 | 0.3333 | 0.4000 |
| Claude-3.7-Sonnet | 0.2100 | 0.0000 | 0.0000 | 0.0000 | 0.0000 | 0.0000 | 0.0000 |

Table S49: Detailed results for benchmark BIOMD0000000459 across all models.

| Model | STE ↓ | RMS (with modifiers) ↑ | | | RMS (without modifiers) ↑ | | |
|---|---|---|---|---|---|---|---|
| | | Precision | Recall | F1 | Precision | Recall | F1 |
| Gemini-2.5-Flash | 0.1024 | 0.0000 | 0.0000 | 0.0000 | 0.2500 | 0.3333 | 0.2857 |
| Gemini-2.5-Pro | 0.0935 | 0.0000 | 0.0000 | 0.0000 | 0.3333 | 0.3333 | 0.3333 |
| Gpt-4.1-mini | 0.6115 | 0.0000 | 0.0000 | 0.0000 | 0.6667 | 0.6667 | 0.6667 |
| Gpt-4.1 | 0.7493 | 0.0000 | 0.0000 | 0.0000 | 0.3333 | 0.3333 | 0.3333 |
| Claude-3.5-Haiku | 0.6208 | 0.0000 | 0.0000 | 0.0000 | 0.6667 | 0.6667 | 0.6667 |
| Claude-3.7-Sonnet | 0.1001 | 0.0000 | 0.0000 | 0.0000 | 0.3750 | 1.0000 | 0.5455 |

Table S50: Detailed results for benchmark BIOMD0000000460 across all models.

| Model | STE ↓ | RMS (with modifiers) ↑ | | | RMS (without modifiers) ↑ | | |
|---|---|---|---|---|---|---|---|
| | | Precision | Recall | F1 | Precision | Recall | F1 |
| Gemini-2.5-Flash | 0.4210 | 0.0000 | 0.0000 | 0.0000 | 0.1667 | 0.3333 | 0.2222 |
| Gemini-2.5-Pro | 0.1022 | 0.0000 | 0.0000 | 0.0000 | 0.6000 | 1.0000 | 0.7500 |
| Gpt-4.1-mini | 0.4094 | 0.0000 | 0.0000 | 0.0000 | 0.6000 | 1.0000 | 0.7500 |
| Gpt-4.1 | 0.5281 | 0.0000 | 0.0000 | 0.0000 | 0.1667 | 0.3333 | 0.2222 |
| Claude-3.5-Haiku | 0.4153 | 0.0000 | 0.0000 | 0.0000 | 0.6667 | 0.6667 | 0.6667 |
| Claude-3.7-Sonnet | 0.2646 | 0.0000 | 0.0000 | 0.0000 | 0.6000 | 1.0000 | 0.7500 |

Table S51: Detailed results for benchmark BIOMD0000000461 across all models.

| Model | STE ↓ | RMS (with modifiers) ↑ | | | RMS (without modifiers) ↑ | | |
|---|---|---|---|---|---|---|---|
| | | Precision | Recall | F1 | Precision | Recall | F1 |
| Gemini-2.5-Flash | 0.0070 | 0.0000 | 0.0000 | 0.0000 | 0.1667 | 0.3333 | 0.2222 |
| Gemini-2.5-Pro | 0.0093 | 0.0000 | 0.0000 | 0.0000 | 0.0000 | 0.0000 | 0.0000 |
| Gpt-4.1-mini | 0.6213 | 0.0000 | 0.0000 | 0.0000 | 0.1429 | 0.3333 | 0.2000 |
| Gpt-4.1 | 0.4508 | 0.0000 | 0.0000 | 0.0000 | 0.0000 | 0.0000 | 0.0000 |
| Claude-3.5-Haiku | 0.7499 | 0.0000 | 0.0000 | 0.0000 | 0.0000 | 0.0000 | 0.0000 |
| Claude-3.7-Sonnet | 0.1315 | 0.0000 | 0.0000 | 0.0000 | 0.2500 | 0.6667 | 0.3636 |

Table S52: Detailed results for benchmark BIOMD0000000462 across all models.

| Model | STE ↓ | RMS (with modifiers) ↑ | | | RMS (without modifiers) ↑ | | |
|---|---|---|---|---|---|---|---|
| | | Precision | Recall | F1 | Precision | Recall | F1 |
| Gemini-2.5-Flash | 0.2277 | 0.0000 | 0.0000 | 0.0000 | 0.0000 | 0.0000 | 0.0000 |
| Gemini-2.5-Pro | 0.2251 | 0.0000 | 0.0000 | 0.0000 | 0.6667 | 0.2500 | 0.3636 |
| Gpt-4.1-mini | 0.5000 | 0.0000 | 0.0000 | 0.0000 | 0.0000 | 0.0000 | 0.0000 |
| Gpt-4.1 | 1.0000 | 0.0000 | 0.0000 | 0.0000 | 0.5000 | 0.2500 | 0.3333 |
| Claude-3.5-Haiku | 0.5000 | 0.0000 | 0.0000 | 0.0000 | 0.0000 | 0.0000 | 0.0000 |
| Claude-3.7-Sonnet | 0.0795 | 0.0000 | 0.0000 | 0.0000 | 0.0000 | 0.0000 | 0.0000 |

Table S53: Detailed results for benchmark BIOMD0000000483 across all models.

| Model | STE ↓ | RMS (with modifiers) ↑ | | | RMS (without modifiers) ↑ | | |
|---|---|---|---|---|---|---|---|
| | | Precision | Recall | F1 | Precision | Recall | F1 |
| Gemini-2.5-Flash | 0.0000 | 0.0000 | 0.0000 | 0.0000 | 0.0000 | 0.0000 | 0.0000 |
| Gemini-2.5-Pro | 0.0000 | 0.0000 | 0.0000 | 0.0000 | 0.3333 | 0.2500 | 0.2857 |
| Gpt-4.1-mini | 0.0000 | 0.0000 | 0.0000 | 0.0000 | 0.3333 | 0.2500 | 0.2857 |
| Gpt-4.1 | 0.0000 | 0.0000 | 0.0000 | 0.0000 | 0.5000 | 0.5000 | 0.5000 |
| Claude-3.5-Haiku | 0.0000 | 0.0000 | 0.0000 | 0.0000 | 0.3333 | 0.1250 | 0.1818 |
| Claude-3.7-Sonnet | 0.0000 | 0.0000 | 0.0000 | 0.0000 | 0.2500 | 0.2500 | 0.2500 |

Table S54: Detailed results for benchmark BIOMD0000000484 across all models.

| Model | STE ↓ | RMS (with modifiers) ↑ | | | RMS (without modifiers) ↑ | | |
|---|---|---|---|---|---|---|---|
| | | Precision | Recall | F1 | Precision | Recall | F1 |
| Gemini-2.5-Flash | 0.0118 | 1.0000 | 0.5000 | 0.6667 | 1.0000 | 0.5000 | 0.6667 |
| Gemini-2.5-Pro | 0.9999 | 1.0000 | 0.5000 | 0.6667 | 1.0000 | 0.5000 | 0.6667 |
| Gpt-4.1-mini | 0.9999 | 0.5000 | 0.5000 | 0.5000 | 1.0000 | 1.0000 | 1.0000 |
| Gpt-4.1 | 0.0000 | 0.0000 | 0.0000 | 0.0000 | 0.0000 | 0.0000 | 0.0000 |
| Claude-3.5-Haiku | 0.9999 | 1.0000 | 0.5000 | 0.6667 | 1.0000 | 0.5000 | 0.6667 |
| Claude-3.7-Sonnet | 0.0000 | 1.0000 | 0.5000 | 0.6667 | 1.0000 | 0.5000 | 0.6667 |

Table S55: Detailed results for benchmark BIOMD0000000485 across all models.

| Model | STE ↓ | RMS (with modifiers) ↑ | | | RMS (without modifiers) ↑ | | |
|---|---|---|---|---|---|---|---|
| | | Precision | Recall | F1 | Precision | Recall | F1 |
| Gemini-2.5-Flash | 0.0034 | 1.0000 | 0.3333 | 0.5000 | 1.0000 | 0.5000 | 0.6667 |
| Gemini-2.5-Pro | 0.0023 | 1.0000 | 0.3333 | 0.5000 | 1.0000 | 0.5000 | 0.6667 |
| Gpt-4.1-mini | 0.7563 | 1.0000 | 0.3333 | 0.5000 | 1.0000 | 0.5000 | 0.6667 |
| Gpt-4.1 | 0.0029 | 1.0000 | 0.3333 | 0.5000 | 1.0000 | 0.5000 | 0.6667 |
| Claude-3.5-Haiku | 0.7004 | 1.0000 | 0.3333 | 0.5000 | 1.0000 | 0.5000 | 0.6667 |
| Claude-3.7-Sonnet | 0.0028 | 0.5000 | 0.3333 | 0.4000 | 0.5000 | 0.5000 | 0.5000 |

Table S56: Detailed results for benchmark BIOMD0000000486 across all models.

| Model | STE ↓ | RMS (with modifiers) ↑ | | | RMS (without modifiers) ↑ | | |
|---|---|---|---|---|---|---|---|
| | | Precision | Recall | F1 | Precision | Recall | F1 |
| Gemini-2.5-Flash | 0.0000 | 0.0000 | 0.0000 | 0.0000 | 1.0000 | 0.5000 | 0.6667 |
| Gemini-2.5-Pro | 0.0000 | 0.0000 | 0.0000 | 0.0000 | 1.0000 | 1.0000 | 1.0000 |
| Gpt-4.1-mini | 0.0000 | 0.0000 | 0.0000 | 0.0000 | 1.0000 | 0.5000 | 0.6667 |
| Gpt-4.1 | 0.0000 | 0.0000 | 0.0000 | 0.0000 | 1.0000 | 0.5000 | 0.6667 |
| Claude-3.5-Haiku | 0.9995 | 0.0000 | 0.0000 | 0.0000 | 1.0000 | 0.5000 | 0.6667 |
| Claude-3.7-Sonnet | 0.0000 | 0.0000 | 0.0000 | 0.0000 | 1.0000 | 0.5000 | 0.6667 |

Table S57: Detailed results for benchmark BIOMD0000000487 across all models.

| Model | STE ↓ | RMS (with modifiers) ↑ | | | RMS (without modifiers) ↑ | | |
|---|---|---|---|---|---|---|---|
| | | Precision | Recall | F1 | Precision | Recall | F1 |
| Gemini-2.5-Flash | 0.0000 | 0.0000 | 0.0000 | 0.0000 | 0.0000 | 0.0000 | 0.0000 |
| Gemini-2.5-Pro | 0.0000 | 0.0000 | 0.0000 | 0.0000 | 0.8000 | 0.6667 | 0.7273 |
| Gpt-4.1-mini | 0.0000 | 0.0000 | 0.0000 | 0.0000 | 0.0000 | 0.0000 | 0.0000 |
| Gpt-4.1 | 0.0000 | 0.0000 | 0.0000 | 0.0000 | 0.0000 | 0.0000 | 0.0000 |
| Claude-3.5-Haiku | 1.0000 | 0.0000 | 0.0000 | 0.0000 | 0.0000 | 0.0000 | 0.0000 |
| Claude-3.7-Sonnet | 0.0000 | 0.0000 | 0.0000 | 0.0000 | 0.0000 | 0.0000 | 0.0000 |

Table S58: Detailed results for benchmark BIOMD0000000573 across all models.

| Model | STE ↓ | RMS (with modifiers) ↑ | | | RMS (without modifiers) ↑ | | |
|---|---|---|---|---|---|---|---|
| | | Precision | Recall | F1 | Precision | Recall | F1 |
| Gemini-2.5-Flash | 0.8462 | 0.5000 | 0.1429 | 0.2222 | 1.0000 | 0.2857 | 0.4444 |
| Gemini-2.5-Pro | 0.9305 | 0.0000 | 0.0000 | 0.0000 | 0.0000 | 0.0000 | 0.0000 |
| Gpt-4.1-mini | 0.9871 | 0.0000 | 0.0000 | 0.0000 | 0.0000 | 0.0000 | 0.0000 |
| Gpt-4.1 | 0.4936 | 0.5000 | 0.1429 | 0.2222 | 1.0000 | 0.2857 | 0.4444 |
| Claude-3.5-Haiku | 0.9726 | 0.0000 | 0.0000 | 0.0000 | 0.0000 | 0.0000 | 0.0000 |
| Claude-3.7-Sonnet | 1.0000 | 0.0000 | 0.0000 | 0.0000 | 0.5000 | 0.1429 | 0.2222 |

Table S59: Detailed results for benchmark BIOMD0000000591 across all models.

| Model | STE ↓ | RMS (with modifiers) ↑ | | | RMS (without modifiers) ↑ | | |
|---|---|---|---|---|---|---|---|
| | | Precision | Recall | F1 | Precision | Recall | F1 |
| Gemini-2.5-Flash | 0.4766 | 0.0000 | 0.0000 | 0.0000 | 0.0000 | 0.0000 | 0.0000 |
| Gemini-2.5-Pro | 0.5343 | 0.0000 | 0.0000 | 0.0000 | 0.1667 | 0.2222 | 0.1905 |
| Gpt-4.1-mini | 0.8804 | 0.0000 | 0.0000 | 0.0000 | 0.0000 | 0.0000 | 0.0000 |
| Gpt-4.1 | 0.6136 | 0.0000 | 0.0000 | 0.0000 | 0.1429 | 0.1111 | 0.1250 |
| Claude-3.5-Haiku | 0.8804 | 0.0000 | 0.0000 | 0.0000 | 0.0000 | 0.0000 | 0.0000 |
| Claude-3.7-Sonnet | 0.5046 | 0.0000 | 0.0000 | 0.0000 | 0.1250 | 0.2222 | 0.1600 |

Table S60: Detailed results for benchmark BIOMD0000000624 across all models.

| Model | STE ↓ | RMS (with modifiers) ↑ | | | RMS (without modifiers) ↑ | | |
|---|---|---|---|---|---|---|---|
| | | Precision | Recall | F1 | Precision | Recall | F1 |
| Gemini-2.5-Flash | 0.1591 | 0.0000 | 0.0000 | 0.0000 | 0.0000 | 0.0000 | 0.0000 |
| Gemini-2.5-Pro | 0.0152 | 0.0000 | 0.0000 | 0.0000 | 0.0000 | 0.0000 | 0.0000 |
| Gpt-4.1-mini | 0.6379 | 0.5000 | 0.6000 | 0.5455 | 0.5000 | 0.6000 | 0.5455 |
| Gpt-4.1 | 0.0894 | 0.2500 | 0.2000 | 0.2222 | 0.2500 | 0.2000 | 0.2222 |
| Claude-3.5-Haiku | 0.6102 | 0.2500 | 0.2000 | 0.2222 | 0.2500 | 0.2000 | 0.2222 |
| Claude-3.7-Sonnet | 0.0369 | 0.0000 | 0.0000 | 0.0000 | 0.0000 | 0.0000 | 0.0000 |

Table S61: Detailed results for benchmark BIOMD0000000626 across all models.

| Model | STE ↓ | RMS (with modifiers) ↑ | | | RMS (without modifiers) ↑ | | |
|---|---|---|---|---|---|---|---|
| | | Precision | Recall | F1 | Precision | Recall | F1 |
| Gemini-2.5-Flash | 0.2852 | 0.0000 | 0.0000 | 0.0000 | 0.0000 | 0.0000 | 0.0000 |
| Gemini-2.5-Pro | 0.3496 | 0.0000 | 0.0000 | 0.0000 | 0.0000 | 0.0000 | 0.0000 |
| Gpt-4.1-mini | 0.8130 | 0.0000 | 0.0000 | 0.0000 | 0.0000 | 0.0000 | 0.0000 |
| Gpt-4.1 | 0.2121 | 0.0000 | 0.0000 | 0.0000 | 0.4000 | 1.0000 | 0.5714 |
| Claude-3.5-Haiku | 0.8031 | 0.0000 | 0.0000 | 0.0000 | 0.0000 | 0.0000 | 0.0000 |
| Claude-3.7-Sonnet | 0.0752 | 0.0000 | 0.0000 | 0.0000 | 0.0000 | 0.0000 | 0.0000 |

Table S62: Detailed results for benchmark BIOMD0000000629 across all models.

| Model | STE ↓ | RMS (with modifiers) ↑ | | | RMS (without modifiers) ↑ | | |
|---|---|---|---|---|---|---|---|
| | | Precision | Recall | F1 | Precision | Recall | F1 |
| Gemini-2.5-Flash | 0.0680 | 0.0000 | 0.0000 | 0.0000 | 0.0000 | 0.0000 | 0.0000 |
| Gemini-2.5-Pro | 0.1034 | 0.0000 | 0.0000 | 0.0000 | 0.0000 | 0.0000 | 0.0000 |
| Gpt-4.1-mini | 0.6441 | 0.0000 | 0.0000 | 0.0000 | 0.0000 | 0.0000 | 0.0000 |
| Gpt-4.1 | 0.1589 | 0.0000 | 0.0000 | 0.0000 | 0.0000 | 0.0000 | 0.0000 |
| Claude-3.5-Haiku | 0.1727 | 0.0000 | 0.0000 | 0.0000 | 0.0000 | 0.0000 | 0.0000 |
| Claude-3.7-Sonnet | 0.1898 | 0.0000 | 0.0000 | 0.0000 | 0.5000 | 0.5000 | 0.5000 |

Table S63: Detailed results for benchmark BIOMD0000000657 across all models.

| Model | STE ↓ | RMS (with modifiers) ↑ | | | RMS (without modifiers) ↑ | | |
|---|---|---|---|---|---|---|---|
| | | Precision | Recall | F1 | Precision | Recall | F1 |
| Gemini-2.5-Flash | 0.5022 | 0.0000 | 0.0000 | 0.0000 | 0.0000 | 0.0000 | 0.0000 |
| Gemini-2.5-Pro | 0.2482 | 0.0000 | 0.0000 | 0.0000 | 0.1667 | 0.3333 | 0.2222 |
| Gpt-4.1-mini | 0.4208 | 0.0000 | 0.0000 | 0.0000 | 0.0000 | 0.0000 | 0.0000 |
| Gpt-4.1 | 0.9890 | 0.0000 | 0.0000 | 0.0000 | 0.0000 | 0.0000 | 0.0000 |
| Claude-3.5-Haiku | 0.9166 | 0.0000 | 0.0000 | 0.0000 | 0.0000 | 0.0000 | 0.0000 |
| Claude-3.7-Sonnet | 1.0000 | 0.0000 | 0.0000 | 0.0000 | 0.0833 | 0.3333 | 0.1333 |

Table S64: Detailed results for benchmark BIOMD0000000663 across all models.

| Model | STE ↓ | RMS (with modifiers) ↑ | | | RMS (without modifiers) ↑ | | |
|---|---|---|---|---|---|---|---|
| | | Precision | Recall | F1 | Precision | Recall | F1 |
| Gemini-2.5-Flash | 0.9007 | 0.2000 | 0.1111 | 0.1429 | 0.2000 | 0.1429 | 0.1667 |
| Gemini-2.5-Pro | 0.4541 | 0.0000 | 0.0000 | 0.0000 | 0.3333 | 0.1429 | 0.2000 |
| Gpt-4.1-mini | 0.8295 | 0.2000 | 0.1111 | 0.1429 | 0.4000 | 0.2857 | 0.3333 |
| Gpt-4.1 | 0.8195 | 0.0000 | 0.0000 | 0.0000 | 0.0000 | 0.0000 | 0.0000 |
| Claude-3.5-Haiku | 0.8609 | 0.0000 | 0.0000 | 0.0000 | 0.0000 | 0.0000 | 0.0000 |
| Claude-3.7-Sonnet | 0.6529 | 0.0000 | 0.0000 | 0.0000 | 0.0000 | 0.0000 | 0.0000 |

Table S65: Detailed results for benchmark BIOMD0000000713 across all models.

| Model | STE ↓ | RMS (with modifiers) ↑ | | | RMS (without modifiers) ↑ | | |
|---|---|---|---|---|---|---|---|
| | | Precision | Recall | F1 | Precision | Recall | F1 |
| Gemini-2.5-Flash | 0.6132 | 0.3333 | 0.1429 | 0.2000 | 0.3333 | 0.1429 | 0.2000 |
| Gemini-2.5-Pro | 0.4825 | 0.0000 | 0.0000 | 0.0000 | 0.6667 | 0.2857 | 0.4000 |
| Gpt-4.1-mini | 0.5950 | 0.2500 | 0.2857 | 0.2667 | 0.5000 | 0.5714 | 0.5333 |
| Gpt-4.1 | 0.5572 | 0.4000 | 0.2857 | 0.3333 | 0.8000 | 0.5714 | 0.6667 |
| Claude-3.5-Haiku | 1.0000 | 0.0000 | 0.0000 | 0.0000 | 0.3333 | 0.1429 | 0.2000 |
| Claude-3.7-Sonnet | 0.7514 | 0.1429 | 0.1429 | 0.1429 | 0.1429 | 0.1429 | 0.1429 |

Table S66: Detailed results for benchmark BIOMD0000000728 across all models.

| Model | STE ↓ | RMS (with modifiers) ↑ | | | RMS (without modifiers) ↑ | | |
|---|---|---|---|---|---|---|---|
| | | Precision | Recall | F1 | Precision | Recall | F1 |
| Gemini-2.5-Flash | 0.5007 | 0.2500 | 0.2500 | 0.2500 | 0.7500 | 0.7500 | 0.7500 |
| Gemini-2.5-Pro | 0.3680 | 0.0000 | 0.0000 | 0.0000 | 0.3333 | 0.2500 | 0.2857 |
| Gpt-4.1-mini | 0.3541 | 0.0000 | 0.0000 | 0.0000 | 0.5000 | 0.5000 | 0.5000 |
| Gpt-4.1 | 0.7721 | 0.0000 | 0.0000 | 0.0000 | 0.3333 | 0.2500 | 0.2857 |
| Claude-3.5-Haiku | 0.5765 | 0.0000 | 0.0000 | 0.0000 | 0.0000 | 0.0000 | 0.0000 |
| Claude-3.7-Sonnet | 0.3408 | 0.0000 | 0.0000 | 0.0000 | 0.3333 | 0.2500 | 0.2857 |

Table S67: Detailed results for benchmark BIOMD0000000732 across all models.

| Model | STE ↓ | RMS (with modifiers) ↑ | | | RMS (without modifiers) ↑ | | |
|---|---|---|---|---|---|---|---|
| | | Precision | Recall | F1 | Precision | Recall | F1 |
| Gemini-2.5-Flash | 0.4081 | 0.2857 | 0.3333 | 0.3077 | 0.4286 | 0.5000 | 0.4615 |
| Gemini-2.5-Pro | 0.3624 | 0.0000 | 0.0000 | 0.0000 | 0.0000 | 0.0000 | 0.0000 |
| Gpt-4.1-mini | 0.5978 | 0.0000 | 0.0000 | 0.0000 | 0.0000 | 0.0000 | 0.0000 |
| Gpt-4.1 | 0.5924 | 0.0000 | 0.0000 | 0.0000 | 0.0000 | 0.0000 | 0.0000 |
| Claude-3.5-Haiku | 0.5962 | 0.0000 | 0.0000 | 0.0000 | 0.0000 | 0.0000 | 0.0000 |
| Claude-3.7-Sonnet | 0.4691 | 0.0000 | 0.0000 | 0.0000 | 0.0000 | 0.0000 | 0.0000 |

Table S68: Detailed results for benchmark BIOMD0000000733 across all models.

| Model | STE ↓ | RMS (with modifiers) ↑ | | | RMS (without modifiers) ↑ | | |
|---|---|---|---|---|---|---|---|
| | | Precision | Recall | F1 | Precision | Recall | F1 |
| Gemini-2.5-Flash | 0.1791 | 0.0000 | 0.0000 | 0.0000 | 0.3333 | 0.1667 | 0.2222 |
| Gemini-2.5-Pro | 0.0431 | 0.2857 | 0.3333 | 0.3077 | 1.0000 | 1.0000 | 1.0000 |
| Gpt-4.1-mini | 0.7708 | 0.0000 | 0.0000 | 0.0000 | 0.3333 | 0.1667 | 0.2222 |
| Gpt-4.1 | 0.7855 | 0.0000 | 0.0000 | 0.0000 | 0.6667 | 0.3333 | 0.4444 |
| Claude-3.5-Haiku | 0.7443 | 0.0000 | 0.0000 | 0.0000 | 0.0000 | 0.0000 | 0.0000 |
| Claude-3.7-Sonnet | 0.0946 | 0.0000 | 0.0000 | 0.0000 | 1.0000 | 0.5000 | 0.6667 |

Table S69: Detailed results for benchmark BIOMD0000000742 across all models.

| Model | STE ↓ | RMS (with modifiers) ↑ | | | RMS (without modifiers) ↑ | | |
|---|---|---|---|---|---|---|---|
| | | Precision | Recall | F1 | Precision | Recall | F1 |
| Gemini-2.5-Flash | 0.3315 | 0.0000 | 0.0000 | 0.0000 | 0.0000 | 0.0000 | 0.0000 |
| Gemini-2.5-Pro | 0.7254 | 0.0000 | 0.0000 | 0.0000 | 1.0000 | 0.5000 | 0.6667 |
| Gpt-4.1-mini | 0.9608 | 0.0000 | 0.0000 | 0.0000 | 0.0000 | 0.0000 | 0.0000 |
| Gpt-4.1 | 0.9407 | 0.0000 | 0.0000 | 0.0000 | 0.0000 | 0.0000 | 0.0000 |
| Claude-3.5-Haiku | 0.9608 | 0.0000 | 0.0000 | 0.0000 | 0.0000 | 0.0000 | 0.0000 |
| Claude-3.7-Sonnet | 0.7308 | 0.0000 | 0.0000 | 0.0000 | 0.0000 | 0.0000 | 0.0000 |

Table S70: Detailed results for benchmark BIOMD0000000753 across all models.

| Model | STE ↓ | RMS (with modifiers) ↑ | | | RMS (without modifiers) ↑ | | |
|---|---|---|---|---|---|---|---|
| | | Precision | Recall | F1 | Precision | Recall | F1 |
| Gemini-2.5-Flash | 0.0200 | 0.0000 | 0.0000 | 0.0000 | 0.0000 | 0.0000 | 0.0000 |
| Gemini-2.5-Pro | 0.7700 | 0.0000 | 0.0000 | 0.0000 | 0.0000 | 0.0000 | 0.0000 |
| Gpt-4.1-mini | 0.5427 | 0.5000 | 0.2857 | 0.3636 | 0.5000 | 0.5000 | 0.5000 |
| Gpt-4.1 | 0.7924 | 0.0000 | 0.0000 | 0.0000 | 0.0000 | 0.0000 | 0.0000 |
| Claude-3.5-Haiku | 0.5112 | 1.0000 | 0.2857 | 0.4444 | 1.0000 | 0.5000 | 0.6667 |
| Claude-3.7-Sonnet | 0.0113 | 0.5000 | 0.2857 | 0.3636 | 0.5000 | 0.5000 | 0.5000 |

Table S71: Detailed results for benchmark BIOMD0000000755 across all models.

| Model | STE ↓ | RMS (with modifiers) ↑ | | | RMS (without modifiers) ↑ | | |
|---|---|---|---|---|---|---|---|
| | | Precision | Recall | F1 | Precision | Recall | F1 |
| Gemini-2.5-Flash | 0.5058 | 0.0000 | 0.0000 | 0.0000 | 0.0000 | 0.0000 | 0.0000 |
| Gemini-2.5-Pro | 0.5712 | 0.2000 | 0.1429 | 0.1667 | 0.2000 | 0.1429 | 0.1667 |
| Gpt-4.1-mini | 0.7655 | 0.1667 | 0.1429 | 0.1538 | 0.1667 | 0.1429 | 0.1538 |
| Gpt-4.1 | 0.7301 | 0.2500 | 0.1429 | 0.1818 | 0.2500 | 0.1429 | 0.1818 |
| Claude-3.5-Haiku | 0.7671 | 1.0000 | 0.2857 | 0.4444 | 1.0000 | 0.2857 | 0.4444 |
| Claude-3.7-Sonnet | 0.5724 | 0.0000 | 0.0000 | 0.0000 | 0.0000 | 0.0000 | 0.0000 |

Table S72: Detailed results for benchmark BIOMD0000000758 across all models.

| Model | STE ↓ | RMS (with modifiers) ↑ | | | RMS (without modifiers) ↑ | | |
|---|---|---|---|---|---|---|---|
| | | Precision | Recall | F1 | Precision | Recall | F1 |
| Gemini-2.5-Flash | 0.5623 | 0.3333 | 0.2500 | 0.2857 | 0.3333 | 0.2500 | 0.2857 |
| Gemini-2.5-Pro | 0.6223 | 0.2500 | 0.2500 | 0.2500 | 0.5000 | 0.5000 | 0.5000 |
| Gpt-4.1-mini | 0.7293 | 0.2000 | 0.2500 | 0.2222 | 0.6000 | 0.7500 | 0.6667 |
| Gpt-4.1 | 0.6609 | 0.0000 | 0.0000 | 0.0000 | 0.6667 | 0.5000 | 0.5714 |
| Claude-3.5-Haiku | 0.5101 | 0.0000 | 0.0000 | 0.0000 | 0.0000 | 0.0000 | 0.0000 |
| Claude-3.7-Sonnet | 0.8190 | 0.0000 | 0.0000 | 0.0000 | 0.5000 | 0.5000 | 0.5000 |

Table S73: Detailed results for benchmark BIOMD0000000760 across all models.

| Model | STE $\downarrow$ | RMS (with modifiers) $\uparrow$ | | | RMS (without modifiers) $\uparrow$ | | |
| --- | --- | --- | --- | --- | --- | --- | --- |
| | | Precision | Recall | F1 | Precision | Recall | F1 |
| Gemini-2.5-Flash | 0.2239 | 0.5000 | 0.5000 | 0.5000 | 0.7500 | 1.0000 | 0.8571 |
| Gemini-2.5-Pro | 0.4864 | 0.2500 | 0.2500 | 0.2500 | 0.3333 | 0.3333 | 0.3333 |
| Gpt-4.1-mini | 0.9457 | 0.2500 | 0.5000 | 0.3333 | 0.3750 | 1.0000 | 0.5455 |
| Gpt-4.1 | 0.1693 | 0.5000 | 0.5000 | 0.5000 | 0.7500 | 1.0000 | 0.8571 |
| Claude-3.5-Haiku | 0.9801 | 0.0000 | 0.0000 | 0.0000 | 0.0000 | 0.0000 | 0.0000 |
| Claude-3.7-Sonnet | 0.1298 | 0.0000 | 0.0000 | 0.0000 | 0.0000 | 0.0000 | 0.0000 |

Table S74: Detailed results for benchmark BIOMD0000000762 across all models.

| Model | STE $\downarrow$ | RMS (with modifiers) $\uparrow$ | | | RMS (without modifiers) $\uparrow$ | | |
| --- | --- | --- | --- | --- | --- | --- | --- |
| | | Precision | Recall | F1 | Precision | Recall | F1 |
| Gemini-2.5-Flash | 0.3301 | 0.2500 | 0.1429 | 0.1818 | 0.7500 | 0.7500 | 0.7500 |
| Gemini-2.5-Pro | 0.3810 | 0.5000 | 0.2857 | 0.3636 | 1.0000 | 1.0000 | 1.0000 |
| Gpt-4.1-mini | 0.6533 | 0.4000 | 0.2857 | 0.3333 | 0.8000 | 1.0000 | 0.8889 |
| Gpt-4.1 | 0.6205 | 0.2500 | 0.1429 | 0.1818 | 0.5000 | 0.5000 | 0.5000 |
| Claude-3.5-Haiku | 0.6062 | 0.0000 | 0.0000 | 0.0000 | 0.0000 | 0.0000 | 0.0000 |
| Claude-3.7-Sonnet | 0.3334 | 0.1429 | 0.1429 | 0.1429 | 0.2857 | 0.5000 | 0.3636 |

Table S75: Detailed results for benchmark BIOMD0000000763 across all models.

| Model | STE $\downarrow$ | RMS (with modifiers) $\uparrow$ | | | RMS (without modifiers) $\uparrow$ | | |
| --- | --- | --- | --- | --- | --- | --- | --- |
| | | Precision | Recall | F1 | Precision | Recall | F1 |
| Gemini-2.5-Flash | 0.8348 | 0.0000 | 0.0000 | 0.0000 | 0.0000 | 0.0000 | 0.0000 |
| Gemini-2.5-Pro | 0.4946 | 0.3333 | 0.2222 | 0.2667 | 0.8333 | 0.8333 | 0.8333 |
| Gpt-4.1-mini | 0.5109 | 0.0000 | 0.0000 | 0.0000 | 0.5000 | 0.1667 | 0.2500 |
| Gpt-4.1 | 0.9041 | 0.3333 | 0.2222 | 0.2667 | 0.5000 | 0.5000 | 0.5000 |
| Claude-3.5-Haiku | 0.3560 | 0.0000 | 0.0000 | 0.0000 | 0.0000 | 0.0000 | 0.0000 |
| Claude-3.7-Sonnet | 0.6461 | 0.5000 | 0.3333 | 0.4000 | 0.6667 | 0.6667 | 0.6667 |

Table S76: Detailed results for benchmark BIOMD0000000767 across all models.

| Model | STE $\downarrow$ | RMS (with modifiers) $\uparrow$ | | | RMS (without modifiers) $\uparrow$ | | |
| --- | --- | --- | --- | --- | --- | --- | --- |
| | | Precision | Recall | F1 | Precision | Recall | F1 |
| Gemini-2.5-Flash | 0.3167 | 0.3333 | 0.2000 | 0.2500 | 0.6667 | 0.5000 | 0.5714 |
| Gemini-2.5-Pro | 0.0859 | 0.5000 | 0.4000 | 0.4444 | 1.0000 | 1.0000 | 1.0000 |
| Gpt-4.1-mini | 0.5740 | 0.0000 | 0.0000 | 0.0000 | 0.5000 | 0.5000 | 0.5000 |
| Gpt-4.1 | 0.4296 | 0.0000 | 0.0000 | 0.0000 | 0.0000 | 0.0000 | 0.0000 |
| Claude-3.5-Haiku | 0.6288 | 0.6667 | 0.4000 | 0.5000 | 0.6667 | 0.5000 | 0.5714 |
| Claude-3.7-Sonnet | 0.0971 | 0.2000 | 0.2000 | 0.2000 | 0.4000 | 0.5000 | 0.4444 |

Table S77: Detailed results for benchmark BIOMD0000000771 across all models.

| Model | STE ↓ | RMS (with modifiers) ↑ | | | RMS (without modifiers) ↑ | | |
|---|---|---|---|---|---|---|---|
| | | Precision | Recall | F1 | Precision | Recall | F1 |
| Gemini-2.5-Flash | 0.5263 | 0.5000 | 0.1667 | 0.2500 | 0.5000 | 0.1667 | 0.2500 |
| Gemini-2.5-Pro | 0.6038 | 0.0000 | 0.0000 | 0.0000 | 0.0000 | 0.0000 | 0.0000 |
| Gpt-4.1-mini | 0.9068 | 0.0000 | 0.0000 | 0.0000 | 0.4000 | 0.3333 | 0.3636 |
| Gpt-4.1 | 0.6406 | 0.5000 | 0.3333 | 0.4000 | 0.5000 | 0.3333 | 0.4000 |
| Claude-3.5-Haiku | 0.9123 | 0.0000 | 0.0000 | 0.0000 | 0.0000 | 0.0000 | 0.0000 |
| Claude-3.7-Sonnet | 1.0000 | 0.3333 | 0.1667 | 0.2222 | 0.3333 | 0.1667 | 0.2222 |

Table S78: Detailed results for benchmark BIOMD0000000772 across all models.

| Model | STE ↓ | RMS (with modifiers) ↑ | | | RMS (without modifiers) ↑ | | |
|---|---|---|---|---|---|---|---|
| | | Precision | Recall | F1 | Precision | Recall | F1 |
| Gemini-2.5-Flash | 0.6903 | 0.0000 | 0.0000 | 0.0000 | 0.0000 | 0.0000 | 0.0000 |
| Gemini-2.5-Pro | 0.4789 | 0.0000 | 0.0000 | 0.0000 | 0.0000 | 0.0000 | 0.0000 |
| Gpt-4.1-mini | 0.5893 | 0.0000 | 0.0000 | 0.0000 | 0.0000 | 0.0000 | 0.0000 |
| Gpt-4.1 | 0.5102 | 0.0000 | 0.0000 | 0.0000 | 0.3333 | 0.1667 | 0.2222 |
| Claude-3.5-Haiku | 0.5716 | 0.0000 | 0.0000 | 0.0000 | 0.0000 | 0.0000 | 0.0000 |
| Claude-3.7-Sonnet | 0.5113 | 0.0000 | 0.0000 | 0.0000 | 0.0000 | 0.0000 | 0.0000 |

Table S79: Detailed results for benchmark BIOMD0000000777 across all models.

| Model | STE ↓ | RMS (with modifiers) ↑ | | | RMS (without modifiers) ↑ | | |
|---|---|---|---|---|---|---|---|
| | | Precision | Recall | F1 | Precision | Recall | F1 |
| Gemini-2.5-Flash | 1.0000 | 0.0000 | 0.0000 | 0.0000 | 0.0000 | 0.0000 | 0.0000 |
| Gemini-2.5-Pro | 0.0943 | 0.5000 | 0.1250 | 0.2000 | 0.5000 | 0.1667 | 0.2500 |
| Gpt-4.1-mini | 0.0511 | 0.0000 | 0.0000 | 0.0000 | 0.0000 | 0.0000 | 0.0000 |
| Gpt-4.1 | 0.3728 | 0.0000 | 0.0000 | 0.0000 | 0.3333 | 0.1667 | 0.2222 |
| Claude-3.5-Haiku | 0.6439 | 0.0000 | 0.0000 | 0.0000 | 0.0000 | 0.0000 | 0.0000 |
| Claude-3.7-Sonnet | 0.0323 | 0.0000 | 0.0000 | 0.0000 | 0.0000 | 0.0000 | 0.0000 |

Table S80: Detailed results for benchmark BIOMD0000000783 across all models.

| Model | STE ↓ | RMS (with modifiers) ↑ | | | RMS (without modifiers) ↑ | | |
|---|---|---|---|---|---|---|---|
| | | Precision | Recall | F1 | Precision | Recall | F1 |
| Gemini-2.5-Flash | 0.0345 | 0.6000 | 0.3750 | 0.4615 | 0.6000 | 0.5000 | 0.5455 |
| Gemini-2.5-Pro | 0.1222 | 0.4000 | 0.2500 | 0.3077 | 0.4000 | 0.3333 | 0.3636 |
| Gpt-4.1-mini | 0.2966 | 0.4444 | 0.5000 | 0.4706 | 0.6667 | 1.0000 | 0.8000 |
| Gpt-4.1 | 0.3509 | 0.4000 | 0.2500 | 0.3077 | 0.4000 | 0.3333 | 0.3636 |
| Claude-3.5-Haiku | 0.5880 | 0.0000 | 0.0000 | 0.0000 | 0.0000 | 0.0000 | 0.0000 |
| Claude-3.7-Sonnet | 0.6153 | 0.1429 | 0.1250 | 0.1333 | 0.2857 | 0.3333 | 0.3077 |

Table S81: Detailed results for benchmark BIOMD0000000785 across all models.

| Model | STE ↓ | RMS (with modifiers) ↑ | | | RMS (without modifiers) ↑ | | |
|---|---|---|---|---|---|---|---|
| | | Precision | Recall | F1 | Precision | Recall | F1 |
| Gemini-2.5-Flash | 0.6810 | 0.5000 | 0.2000 | 0.2857 | 0.5000 | 0.2500 | 0.3333 |
| Gemini-2.5-Pro | 0.7157 | 0.5000 | 0.4000 | 0.4444 | 0.5000 | 0.5000 | 0.5000 |
| Gpt-4.1-mini | 0.9884 | 1.0000 | 0.4000 | 0.5714 | 1.0000 | 0.5000 | 0.6667 |
| Gpt-4.1 | 0.5916 | 0.7500 | 0.6000 | 0.6667 | 0.7500 | 0.7500 | 0.7500 |
| Claude-3.5-Haiku | 0.7081 | 0.0000 | 0.0000 | 0.0000 | 0.0000 | 0.0000 | 0.0000 |
| Claude-3.7-Sonnet | 0.1420 | 0.5000 | 0.6000 | 0.5455 | 0.5000 | 0.7500 | 0.6000 |

Table S82: Detailed results for benchmark BIOMD0000000793 across all models.

| Model | STE ↓ | RMS (with modifiers) ↑ | | | RMS (without modifiers) ↑ | | |
|---|---|---|---|---|---|---|---|
| | | Precision | Recall | F1 | Precision | Recall | F1 |
| Gemini-2.5-Flash | 0.2251 | 0.6667 | 0.4000 | 0.5000 | 0.6667 | 0.5000 | 0.5714 |
| Gemini-2.5-Pro | 0.1611 | 0.5000 | 0.2000 | 0.2857 | 0.5000 | 0.2500 | 0.3333 |
| Gpt-4.1-mini | 0.2318 | 0.0000 | 0.0000 | 0.0000 | 0.0000 | 0.0000 | 0.0000 |
| Gpt-4.1 | 0.4011 | 0.0000 | 0.0000 | 0.0000 | 1.0000 | 0.2500 | 0.4000 |
| Claude-3.5-Haiku | 0.3884 | 0.0000 | 0.0000 | 0.0000 | 0.0000 | 0.0000 | 0.0000 |
| Claude-3.7-Sonnet | 0.2087 | 0.0000 | 0.0000 | 0.0000 | 0.0000 | 0.0000 | 0.0000 |

Table S83: Detailed results for benchmark BIOMD0000000795 across all models.

| Model | STE ↓ | RMS (with modifiers) ↑ | | | RMS (without modifiers) ↑ | | |
|---|---|---|---|---|---|---|---|
| | | Precision | Recall | F1 | Precision | Recall | F1 |
| Gemini-2.5-Flash | 0.2925 | 0.2500 | 0.1667 | 0.2000 | 0.7500 | 0.7500 | 0.7500 |
| Gemini-2.5-Pro | 0.2218 | 0.3333 | 0.1667 | 0.2222 | 0.6667 | 0.5000 | 0.5714 |
| Gpt-4.1-mini | 0.4772 | 0.4000 | 0.3333 | 0.3636 | 0.6000 | 0.7500 | 0.6667 |
| Gpt-4.1 | 0.2908 | 0.3333 | 0.1667 | 0.2222 | 0.6667 | 0.5000 | 0.5714 |
| Claude-3.5-Haiku | 0.7931 | 0.0000 | 0.0000 | 0.0000 | 0.0000 | 0.0000 | 0.0000 |
| Claude-3.7-Sonnet | 0.2540 | 0.2500 | 0.1667 | 0.2000 | 0.5000 | 0.5000 | 0.5000 |

Table S84: Detailed results for benchmark BIOMD0000000799 across all models.

| Model | STE ↓ | RMS (with modifiers) ↑ | | | RMS (without modifiers) ↑ | | |
|---|---|---|---|---|---|---|---|
| | | Precision | Recall | F1 | Precision | Recall | F1 |
| Gemini-2.5-Flash | 0.0052 | 0.6667 | 0.5000 | 0.5714 | 0.6667 | 0.5000 | 0.5714 |
| Gemini-2.5-Pro | 0.0139 | 0.0000 | 0.0000 | 0.0000 | 0.0000 | 0.0000 | 0.0000 |
| Gpt-4.1-mini | 0.0539 | 1.0000 | 0.5000 | 0.6667 | 1.0000 | 0.5000 | 0.6667 |
| Gpt-4.1 | 0.0477 | 0.6667 | 0.5000 | 0.5714 | 0.6667 | 0.5000 | 0.5714 |
| Claude-3.5-Haiku | 0.1059 | 1.0000 | 0.5000 | 0.6667 | 1.0000 | 0.5000 | 0.6667 |
| Claude-3.7-Sonnet | 0.0007 | 0.5000 | 0.5000 | 0.5000 | 1.0000 | 1.0000 | 1.0000 |

Table S85: Detailed results for benchmark BIOMD0000000800 across all models.

| Model | STE ↓ | RMS (with modifiers) ↑ | | | RMS (without modifiers) ↑ | | |
|---|---|---|---|---|---|---|---|
| | | Precision | Recall | F1 | Precision | Recall | F1 |
| Gemini-2.5-Flash | 0.6423 | 0.0000 | 0.0000 | 0.0000 | 0.0000 | 0.0000 | 0.0000 |
| Gemini-2.5-Pro | 0.6242 | 0.0000 | 0.0000 | 0.0000 | 0.0000 | 0.0000 | 0.0000 |
| Gpt-4.1-mini | 0.4681 | 0.6000 | 0.5000 | 0.5455 | 0.6000 | 0.5000 | 0.5455 |
| Gpt-4.1 | 0.7413 | 0.0000 | 0.0000 | 0.0000 | 1.0000 | 0.5000 | 0.6667 |
| Claude-3.5-Haiku | 0.4336 | 0.0000 | 0.0000 | 0.0000 | 0.0000 | 0.0000 | 0.0000 |
| Claude-3.7-Sonnet | 0.5820 | 0.2500 | 0.1667 | 0.2000 | 0.2500 | 0.1667 | 0.2000 |

Table S86: Detailed results for benchmark BIOMD0000000813 across all models.

| Model | STE ↓ | RMS (with modifiers) ↑ | | | RMS (without modifiers) ↑ | | |
|---|---|---|---|---|---|---|---|
| | | Precision | Recall | F1 | Precision | Recall | F1 |
| Gemini-2.5-Flash | 0.6201 | 0.0000 | 0.0000 | 0.0000 | 0.0000 | 0.0000 | 0.0000 |
| Gemini-2.5-Pro | 0.1664 | 0.5000 | 0.2500 | 0.3333 | 0.5000 | 0.3333 | 0.4000 |
| Gpt-4.1-mini | 0.9950 | 0.0000 | 0.0000 | 0.0000 | 0.0000 | 0.0000 | 0.0000 |
| Gpt-4.1 | 0.2338 | 0.7500 | 0.3750 | 0.5000 | 0.7500 | 0.5000 | 0.6000 |
| Claude-3.5-Haiku | 0.4548 | 0.0000 | 0.0000 | 0.0000 | 0.5000 | 0.1667 | 0.2500 |
| Claude-3.7-Sonnet | 0.1605 | 0.5000 | 0.2500 | 0.3333 | 0.5000 | 0.3333 | 0.4000 |

Table S87: Detailed results for benchmark BIOMD0000000821 across all models.

| Model | STE ↓ | RMS (with modifiers) ↑ | | | RMS (without modifiers) ↑ | | |
|---|---|---|---|---|---|---|---|
| | | Precision | Recall | F1 | Precision | Recall | F1 |
| Gemini-2.5-Flash | 0.3363 | 0.0000 | 0.0000 | 0.0000 | 0.0000 | 0.0000 | 0.0000 |
| Gemini-2.5-Pro | 0.0417 | 0.3333 | 0.2857 | 0.3077 | 0.5000 | 0.6000 | 0.5455 |
| Gpt-4.1-mini | 0.9953 | 0.0000 | 0.0000 | 0.0000 | 0.0000 | 0.0000 | 0.0000 |
| Gpt-4.1 | 0.6666 | 0.2500 | 0.1429 | 0.1818 | 0.5000 | 0.4000 | 0.4444 |
| Claude-3.5-Haiku | 0.6647 | 0.0000 | 0.0000 | 0.0000 | 0.0000 | 0.0000 | 0.0000 |
| Claude-3.7-Sonnet | 0.0721 | 0.0000 | 0.0000 | 0.0000 | 0.0000 | 0.0000 | 0.0000 |

Table S88: Detailed results for benchmark BIOMD0000000827 across all models.

| Model | STE ↓ | RMS (with modifiers) ↑ | | | RMS (without modifiers) ↑ | | |
|---|---|---|---|---|---|---|---|
| | | Precision | Recall | F1 | Precision | Recall | F1 |
| Gemini-2.5-Flash | 0.7074 | 0.0000 | 0.0000 | 0.0000 | 0.0000 | 0.0000 | 0.0000 |
| Gemini-2.5-Pro | 0.2255 | 0.3750 | 0.3333 | 0.3529 | 0.3750 | 0.3333 | 0.3529 |
| Gpt-4.1-mini | 0.8146 | 0.2857 | 0.2222 | 0.2500 | 0.2857 | 0.2222 | 0.2500 |
| Gpt-4.1 | 0.5029 | 0.1667 | 0.1111 | 0.1333 | 0.1667 | 0.1111 | 0.1333 |
| Claude-3.5-Haiku | 1.0000 | 0.0000 | 0.0000 | 0.0000 | 0.0000 | 0.0000 | 0.0000 |
| Claude-3.7-Sonnet | 0.2436 | 0.0000 | 0.0000 | 0.0000 | 0.0000 | 0.0000 | 0.0000 |

Table S89: Detailed results for benchmark BIOMD0000000830 across all models.

| Model | STE ↓ | RMS (with modifiers) ↑ | | | RMS (without modifiers) ↑ | | |
| --- | --- | --- | --- | --- | --- | --- | --- |
| | | Precision | Recall | F1 | Precision | Recall | F1 |
| Gemini-2.5-Flash | 0.1653 | 0.0000 | 0.0000 | 0.0000 | 0.0000 | 0.0000 | 0.0000 |
| Gemini-2.5-Pro | 0.0600 | 0.0000 | 0.0000 | 0.0000 | 0.0000 | 0.0000 | 0.0000 |
| Gpt-4.1-mini | 0.4087 | 0.1667 | 0.1429 | 0.1538 | 0.1667 | 0.1429 | 0.1538 |
| Gpt-4.1 | 0.2358 | 0.2000 | 0.1429 | 0.1667 | 0.2000 | 0.1429 | 0.1667 |
| Claude-3.5-Haiku | 0.4174 | 0.0000 | 0.0000 | 0.0000 | 0.0000 | 0.0000 | 0.0000 |
| Claude-3.7-Sonnet | 0.0891 | 0.2308 | 0.4286 | 0.3000 | 0.2308 | 0.4286 | 0.3000 |

Table S90: Detailed results for benchmark BIOMD0000000838 across all models.

| Model | STE ↓ | RMS (with modifiers) ↑ | | | RMS (without modifiers) ↑ | | |
| --- | --- | --- | --- | --- | --- | --- | --- |
| | | Precision | Recall | F1 | Precision | Recall | F1 |
| Gemini-2.5-Flash | 1.0000 | 0.0000 | 0.0000 | 0.0000 | 0.0000 | 0.0000 | 0.0000 |
| Gemini-2.5-Pro | 0.0065 | 0.5000 | 0.5000 | 0.5000 | 1.0000 | 1.0000 | 1.0000 |
| Gpt-4.1-mini | 0.5255 | 0.3333 | 0.3333 | 0.3333 | 0.5000 | 0.5000 | 0.5000 |
| Gpt-4.1 | 0.0984 | 0.1667 | 0.1667 | 0.1667 | 0.5000 | 0.5000 | 0.5000 |
| Claude-3.5-Haiku | 0.6204 | 0.0000 | 0.0000 | 0.0000 | 0.0000 | 0.0000 | 0.0000 |
| Claude-3.7-Sonnet | 0.2134 | 0.2500 | 0.1667 | 0.2000 | 0.2500 | 0.1667 | 0.2000 |

Table S91: Detailed results for benchmark BIOMD0000000847 across all models.

| Model | STE ↓ | RMS (with modifiers) ↑ | | | RMS (without modifiers) ↑ | | |
| --- | --- | --- | --- | --- | --- | --- | --- |
| | | Precision | Recall | F1 | Precision | Recall | F1 |
| Gemini-2.5-Flash | 0.3448 | 0.0000 | 0.0000 | 0.0000 | 0.0000 | 0.0000 | 0.0000 |
| Gemini-2.5-Pro | 0.7626 | 0.0000 | 0.0000 | 0.0000 | 0.0000 | 0.0000 | 0.0000 |
| Gpt-4.1-mini | 0.7443 | 0.5714 | 0.6667 | 0.6154 | 0.5714 | 0.6667 | 0.6154 |
| Gpt-4.1 | 0.7050 | 0.2500 | 0.3333 | 0.2857 | 0.2500 | 0.3333 | 0.2857 |
| Claude-3.5-Haiku | 0.9991 | 0.0000 | 0.0000 | 0.0000 | 0.0000 | 0.0000 | 0.0000 |
| Claude-3.7-Sonnet | 0.6848 | 0.3750 | 0.5000 | 0.4286 | 0.3750 | 0.5000 | 0.4286 |

Table S92: Detailed results for benchmark BIOMD0000000850 across all models.

| Model | STE ↓ | RMS (with modifiers) ↑ | | | RMS (without modifiers) ↑ | | |
| --- | --- | --- | --- | --- | --- | --- | --- |
| | | Precision | Recall | F1 | Precision | Recall | F1 |
| Gemini-2.5-Flash | 0.0257 | 0.0000 | 0.0000 | 0.0000 | 0.0000 | 0.0000 | 0.0000 |
| Gemini-2.5-Pro | 0.0143 | 0.5000 | 0.2500 | 0.3333 | 1.0000 | 0.5000 | 0.6667 |
| Gpt-4.1-mini | 0.4542 | 0.5000 | 0.2500 | 0.3333 | 0.5000 | 0.2500 | 0.3333 |
| Gpt-4.1 | 0.4082 | 0.2500 | 0.2500 | 0.2500 | 0.2500 | 0.2500 | 0.2500 |
| Claude-3.5-Haiku | 0.4525 | 0.0000 | 0.0000 | 0.0000 | 0.0000 | 0.0000 | 0.0000 |
| Claude-3.7-Sonnet | 0.1244 | 0.3333 | 0.5000 | 0.4000 | 0.3333 | 0.5000 | 0.4000 |

Table S93: Detailed results for benchmark BIOMD0000000876 across all models.

| Model | STE ↓ | RMS (with modifiers) ↑ | | | RMS (without modifiers) ↑ | | |
| --- | --- | --- | --- | --- | --- | --- | --- |
| | | Precision | Recall | F1 | Precision | Recall | F1 |
| Gemini-2.5-Flash | 0.6702 | 0.0000 | 0.0000 | 0.0000 | 0.0000 | 0.0000 | 0.0000 |
| Gemini-2.5-Pro | 0.3866 | 0.5000 | 0.3333 | 0.4000 | 0.6667 | 0.5000 | 0.5714 |
| Gpt-4.1-mini | 0.8588 | 0.3333 | 0.2222 | 0.2667 | 0.3333 | 0.2500 | 0.2857 |
| Gpt-4.1 | 0.9531 | 0.5000 | 0.2222 | 0.3077 | 0.5000 | 0.2500 | 0.3333 |
| Claude-3.5-Haiku | 0.5998 | 0.0000 | 0.0000 | 0.0000 | 0.0000 | 0.0000 | 0.0000 |
| Claude-3.7-Sonnet | 0.7690 | 0.2500 | 0.1111 | 0.1538 | 0.2500 | 0.1250 | 0.1667 |

Table S94: Detailed results for benchmark BIOMD0000000877 across all models.

| Model | STE ↓ | RMS (with modifiers) ↑ | | | RMS (without modifiers) ↑ | | |
| --- | --- | --- | --- | --- | --- | --- | --- |
| | | Precision | Recall | F1 | Precision | Recall | F1 |
| Gemini-2.5-Flash | 0.1082 | 0.5000 | 0.2222 | 0.3077 | 0.7500 | 0.3750 | 0.5000 |
| Gemini-2.5-Pro | 0.3931 | 0.2500 | 0.1111 | 0.1538 | 0.2500 | 0.1250 | 0.1667 |
| Gpt-4.1-mini | 0.7864 | 0.5000 | 0.4444 | 0.4706 | 0.6250 | 0.6250 | 0.6250 |
| Gpt-4.1 | 0.2414 | 0.6667 | 0.2222 | 0.3333 | 1.0000 | 0.3750 | 0.5455 |
| Claude-3.5-Haiku | 0.7771 | 0.0000 | 0.0000 | 0.0000 | 0.0000 | 0.0000 | 0.0000 |
| Claude-3.7-Sonnet | 0.6846 | 0.2500 | 0.1111 | 0.1538 | 0.2500 | 0.1250 | 0.1667 |

Table S95: Detailed results for benchmark BIOMD0000000878 across all models.

| Model | STE ↓ | RMS (with modifiers) ↑ | | | RMS (without modifiers) ↑ | | |
| --- | --- | --- | --- | --- | --- | --- | --- |
| | | Precision | Recall | F1 | Precision | Recall | F1 |
| Gemini-2.5-Flash | 0.3099 | 0.0000 | 0.0000 | 0.0000 | 0.0000 | 0.0000 | 0.0000 |
| Gemini-2.5-Pro | 0.1889 | 0.1667 | 0.1667 | 0.1667 | 1.0000 | 1.0000 | 1.0000 |
| Gpt-4.1-mini | 0.4244 | 0.0000 | 0.0000 | 0.0000 | 0.0000 | 0.0000 | 0.0000 |
| Gpt-4.1 | 0.6055 | 0.0000 | 0.0000 | 0.0000 | 0.0000 | 0.0000 | 0.0000 |
| Claude-3.5-Haiku | 0.5460 | 0.0000 | 0.0000 | 0.0000 | 0.0000 | 0.0000 | 0.0000 |
| Claude-3.7-Sonnet | 0.2081 | 0.0000 | 0.0000 | 0.0000 | 0.0000 | 0.0000 | 0.0000 |

Table S96: Detailed results for benchmark BIOMD0000000882 across all models.

| Model | STE ↓ | RMS (with modifiers) ↑ | | | RMS (without modifiers) ↑ | | |
| --- | --- | --- | --- | --- | --- | --- | --- |
| | | Precision | Recall | F1 | Precision | Recall | F1 |
| Gemini-2.5-Flash | 0.4930 | 0.0000 | 0.0000 | 0.0000 | 0.0000 | 0.0000 | 0.0000 |
| Gemini-2.5-Pro | 0.2337 | 0.0000 | 0.0000 | 0.0000 | 0.0000 | 0.0000 | 0.0000 |
| Gpt-4.1-mini | 0.4131 | 0.0000 | 0.0000 | 0.0000 | 0.0000 | 0.0000 | 0.0000 |
| Gpt-4.1 | 0.4068 | 0.0000 | 0.0000 | 0.0000 | 0.0000 | 0.0000 | 0.0000 |
| Claude-3.5-Haiku | 0.4811 | 0.0000 | 0.0000 | 0.0000 | 0.0000 | 0.0000 | 0.0000 |
| Claude-3.7-Sonnet | 0.3510 | 0.0000 | 0.0000 | 0.0000 | 0.5000 | 0.3333 | 0.4000 |

Table S97: Detailed results for benchmark BIOMD0000000885 across all models.

| Model | STE ↓ | RMS (with modifiers) ↑ | | | RMS (without modifiers) ↑ | | |
|---|---|---|---|---|---|---|---|
| | | Precision | Recall | F1 | Precision | Recall | F1 |
| Gemini-2.5-Flash | 0.5590 | 0.0000 | 0.0000 | 0.0000 | 0.0000 | 0.0000 | 0.0000 |
| Gemini-2.5-Pro | 0.3732 | 0.0000 | 0.0000 | 0.0000 | 0.0000 | 0.0000 | 0.0000 |
| Gpt-4.1-mini | 0.9399 | 0.5000 | 0.2500 | 0.3333 | 0.7500 | 0.3750 | 0.5000 |
| Gpt-4.1 | 0.2855 | 0.5714 | 0.5000 | 0.5333 | 0.8571 | 0.7500 | 0.8000 |
| Claude-3.5-Haiku | 0.4602 | 0.0000 | 0.0000 | 0.0000 | 0.0000 | 0.0000 | 0.0000 |
| Claude-3.7-Sonnet | 0.2594 | 0.0000 | 0.0000 | 0.0000 | 0.0000 | 0.0000 | 0.0000 |

Table S98: Detailed results for benchmark BIOMD0000000888 across all models.

| Model | STE ↓ | RMS (with modifiers) ↑ | | | RMS (without modifiers) ↑ | | |
|---|---|---|---|---|---|---|---|
| | | Precision | Recall | F1 | Precision | Recall | F1 |
| Gemini-2.5-Flash | 0.4978 | 0.0000 | 0.0000 | 0.0000 | 0.0000 | 0.0000 | 0.0000 |
| Gemini-2.5-Pro | 0.9439 | 0.0000 | 0.0000 | 0.0000 | 0.4000 | 0.2500 | 0.3077 |
| Gpt-4.1-mini | 0.9667 | 0.0000 | 0.0000 | 0.0000 | 0.0000 | 0.0000 | 0.0000 |
| Gpt-4.1 | 0.7537 | 0.0000 | 0.0000 | 0.0000 | 0.0000 | 0.0000 | 0.0000 |
| Claude-3.5-Haiku | 0.9916 | 0.0000 | 0.0000 | 0.0000 | 0.0000 | 0.0000 | 0.0000 |
| Claude-3.7-Sonnet | 0.7780 | 0.0000 | 0.0000 | 0.0000 | 0.0000 | 0.0000 | 0.0000 |

Table S99: Detailed results for benchmark BIOMD0000000891 across all models.

| Model | STE ↓ | RMS (with modifiers) ↑ | | | RMS (without modifiers) ↑ | | |
|---|---|---|---|---|---|---|---|
| | | Precision | Recall | F1 | Precision | Recall | F1 |
| Gemini-2.5-Flash | 0.4385 | 0.0000 | 0.0000 | 0.0000 | 0.0000 | 0.0000 | 0.0000 |
| Gemini-2.5-Pro | 0.7968 | 0.0000 | 0.0000 | 0.0000 | 0.0000 | 0.0000 | 0.0000 |
| Gpt-4.1-mini | 0.6286 | 0.0000 | 0.0000 | 0.0000 | 0.0000 | 0.0000 | 0.0000 |
| Gpt-4.1 | 0.3810 | 0.0000 | 0.0000 | 0.0000 | 0.0000 | 0.0000 | 0.0000 |
| Claude-3.5-Haiku | 0.6326 | 0.0000 | 0.0000 | 0.0000 | 0.0000 | 0.0000 | 0.0000 |
| Claude-3.7-Sonnet | 0.2680 | 0.2000 | 0.1667 | 0.1818 | 0.2000 | 0.1667 | 0.1818 |

Table S100: Detailed results for benchmark BIOMD0000000892 across all models.

| Model | STE ↓ | RMS (with modifiers) ↑ | | | RMS (without modifiers) ↑ | | |
|---|---|---|---|---|---|---|---|
| | | Precision | Recall | F1 | Precision | Recall | F1 |
| Gemini-2.5-Flash | 0.2837 | 0.0000 | 0.0000 | 0.0000 | 0.0000 | 0.0000 | 0.0000 |
| Gemini-2.5-Pro | 0.2568 | 0.4000 | 0.2500 | 0.3077 | 0.6000 | 0.3750 | 0.4615 |
| Gpt-4.1-mini | 0.6208 | 0.2222 | 0.2500 | 0.2353 | 0.3333 | 0.3750 | 0.3529 |
| Gpt-4.1 | 0.9315 | 0.0000 | 0.0000 | 0.0000 | 0.0000 | 0.0000 | 0.0000 |
| Claude-3.5-Haiku | 0.6680 | 0.0000 | 0.0000 | 0.0000 | 0.0000 | 0.0000 | 0.0000 |
| Claude-3.7-Sonnet | 0.3288 | 0.0000 | 0.0000 | 0.0000 | 0.0000 | 0.0000 | 0.0000 |

Table S101: Detailed results for benchmark BIOMD0000000893 across all models.

| Model | STE ↓ | RMS (with modifiers) ↑ | | | RMS (without modifiers) ↑ | | |
|---|---|---|---|---|---|---|---|
| | | Precision | Recall | F1 | Precision | Recall | F1 |
| Gemini-2.5-Flash | 1.0000 | 0.0000 | 0.0000 | 0.0000 | 0.0000 | 0.0000 | 0.0000 |
| Gemini-2.5-Pro | 0.6606 | 0.1667 | 0.1667 | 0.1667 | 0.6667 | 0.6667 | 0.6667 |
| Gpt-4.1-mini | 1.0000 | 0.0000 | 0.0000 | 0.0000 | 0.0000 | 0.0000 | 0.0000 |
| Gpt-4.1 | 0.7179 | 0.1429 | 0.1667 | 0.1538 | 0.7143 | 0.8333 | 0.7692 |
| Claude-3.5-Haiku | 0.9990 | 0.0000 | 0.0000 | 0.0000 | 0.0000 | 0.0000 | 0.0000 |
| Claude-3.7-Sonnet | 0.6376 | 0.2000 | 0.3333 | 0.2500 | 0.7500 | 1.0000 | 0.8571 |

Table S102: Detailed results for benchmark BIOMD0000000894 across all models.

| Model | STE ↓ | RMS (with modifiers) ↑ | | | RMS (without modifiers) ↑ | | |
|---|---|---|---|---|---|---|---|
| | | Precision | Recall | F1 | Precision | Recall | F1 |
| Gemini-2.5-Flash | 0.8991 | 0.0000 | 0.0000 | 0.0000 | 0.0000 | 0.0000 | 0.0000 |
| Gemini-2.5-Pro | 0.8153 | 0.0000 | 0.0000 | 0.0000 | 0.3333 | 0.3333 | 0.3333 |
| Gpt-4.1-mini | 0.9424 | 0.0000 | 0.0000 | 0.0000 | 0.0000 | 0.0000 | 0.0000 |
| Gpt-4.1 | 0.4601 | 0.2500 | 0.1667 | 0.2000 | 0.5000 | 0.3333 | 0.4000 |
| Claude-3.5-Haiku | 0.7297 | 0.0000 | 0.0000 | 0.0000 | 0.5000 | 0.1667 | 0.2500 |
| Claude-3.7-Sonnet | 0.4252 | 0.3333 | 0.3333 | 0.3333 | 0.6667 | 0.6667 | 0.6667 |

Table S103: Detailed results for benchmark BIOMD0000000897 across all models.

| Model | STE ↓ | RMS (with modifiers) ↑ | | | RMS (without modifiers) ↑ | | |
|---|---|---|---|---|---|---|---|
| | | Precision | Recall | F1 | Precision | Recall | F1 |
| Gemini-2.5-Flash | 1.0000 | 0.0000 | 0.0000 | 0.0000 | 0.0000 | 0.0000 | 0.0000 |
| Gemini-2.5-Pro | 0.1853 | 0.2500 | 0.2500 | 0.2500 | 1.0000 | 1.0000 | 1.0000 |
| Gpt-4.1-mini | 0.7894 | 0.2000 | 0.2500 | 0.2222 | 0.8000 | 1.0000 | 0.8889 |
| Gpt-4.1 | 0.1427 | 0.2500 | 0.2500 | 0.2500 | 0.7500 | 0.7500 | 0.7500 |
| Claude-3.5-Haiku | 0.0313 | 0.0000 | 0.0000 | 0.0000 | 0.0000 | 0.0000 | 0.0000 |
| Claude-3.7-Sonnet | 0.0221 | 0.5000 | 0.5000 | 0.5000 | 1.0000 | 1.0000 | 1.0000 |

Table S104: Detailed results for benchmark BIOMD0000000899 across all models.

| Model | STE ↓ | RMS (with modifiers) ↑ | | | RMS (without modifiers) ↑ | | |
|---|---|---|---|---|---|---|---|
| | | Precision | Recall | F1 | Precision | Recall | F1 |
| Gemini-2.5-Flash | 0.5355 | 0.1111 | 0.1250 | 0.1176 | 0.0000 | 0.0000 | 0.0000 |
| Gemini-2.5-Pro | 0.3371 | 0.2500 | 0.2500 | 0.2500 | 0.3750 | 0.3750 | 0.3750 |
| Gpt-4.1-mini | 0.6812 | 0.2000 | 0.1250 | 0.1538 | 0.2000 | 0.1250 | 0.1538 |
| Gpt-4.1 | 0.5021 | 0.0000 | 0.0000 | 0.0000 | 0.0000 | 0.0000 | 0.0000 |
| Claude-3.5-Haiku | 0.7967 | 0.0000 | 0.0000 | 0.0000 | 0.0000 | 0.0000 | 0.0000 |
| Claude-3.7-Sonnet | 0.6056 | 0.1667 | 0.1250 | 0.1429 | 0.1667 | 0.1250 | 0.1429 |

Table S105: Detailed results for benchmark BIOMD0000000902 across all models.

| Model | STE ↓ | RMS (with modifiers) ↑ | | | RMS (without modifiers) ↑ | | |
|---|---|---|---|---|---|---|---|
| | | Precision | Recall | F1 | Precision | Recall | F1 |
| Gemini-2.5-Flash | 0.5301 | 0.0000 | 0.0000 | 0.0000 | 0.0000 | 0.0000 | 0.0000 |
| Gemini-2.5-Pro | 0.3585 | 0.0000 | 0.0000 | 0.0000 | 0.0000 | 0.0000 | 0.0000 |
| Gpt-4.1-mini | 0.6784 | 0.0000 | 0.0000 | 0.0000 | 0.0000 | 0.0000 | 0.0000 |
| Gpt-4.1 | 0.5147 | 0.0000 | 0.0000 | 0.0000 | 0.2500 | 0.1667 | 0.2000 |
| Claude-3.5-Haiku | 0.6554 | 0.0000 | 0.0000 | 0.0000 | 0.0000 | 0.0000 | 0.0000 |
| Claude-3.7-Sonnet | 0.5068 | 0.0000 | 0.0000 | 0.0000 | 0.0000 | 0.0000 | 0.0000 |

Table S106: Detailed results for benchmark BIOMD0000000906 across all models.

| Model | STE ↓ | RMS (with modifiers) ↑ | | | RMS (without modifiers) ↑ | | |
|---|---|---|---|---|---|---|---|
| | | Precision | Recall | F1 | Precision | Recall | F1 |
| Gemini-2.5-Flash | 0.4454 | 0.0000 | 0.0000 | 0.0000 | 0.0000 | 0.0000 | 0.0000 |
| Gemini-2.5-Pro | 0.0291 | 0.2857 | 0.3333 | 0.3077 | 0.5714 | 0.6667 | 0.6154 |
| Gpt-4.1-mini | 0.9590 | 0.1429 | 0.1667 | 0.1538 | 0.4286 | 0.5000 | 0.4615 |
| Gpt-4.1 | 0.0571 | 0.3333 | 0.3333 | 0.3333 | 1.0000 | 1.0000 | 1.0000 |
| Claude-3.5-Haiku | 0.5222 | 0.0000 | 0.0000 | 0.0000 | 0.3333 | 0.1667 | 0.2222 |
| Claude-3.7-Sonnet | 0.0188 | 0.4444 | 0.6667 | 0.5333 | 1.0000 | 1.0000 | 1.0000 |

Table S107: Detailed results for benchmark BIOMD0000000909 across all models.

| Model | STE ↓ | RMS (with modifiers) ↑ | | | RMS (without modifiers) ↑ | | |
|---|---|---|---|---|---|---|---|
| | | Precision | Recall | F1 | Precision | Recall | F1 |
| Gemini-2.5-Flash | 0.2396 | 0.0000 | 0.0000 | 0.0000 | 0.0000 | 0.0000 | 0.0000 |
| Gemini-2.5-Pro | 0.0270 | 0.3333 | 0.1250 | 0.1818 | 0.3333 | 0.1250 | 0.1818 |
| Gpt-4.1-mini | 0.0471 | 0.0000 | 0.0000 | 0.0000 | 0.0000 | 0.0000 | 0.0000 |
| Gpt-4.1 | 0.4317 | 0.0000 | 0.0000 | 0.0000 | 0.5000 | 0.1250 | 0.2000 |
| Claude-3.5-Haiku | 0.4575 | 0.0000 | 0.0000 | 0.0000 | 0.0000 | 0.0000 | 0.0000 |
| Claude-3.7-Sonnet | 0.1786 | 0.0000 | 0.0000 | 0.0000 | 0.0000 | 0.0000 | 0.0000 |

Table S108: Detailed results for benchmark BIOMD0000000910 across all models.

| Model | STE ↓ | RMS (with modifiers) ↑ | | | RMS (without modifiers) ↑ | | |
|---|---|---|---|---|---|---|---|
| | | Precision | Recall | F1 | Precision | Recall | F1 |
| Gemini-2.5-Flash | 0.4147 | 0.3333 | 0.2000 | 0.2500 | 0.6667 | 0.4000 | 0.5000 |
| Gemini-2.5-Pro | 0.3626 | 0.1429 | 0.2000 | 0.1667 | 0.7143 | 1.0000 | 0.8333 |
| Gpt-4.1-mini | 0.9724 | 0.0000 | 0.0000 | 0.0000 | 0.4000 | 0.4000 | 0.4000 |
| Gpt-4.1 | 0.1799 | 0.1111 | 0.2000 | 0.1429 | 0.5556 | 1.0000 | 0.7143 |
| Claude-3.5-Haiku | 0.4670 | 0.0000 | 0.0000 | 0.0000 | 0.0000 | 0.0000 | 0.0000 |
| Claude-3.7-Sonnet | 0.6632 | 0.1250 | 0.2000 | 0.1538 | 0.5000 | 0.8000 | 0.6154 |

Table S109: Detailed results for benchmark BIOMD0000000911 across all models.

| Model | STE ↓ | RMS (with modifiers) ↑ | | | RMS (without modifiers) ↑ | | |
|---|---|---|---|---|---|---|---|
| | | Precision | Recall | F1 | Precision | Recall | F1 |
| Gemini-2.5-Flash | 0.1151 | 0.0000 | 0.0000 | 0.0000 | 0.0000 | 0.0000 | 0.0000 |
| Gemini-2.5-Pro | 0.0543 | 0.0000 | 0.0000 | 0.0000 | 0.0000 | 0.0000 | 0.0000 |
| Gpt-4.1-mini | 0.5230 | 0.0000 | 0.0000 | 0.0000 | 0.0000 | 0.0000 | 0.0000 |
| Gpt-4.1 | 0.1657 | 0.3333 | 0.1667 | 0.2222 | 0.6667 | 0.3333 | 0.4444 |
| Claude-3.5-Haiku | 0.7329 | 0.0000 | 0.0000 | 0.0000 | 0.5000 | 0.1667 | 0.2500 |
| Claude-3.7-Sonnet | 0.0385 | 0.4286 | 0.5000 | 0.4615 | 0.8571 | 1.0000 | 0.9231 |

Table S110: Detailed results for benchmark BIOMD0000000912 across all models.

| Model | STE ↓ | RMS (with modifiers) ↑ | | | RMS (without modifiers) ↑ | | |
|---|---|---|---|---|---|---|---|
| | | Precision | Recall | F1 | Precision | Recall | F1 |
| Gemini-2.5-Flash | 1.0000 | 0.3333 | 0.3333 | 0.3333 | 0.5000 | 0.5000 | 0.5000 |
| Gemini-2.5-Pro | 0.7861 | 0.3333 | 0.3333 | 0.3333 | 0.6667 | 0.6667 | 0.6667 |
| Gpt-4.1-mini | 0.9923 | 0.3333 | 0.3333 | 0.3333 | 0.5000 | 0.5000 | 0.5000 |
| Gpt-4.1 | 0.9186 | 0.5000 | 0.5000 | 0.5000 | 1.0000 | 1.0000 | 1.0000 |
| Claude-3.5-Haiku | 0.9186 | 0.0000 | 0.0000 | 0.0000 | 0.0000 | 0.0000 | 0.0000 |
| Claude-3.7-Sonnet | 0.9276 | 0.3333 | 0.5000 | 0.4000 | 0.6667 | 1.0000 | 0.8000 |

Table S111: Detailed results for benchmark BIOMD0000000914 across all models.

| Model | STE ↓ | RMS (with modifiers) ↑ | | | RMS (without modifiers) ↑ | | |
|---|---|---|---|---|---|---|---|
| | | Precision | Recall | F1 | Precision | Recall | F1 |
| Gemini-2.5-Flash | 0.6481 | 0.0000 | 0.0000 | 0.0000 | 0.0000 | 0.0000 | 0.0000 |
| Gemini-2.5-Pro | 0.1487 | 0.2000 | 0.1111 | 0.1429 | 0.2000 | 0.1111 | 0.1429 |
| Gpt-4.1-mini | 0.7583 | 0.1250 | 0.1111 | 0.1176 | 0.2500 | 0.2222 | 0.2353 |
| Gpt-4.1 | 0.3154 | 0.2500 | 0.1111 | 0.1538 | 0.5000 | 0.2222 | 0.3077 |
| Claude-3.5-Haiku | 0.6166 | 0.2500 | 0.1111 | 0.1538 | 0.2500 | 0.1111 | 0.1538 |
| Claude-3.7-Sonnet | 1.0000 | 0.2000 | 0.1111 | 0.1429 | 0.2000 | 0.1111 | 0.1429 |

Table S112: Detailed results for benchmark BIOMD0000000916 across all models.

| Model | STE ↓ | RMS (with modifiers) ↑ | | | RMS (without modifiers) ↑ | | |
|---|---|---|---|---|---|---|---|
| | | Precision | Recall | F1 | Precision | Recall | F1 |
| Gemini-2.5-Flash | 0.8922 | 0.7500 | 0.7500 | 0.7500 | 0.7500 | 0.7500 | 0.7500 |
| Gemini-2.5-Pro | 0.4098 | 0.7500 | 0.7500 | 0.7500 | 0.7500 | 0.7500 | 0.7500 |
| Gpt-4.1-mini | 0.9948 | 0.5000 | 0.5000 | 0.5000 | 0.5000 | 0.5000 | 0.5000 |
| Gpt-4.1 | 0.5238 | 0.7500 | 0.7500 | 0.7500 | 0.7500 | 0.7500 | 0.7500 |
| Claude-3.5-Haiku | 0.7434 | 0.2500 | 0.2500 | 0.2500 | 0.2500 | 0.2500 | 0.2500 |
| Claude-3.7-Sonnet | 0.5633 | 0.6000 | 0.7500 | 0.6667 | 0.6000 | 0.7500 | 0.6667 |

Table S113: Detailed results for benchmark BIOMD0000000919 across all models.

| Model | STE ↓ | RMS (with modifiers) ↑ | | | RMS (without modifiers) ↑ | | |
|---|---|---|---|---|---|---|---|
| | | Precision | Recall | F1 | Precision | Recall | F1 |
| Gemini-2.5-Flash | 0.2121 | 0.0000 | 0.0000 | 0.0000 | 0.0000 | 0.0000 | 0.0000 |
| Gemini-2.5-Pro | 0.0054 | 0.7500 | 0.6000 | 0.6667 | 1.0000 | 1.0000 | 1.0000 |
| Gpt-4.1-mini | 0.4756 | 0.3333 | 0.2000 | 0.2500 | 0.6667 | 0.5000 | 0.5714 |
| Gpt-4.1 | 0.7068 | 0.6667 | 0.4000 | 0.5000 | 1.0000 | 0.7500 | 0.8571 |
| Claude-3.5-Haiku | 0.2902 | 0.0000 | 0.0000 | 0.0000 | 0.0000 | 0.0000 | 0.0000 |
| Claude-3.7-Sonnet | 0.2657 | 0.0000 | 0.0000 | 0.0000 | 0.0000 | 0.0000 | 0.0000 |

Table S114: Detailed results for benchmark BIOMD0000000920 across all models.

| Model | STE ↓ | RMS (with modifiers) ↑ | | | RMS (without modifiers) ↑ | | |
|---|---|---|---|---|---|---|---|
| | | Precision | Recall | F1 | Precision | Recall | F1 |
| Gemini-2.5-Flash | 0.5930 | 0.0000 | 0.0000 | 0.0000 | 0.0000 | 0.0000 | 0.0000 |
| Gemini-2.5-Pro | 0.3744 | 0.0000 | 0.0000 | 0.0000 | 0.0000 | 0.0000 | 0.0000 |
| Gpt-4.1-mini | 0.3782 | 0.0000 | 0.0000 | 0.0000 | 0.0000 | 0.0000 | 0.0000 |
| Gpt-4.1 | 0.2855 | 0.0000 | 0.0000 | 0.0000 | 0.0000 | 0.0000 | 0.0000 |
| Claude-3.5-Haiku | 0.3928 | 0.0000 | 0.0000 | 0.0000 | 0.6667 | 0.5000 | 0.5714 |
| Claude-3.7-Sonnet | 0.2175 | 0.0000 | 0.0000 | 0.0000 | 0.0000 | 0.0000 | 0.0000 |

Table S115: Detailed results for benchmark BIOMD0000000922 across all models.

| Model | STE ↓ | RMS (with modifiers) ↑ | | | RMS (without modifiers) ↑ | | |
|---|---|---|---|---|---|---|---|
| | | Precision | Recall | F1 | Precision | Recall | F1 |
| Gemini-2.5-Flash | 1.0000 | 0.0000 | 0.0000 | 0.0000 | 0.0000 | 0.0000 | 0.0000 |
| Gemini-2.5-Pro | 0.0316 | 0.0000 | 0.0000 | 0.0000 | 0.0000 | 0.0000 | 0.0000 |
| Gpt-4.1-mini | 0.0601 | 0.0000 | 0.0000 | 0.0000 | 0.0000 | 0.0000 | 0.0000 |
| Gpt-4.1 | 0.0601 | 0.0000 | 0.0000 | 0.0000 | 0.0000 | 0.0000 | 0.0000 |
| Claude-3.5-Haiku | 0.5956 | 0.0000 | 0.0000 | 0.0000 | 0.0000 | 0.0000 | 0.0000 |
| Claude-3.7-Sonnet | 0.0746 | 0.0000 | 0.0000 | 0.0000 | 0.0000 | 0.0000 | 0.0000 |

Table S116: Detailed results for benchmark BIOMD0000000923 across all models.

| Model | STE ↓ | RMS (with modifiers) ↑ | | | RMS (without modifiers) ↑ | | |
|---|---|---|---|---|---|---|---|
| | | Precision | Recall | F1 | Precision | Recall | F1 |
| Gemini-2.5-Flash | 0.8408 | 0.0000 | 0.0000 | 0.0000 | 0.0000 | 0.0000 | 0.0000 |
| Gemini-2.5-Pro | 0.6273 | 0.0000 | 0.0000 | 0.0000 | 0.0000 | 0.0000 | 0.0000 |
| Gpt-4.1-mini | 0.6000 | 0.0000 | 0.0000 | 0.0000 | 0.0000 | 0.0000 | 0.0000 |
| Gpt-4.1 | 0.9304 | 0.1000 | 0.2500 | 0.1429 | 0.1000 | 0.2500 | 0.1429 |
| Claude-3.5-Haiku | 0.6447 | 0.0000 | 0.0000 | 0.0000 | 0.0000 | 0.0000 | 0.0000 |
| Claude-3.7-Sonnet | 0.7257 | 0.0000 | 0.0000 | 0.0000 | 0.3333 | 0.5000 | 0.4000 |

Table S117: Detailed results for benchmark BIOMD0000000927 across all models.

| Model | STE ↓ | RMS (with modifiers) ↑ | | | RMS (without modifiers) ↑ | | |
|---|---|---|---|---|---|---|---|
| | | Precision | Recall | F1 | Precision | Recall | F1 |
| Gemini-2.5-Flash | 0.0244 | 0.3333 | 0.2000 | 0.2500 | 0.6667 | 0.4000 | 0.5000 |
| Gemini-2.5-Pro | 0.0033 | 0.0000 | 0.0000 | 0.0000 | 0.0000 | 0.0000 | 0.0000 |
| Gpt-4.1-mini | 0.2331 | 0.0000 | 0.0000 | 0.0000 | 0.0000 | 0.0000 | 0.0000 |
| Gpt-4.1 | 0.2655 | 0.6000 | 0.6000 | 0.6000 | 0.6000 | 0.6000 | 0.6000 |
| Claude-3.5-Haiku | 0.3901 | 0.0000 | 0.0000 | 0.0000 | 0.0000 | 0.0000 | 0.0000 |
| Claude-3.7-Sonnet | 0.0465 | 0.0000 | 0.0000 | 0.0000 | 0.0000 | 0.0000 | 0.0000 |

Table S118: Detailed results for benchmark BIOMD0000000932 across all models.

| Model | STE ↓ | RMS (with modifiers) ↑ | | | RMS (without modifiers) ↑ | | |
|---|---|---|---|---|---|---|---|
| | | Precision | Recall | F1 | Precision | Recall | F1 |
| Gemini-2.5-Flash | 0.5368 | 0.2000 | 0.2000 | 0.2000 | 0.4000 | 0.4000 | 0.4000 |
| Gemini-2.5-Pro | 0.5107 | 0.0000 | 0.0000 | 0.0000 | 0.0000 | 0.0000 | 0.0000 |
| Gpt-4.1-mini | 0.9231 | 0.1667 | 0.2000 | 0.1818 | 0.3333 | 0.4000 | 0.3636 |
| Gpt-4.1 | 0.5627 | 0.5000 | 0.4000 | 0.4444 | 0.5000 | 0.4000 | 0.4444 |
| Claude-3.5-Haiku | 0.7265 | 0.0000 | 0.0000 | 0.0000 | 0.0000 | 0.0000 | 0.0000 |
| Claude-3.7-Sonnet | 0.3873 | 0.4000 | 0.4000 | 0.4000 | 0.4000 | 0.4000 | 0.4000 |

Table S119: Detailed results for benchmark BIOMD0000000935 across all models.

| Model | STE ↓ | RMS (with modifiers) ↑ | | | RMS (without modifiers) ↑ | | |
|---|---|---|---|---|---|---|---|
| | | Precision | Recall | F1 | Precision | Recall | F1 |
| Gemini-2.5-Flash | 0.1418 | 0.6667 | 0.5000 | 0.5714 | 0.6667 | 0.5000 | 0.5714 |
| Gemini-2.5-Pro | 0.1098 | 0.5000 | 0.5000 | 0.5000 | 0.7500 | 0.7500 | 0.7500 |
| Gpt-4.1-mini | 0.3588 | 0.6667 | 0.5000 | 0.5714 | 0.6667 | 0.5000 | 0.5714 |
| Gpt-4.1 | 0.0974 | 0.5000 | 0.5000 | 0.5000 | 0.7500 | 0.7500 | 0.7500 |
| Claude-3.5-Haiku | 0.9961 | 0.0000 | 0.0000 | 0.0000 | 0.0000 | 0.0000 | 0.0000 |
| Claude-3.7-Sonnet | 0.1399 | 0.6667 | 0.5000 | 0.5714 | 0.6667 | 0.5000 | 0.5714 |

Table S120: Detailed results for benchmark BIOMD0000000937 across all models.

| Model | STE ↓ | RMS (with modifiers) ↑ | | | RMS (without modifiers) ↑ | | |
|---|---|---|---|---|---|---|---|
| | | Precision | Recall | F1 | Precision | Recall | F1 |
| Gemini-2.5-Flash | 0.9965 | 0.0000 | 0.0000 | 0.0000 | 0.2500 | 0.1667 | 0.2000 |
| Gemini-2.5-Pro | 0.4519 | 0.5000 | 0.3333 | 0.4000 | 0.5000 | 0.3333 | 0.4000 |
| Gpt-4.1-mini | 0.3588 | 0.0000 | 0.0000 | 0.0000 | 0.0000 | 0.0000 | 0.0000 |
| Gpt-4.1 | 0.8812 | 0.2857 | 0.3333 | 0.3077 | 0.4286 | 0.5000 | 0.4615 |
| Claude-3.5-Haiku | 0.9965 | 0.0000 | 0.0000 | 0.0000 | 0.0000 | 0.0000 | 0.0000 |
| Claude-3.7-Sonnet | 0.4872 | 1.0000 | 1.0000 | 1.0000 | 1.0000 | 1.0000 | 1.0000 |

Table S121: Detailed results for benchmark BIOMD0000000957 across all models.

| Model | STE ↓ | RMS (with modifiers) ↑ | | | RMS (without modifiers) ↑ | | |
|---|---|---|---|---|---|---|---|
| | | Precision | Recall | F1 | Precision | Recall | F1 |
| Gemini-2.5-Flash | 0.4749 | 0.5000 | 0.3333 | 0.4000 | 0.5000 | 0.3333 | 0.4000 |
| Gemini-2.5-Pro | 0.4439 | 0.7500 | 1.0000 | 0.8571 | 0.7500 | 1.0000 | 0.8571 |
| Gpt-4.1-mini | 0.7762 | 0.0000 | 0.0000 | 0.0000 | 0.0000 | 0.0000 | 0.0000 |
| Gpt-4.1 | 0.4116 | 0.5000 | 0.3333 | 0.4000 | 0.5000 | 0.3333 | 0.4000 |
| Claude-3.5-Haiku | 0.5290 | 0.0000 | 0.0000 | 0.0000 | 0.0000 | 0.0000 | 0.0000 |
| Claude-3.7-Sonnet | 0.4518 | 0.0000 | 0.0000 | 0.0000 | 0.0000 | 0.0000 | 0.0000 |

Table S122: Detailed results for benchmark BIOMD0000000966 across all models.

| Model | STE ↓ | RMS (with modifiers) ↑ | | | RMS (without modifiers) ↑ | | |
|---|---|---|---|---|---|---|---|
| | | Precision | Recall | F1 | Precision | Recall | F1 |
| Gemini-2.5-Flash | 0.4007 | 0.0000 | 0.0000 | 0.0000 | 0.0000 | 0.0000 | 0.0000 |
| Gemini-2.5-Pro | 0.1992 | 0.5000 | 0.3333 | 0.4000 | 0.5000 | 0.3333 | 0.4000 |
| Gpt-4.1-mini | 0.7892 | 0.0000 | 0.0000 | 0.0000 | 0.0000 | 0.0000 | 0.0000 |
| Gpt-4.1 | 0.3250 | 0.3333 | 0.1667 | 0.2222 | 0.3333 | 0.1667 | 0.2222 |
| Claude-3.5-Haiku | 0.6848 | 0.0000 | 0.0000 | 0.0000 | 0.0000 | 0.0000 | 0.0000 |
| Claude-3.7-Sonnet | 0.1281 | 0.0000 | 0.0000 | 0.0000 | 0.2500 | 0.1667 | 0.2000 |

Table S123: Detailed results for benchmark BIOMD0000000967 across all models.

| Model | STE ↓ | RMS (with modifiers) ↑ | | | RMS (without modifiers) ↑ | | |
|---|---|---|---|---|---|---|---|
| | | Precision | Recall | F1 | Precision | Recall | F1 |
| Gemini-2.5-Flash | 0.7606 | 0.0000 | 0.0000 | 0.0000 | 0.0000 | 0.0000 | 0.0000 |
| Gemini-2.5-Pro | 0.4548 | 0.2000 | 0.1111 | 0.1429 | 0.4000 | 0.2222 | 0.2857 |
| Gpt-4.1-mini | 0.7703 | 0.2000 | 0.1111 | 0.1429 | 0.2000 | 0.1111 | 0.1429 |
| Gpt-4.1 | 0.6499 | 0.1429 | 0.1111 | 0.1250 | 0.2857 | 0.2222 | 0.2500 |
| Claude-3.5-Haiku | 1.0000 | 0.0000 | 0.0000 | 0.0000 | 0.0000 | 0.0000 | 0.0000 |
| Claude-3.7-Sonnet | 0.6277 | 0.1667 | 0.1111 | 0.1333 | 0.3333 | 0.2222 | 0.2667 |

Table S124: Detailed results for benchmark BIOMD0000000970 across all models.

| Model | STE ↓ | RMS (with modifiers) ↑ | | | RMS (without modifiers) ↑ | | |
|---|---|---|---|---|---|---|---|
| | | Precision | Recall | F1 | Precision | Recall | F1 |
| Gemini-2.5-Flash | 0.5160 | 0.6667 | 0.6667 | 0.6667 | 1.0000 | 1.0000 | 1.0000 |
| Gemini-2.5-Pro | 0.4248 | 0.5000 | 0.6667 | 0.5714 | 0.7500 | 1.0000 | 0.8571 |
| Gpt-4.1-mini | 0.6118 | 0.0000 | 0.0000 | 0.0000 | 0.0000 | 0.0000 | 0.0000 |
| Gpt-4.1 | 0.5339 | 0.0000 | 0.0000 | 0.0000 | 0.3333 | 0.3333 | 0.3333 |
| Claude-3.5-Haiku | 0.6859 | 0.0000 | 0.0000 | 0.0000 | 0.3333 | 0.3333 | 0.3333 |
| Claude-3.7-Sonnet | 0.4909 | 0.4000 | 0.6667 | 0.5000 | 0.6000 | 1.0000 | 0.7500 |

Table S125: Detailed results for benchmark BIOMD0000001013 across all models.

| Model | STE ↓ | RMS (with modifiers) ↑ | | | RMS (without modifiers) ↑ | | |
|---|---|---|---|---|---|---|---|
| | | Precision | Recall | F1 | Precision | Recall | F1 |
| Gemini-2.5-Flash | 0.4926 | 0.0000 | 0.0000 | 0.0000 | 0.5000 | 0.2500 | 0.3333 |
| Gemini-2.5-Pro | 0.5010 | 0.2500 | 0.2500 | 0.2500 | 0.5000 | 0.5000 | 0.5000 |
| Gpt-4.1-mini | 0.9569 | 0.0000 | 0.0000 | 0.0000 | 0.5000 | 0.2500 | 0.3333 |
| Gpt-4.1 | 0.5601 | 0.3333 | 0.2500 | 0.2857 | 0.6667 | 0.5000 | 0.5714 |
| Claude-3.5-Haiku | 0.6273 | 0.0000 | 0.0000 | 0.0000 | 0.0000 | 0.0000 | 0.0000 |
| Claude-3.7-Sonnet | 0.2703 | 0.0000 | 0.0000 | 0.0000 | 0.3333 | 0.2500 | 0.2857 |

Table S126: Detailed results for benchmark BIOMD0000001016 across all models.

| Model | STE ↓ | RMS (with modifiers) ↑ | | | RMS (without modifiers) ↑ | | |
|---|---|---|---|---|---|---|---|
| | | Precision | Recall | F1 | Precision | Recall | F1 |
| Gemini-2.5-Flash | 0.3622 | 0.0000 | 0.0000 | 0.0000 | 0.0000 | 0.0000 | 0.0000 |
| Gemini-2.5-Pro | 0.3616 | 0.0000 | 0.0000 | 0.0000 | 0.2500 | 0.1111 | 0.1538 |
| Gpt-4.1-mini | 0.8454 | 0.3333 | 0.2222 | 0.2667 | 0.3333 | 0.2222 | 0.2667 |
| Gpt-4.1 | 0.6198 | 0.0000 | 0.0000 | 0.0000 | 0.0000 | 0.0000 | 0.0000 |
| Claude-3.5-Haiku | 0.6599 | 0.0000 | 0.0000 | 0.0000 | 0.0000 | 0.0000 | 0.0000 |
| Claude-3.7-Sonnet | 0.7653 | 0.0000 | 0.0000 | 0.0000 | 0.1667 | 0.1111 | 0.1333 |

Table S127: Detailed results for benchmark BIOMD0000001022 across all models.

| Model | STE ↓ | RMS (with modifiers) ↑ | | | RMS (without modifiers) ↑ | | |
|---|---|---|---|---|---|---|---|
| | | Precision | Recall | F1 | Precision | Recall | F1 |
| Gemini-2.5-Flash | 0.1473 | 0.0000 | 0.0000 | 0.0000 | 0.0000 | 0.0000 | 0.0000 |
| Gemini-2.5-Pro | 0.7169 | 0.0000 | 0.0000 | 0.0000 | 0.0000 | 0.0000 | 0.0000 |
| Gpt-4.1-mini | 0.7150 | 0.0000 | 0.0000 | 0.0000 | 0.0000 | 0.0000 | 0.0000 |
| Gpt-4.1 | 0.8688 | 0.0000 | 0.0000 | 0.0000 | 0.0000 | 0.0000 | 0.0000 |
| Claude-3.5-Haiku | 0.8103 | 0.0000 | 0.0000 | 0.0000 | 0.0000 | 0.0000 | 0.0000 |
| Claude-3.7-Sonnet | 0.8953 | 0.0000 | 0.0000 | 0.0000 | 0.3333 | 0.1667 | 0.2222 |

Table S128: Detailed results for benchmark BIOMD0000001023 across all models.

| Model | STE ↓ | RMS (with modifiers) ↑ | | | RMS (without modifiers) ↑ | | |
|---|---|---|---|---|---|---|---|
| | | Precision | Recall | F1 | Precision | Recall | F1 |
| Gemini-2.5-Flash | 0.5806 | 0.0000 | 0.0000 | 0.0000 | 0.5000 | 0.1667 | 0.2500 |
| Gemini-2.5-Pro | 0.3232 | 0.0000 | 0.0000 | 0.0000 | 0.3333 | 0.1667 | 0.2222 |
| Gpt-4.1-mini | 0.4891 | 0.0000 | 0.0000 | 0.0000 | 0.3333 | 0.1667 | 0.2222 |
| Gpt-4.1 | 0.4076 | 0.0000 | 0.0000 | 0.0000 | 0.7500 | 0.5000 | 0.6000 |
| Claude-3.5-Haiku | 0.5033 | 0.0000 | 0.0000 | 0.0000 | 0.0000 | 0.0000 | 0.0000 |
| Claude-3.7-Sonnet | 0.3154 | 0.0000 | 0.0000 | 0.0000 | 0.4000 | 0.3333 | 0.3636 |

Table S129: Detailed results for benchmark BIOMD0000001024 across all models.

| Model | STE ↓ | RMS (with modifiers) ↑ | | | RMS (without modifiers) ↑ | | |
|---|---|---|---|---|---|---|---|
| | | Precision | Recall | F1 | Precision | Recall | F1 |
| Gemini-2.5-Flash | 0.8279 | 0.3333 | 0.2500 | 0.2857 | 0.3333 | 0.2500 | 0.2857 |
| Gemini-2.5-Pro | 0.9429 | 0.0000 | 0.0000 | 0.0000 | 0.3333 | 0.2500 | 0.2857 |
| Gpt-4.1-mini | 0.8474 | 0.2500 | 0.2500 | 0.2500 | 0.7500 | 0.7500 | 0.7500 |
| Gpt-4.1 | 0.9378 | 0.0000 | 0.0000 | 0.0000 | 0.0000 | 0.0000 | 0.0000 |
| Claude-3.5-Haiku | 0.8567 | 0.0000 | 0.0000 | 0.0000 | 0.0000 | 0.0000 | 0.0000 |
| Claude-3.7-Sonnet | 0.8813 | 0.0000 | 0.0000 | 0.0000 | 0.3333 | 0.2500 | 0.2857 |

Table S130: Detailed results for benchmark BIOMD0000001031 across all models.

| Model | STE ↓ | RMS (with modifiers) ↑ | | | RMS (without modifiers) ↑ | | |
|---|---|---|---|---|---|---|---|
| | | Precision | Recall | F1 | Precision | Recall | F1 |
| Gemini-2.5-Flash | 0.8606 | 0.0000 | 0.0000 | 0.0000 | 0.0000 | 0.0000 | 0.0000 |
| Gemini-2.5-Pro | 0.8863 | 0.6000 | 0.7500 | 0.6667 | 0.8000 | 1.0000 | 0.8889 |
| Gpt-4.1-mini | 0.8396 | 0.3333 | 0.2500 | 0.2857 | 0.3333 | 0.2500 | 0.2857 |
| Gpt-4.1 | 0.9275 | 0.2000 | 0.2500 | 0.2222 | 0.4000 | 0.5000 | 0.4444 |
| Claude-3.5-Haiku | 0.9390 | 1.0000 | 0.2500 | 0.4000 | 1.0000 | 0.2500 | 0.4000 |
| Claude-3.7-Sonnet | 0.8417 | 0.3333 | 0.2500 | 0.2857 | 0.6667 | 0.5000 | 0.5714 |

Table S131: Detailed results for benchmark BIOMD0000001035 across all models.

| Model | STE ↓ | RMS (with modifiers) ↑ | | | RMS (without modifiers) ↑ | | |
|---|---|---|---|---|---|---|---|
| | | Precision | Recall | F1 | Precision | Recall | F1 |
| Gemini-2.5-Flash | 0.8342 | 0.0000 | 0.0000 | 0.0000 | 0.0000 | 0.0000 | 0.0000 |
| Gemini-2.5-Pro | 0.1295 | 0.0000 | 0.0000 | 0.0000 | 0.0000 | 0.0000 | 0.0000 |
| Gpt-4.1-mini | 0.9652 | 0.2500 | 0.1250 | 0.1667 | 0.2500 | 0.1250 | 0.1667 |
| Gpt-4.1 | 0.7968 | 0.0000 | 0.0000 | 0.0000 | 0.0000 | 0.0000 | 0.0000 |
| Claude-3.5-Haiku | 0.5595 | 0.0000 | 0.0000 | 0.0000 | 0.0000 | 0.0000 | 0.0000 |
| Claude-3.7-Sonnet | 0.4633 | 0.0000 | 0.0000 | 0.0000 | 0.0000 | 0.0000 | 0.0000 |

Table S132: Detailed results for benchmark BIOMD0000001036 across all models.

| Model | STE ↓ | RMS (with modifiers) ↑ | | | RMS (without modifiers) ↑ | | |
|---|---|---|---|---|---|---|---|
| | | Precision | Recall | F1 | Precision | Recall | F1 |
| Gemini-2.5-Flash | 0.9579 | 0.0000 | 0.0000 | 0.0000 | 0.0000 | 0.0000 | 0.0000 |
| Gemini-2.5-Pro | 0.2877 | 0.5000 | 0.5000 | 0.5000 | 0.6667 | 0.6667 | 0.6667 |
| Gpt-4.1-mini | 0.9116 | 0.0000 | 0.0000 | 0.0000 | 0.0000 | 0.0000 | 0.0000 |
| Gpt-4.1 | 0.5812 | 0.2727 | 0.5000 | 0.3529 | 0.3636 | 0.6667 | 0.4706 |
| Claude-3.5-Haiku | 0.9709 | 0.0000 | 0.0000 | 0.0000 | 0.0000 | 0.0000 | 0.0000 |
| Claude-3.7-Sonnet | 0.4023 | 0.5000 | 0.5000 | 0.5000 | 0.6667 | 0.6667 | 0.6667 |

Table S133: Detailed results for benchmark BIOMD0000001037 across all models.

| Model | STE ↓ | RMS (with modifiers) ↑ | | | RMS (without modifiers) ↑ | | |
| --- | --- | --- | --- | --- | --- | --- | --- |
| | | Precision | Recall | F1 | Precision | Recall | F1 |
| Gemini-2.5-Flash | 0.4483 | 0.0000 | 0.0000 | 0.0000 | 0.0000 | 0.0000 | 0.0000 |
| Gemini-2.5-Pro | 0.1832 | 0.0000 | 0.0000 | 0.0000 | 0.3333 | 0.3333 | 0.3333 |
| Gpt-4.1-mini | 0.5196 | 0.0000 | 0.0000 | 0.0000 | 0.0000 | 0.0000 | 0.0000 |
| Gpt-4.1 | 0.5216 | 0.0000 | 0.0000 | 0.0000 | 0.0000 | 0.0000 | 0.0000 |
| Claude-3.5-Haiku | 0.5230 | 0.0000 | 0.0000 | 0.0000 | 0.0000 | 0.0000 | 0.0000 |
| Claude-3.7-Sonnet | 0.8787 | 0.0000 | 0.0000 | 0.0000 | 0.0000 | 0.0000 | 0.0000 |

Table S134: Detailed results for benchmark BIOMD0000001038 across all models.

| Model | STE ↓ | RMS (with modifiers) ↑ | | | RMS (without modifiers) ↑ | | |
| --- | --- | --- | --- | --- | --- | --- | --- |
| | | Precision | Recall | F1 | Precision | Recall | F1 |
| Gemini-2.5-Flash | 0.2453 | 0.0000 | 0.0000 | 0.0000 | 0.0000 | 0.0000 | 0.0000 |
| Gemini-2.5-Pro | 0.2790 | 0.0000 | 0.0000 | 0.0000 | 0.2000 | 0.1667 | 0.1818 |
| Gpt-4.1-mini | 0.7061 | 0.0000 | 0.0000 | 0.0000 | 0.2500 | 0.1667 | 0.2000 |
| Gpt-4.1 | 0.9284 | 0.2500 | 0.1667 | 0.2000 | 0.2500 | 0.1667 | 0.2000 |
| Claude-3.5-Haiku | 0.4480 | 0.0000 | 0.0000 | 0.0000 | 0.0000 | 0.0000 | 0.0000 |
| Claude-3.7-Sonnet | 0.5256 | 0.0000 | 0.0000 | 0.0000 | 0.0000 | 0.0000 | 0.0000 |

Table S135: Detailed results for benchmark BIOMD0000001045 across all models.

| Model | STE ↓ | RMS (with modifiers) ↑ | | | RMS (without modifiers) ↑ | | |
| --- | --- | --- | --- | --- | --- | --- | --- |
| | | Precision | Recall | F1 | Precision | Recall | F1 |
| Gemini-2.5-Flash | 0.4787 | 1.0000 | 1.0000 | 1.0000 | 1.0000 | 1.0000 | 1.0000 |
| Gemini-2.5-Pro | 0.4427 | 1.0000 | 1.0000 | 1.0000 | 1.0000 | 1.0000 | 1.0000 |
| Gpt-4.1-mini | 0.7036 | 1.0000 | 1.0000 | 1.0000 | 1.0000 | 1.0000 | 1.0000 |
| Gpt-4.1 | 0.4207 | 1.0000 | 1.0000 | 1.0000 | 1.0000 | 1.0000 | 1.0000 |
| Claude-3.5-Haiku | 0.9855 | 0.6667 | 1.0000 | 0.8000 | 0.6667 | 1.0000 | 0.8000 |
| Claude-3.7-Sonnet | 0.5421 | 0.3333 | 0.5000 | 0.4000 | 0.5000 | 0.5000 | 0.5000 |

Table S136: Detailed results for benchmark BIOMD0000001048 across all models.

| Model | STE ↓ | RMS (with modifiers) ↑ | | | RMS (without modifiers) ↑ | | |
| --- | --- | --- | --- | --- | --- | --- | --- |
| | | Precision | Recall | F1 | Precision | Recall | F1 |
| Gemini-2.5-Flash | 0.1173 | 0.2500 | 0.1667 | 0.2000 | 0.2500 | 0.2500 | 0.2500 |
| Gemini-2.5-Pro | 0.0913 | 0.3333 | 0.1667 | 0.2222 | 0.3333 | 0.2500 | 0.2857 |
| Gpt-4.1-mini | 0.3691 | 0.2500 | 0.1667 | 0.2000 | 0.2500 | 0.2500 | 0.2500 |
| Gpt-4.1 | 0.7415 | 0.3333 | 0.1667 | 0.2222 | 0.3333 | 0.2500 | 0.2857 |
| Claude-3.5-Haiku | 0.5294 | 0.0000 | 0.0000 | 0.0000 | 0.0000 | 0.0000 | 0.0000 |
| Claude-3.7-Sonnet | 0.3396 | 0.1667 | 0.1667 | 0.1667 | 0.1667 | 0.2500 | 0.2000 |

Table S137: Detailed results for benchmark BIOMD0000001057 across all models.

| Model | STE ↓ | RMS (with modifiers) ↑ | | | RMS (without modifiers) ↑ | | |
|---|---|---|---|---|---|---|---|
| | | Precision | Recall | F1 | Precision | Recall | F1 |
| Gemini-2.5-Flash | 0.0068 | 0.6000 | 0.6000 | 0.6000 | 0.8000 | 1.0000 | 0.8889 |
| Gemini-2.5-Pro | 0.0068 | 0.7500 | 0.6000 | 0.6667 | 1.0000 | 1.0000 | 1.0000 |
| Gpt-4.1-mini | 0.2356 | 0.5000 | 0.4000 | 0.4444 | 0.7500 | 0.7500 | 0.7500 |
| Gpt-4.1 | 0.0017 | 0.7500 | 0.6000 | 0.6667 | 1.0000 | 1.0000 | 1.0000 |
| Claude-3.5-Haiku | 0.9990 | 0.0000 | 0.0000 | 0.0000 | 0.0000 | 0.0000 | 0.0000 |
| Claude-3.7-Sonnet | 0.0250 | 0.7500 | 0.6000 | 0.6667 | 1.0000 | 1.0000 | 1.0000 |

