# OpenReview forum: "Measuring Scientific Capabilities of Language Models with a Systems Biology Dry Lab"
_NeurIPS.cc/2025/Datasets_and_Benchmarks_Track — NeurIPS 2025 Datasets and Benchmarks Track poster_

### Official Review · Reviewer_pjT8 · 2025-07-01

**Rating:** 4
**Confidence:** 4

**Summary:**

The paper introduces SciGym, a benchmark system for experimentation capabilities of LLMs using biological systems dynamics modeled with ODEs. The benchmark provides a collection of ODE systems, and the necessary infrastructure to run them. A panel of LLMs are evaluated in an agentic framework that allows the LLMs to iteratively experiment, perturbing the system and acquiring data.

The idea of using ODE models to simulate biological discovery with perturbations is very good. However, the way the interaction of the LLM is done very narrowly sticks to the formal ODE framework, leaving it unclear how much biological research is really happening in this benchmark, and narrowing the scope. See Limitations section.

**Additional Feedback:**

The use of mathematical notation in section 2 for system components is very heavy, and in my opinion makes enzymatic ODE systems appear more complicated than they really are to newcomers to this field that haven't encountered them before in a biochemistry context.

**Dataset Code Accessibility:**

Partly

**Dataset Code Comments:**

Given that the benchmark inherently requires a lot of code and engineering, rather than just a dataset, the documentation provided on Github seems to be very limited. The high level organization of the repo is not described, and e.g. the "Setup an agent" section tells me how to run an existing agent, rather than the actually most relevant thing, how to quickly onboard and evaluate a new LLM. It would also be a good idea to document how to add and run new ODE systems, and how to potentially, if possible, modulate the difficulty as is touched upon in the manuscript.

**Ethical Considerations:**

No, there are no or only very minor ethics concerns

**Final Justification:**

The authors provided a key experiment in the rebuttal that supports that the benchmark is actually evaluating biological reasoning. Given the rebuttal rules i'll have to take their word that the code quality will be improved for release.

**Limitations Weaknesses:**

1) The way the benchmark is laid out, the biological discovery process is abstracted away to a high degree. It is unclear to me whether the benchmark requires the models to reason about systems biologically, like a biologist would do (considering ways molecules can interact, extensive background knowledge on common patterns and mechanisms, known gene pathways etc), or whether this ends up just being reasoning over ODE systems that could be arbitrary.
2) It is unclear to me to what degree a model's scientific discovery aptitude is conflated with its ability to write useful code, and mastering the SBML language and idiosyncrasies that can be considered very niche and may not be represented well in pre-training corpuses.
3) Though being familiar with perturbation biology and ODE models to some degree (not in an ML context), the paper in my view doesn't do a great job right now at communicating how *exactly* the tasks are defined. From Figure 3, I take that missing reaction components need to be discovered. Later on, it is mentioned that the agent is allowed to conduct perturbations. How can the model perturb a reaction component it has not discovered yet?

**Strengths Contributions:**

The concept of using validated ODE models of biological systems to simulate scientific discovery processes is highly interesting and novel. Data is well curated, and a codebase for interacting with ODE systems is provided.

---

> ### Author Rebuttal · Authors · 2025-07-30
>
> # Q1: Whether the benchmark requires the models to reason about systems biologically
>
> Thank you for raising this important question about whether SciGym tests biology reasoning. Below, we first provide a summary of our response, then detail our experiments and discussions.
>
>
> **Summary:** SciGym uses models studied by actual biologists to represent real biological processes, so reasoning like a biologist would be essential to recover these models in SciGym. We designed a controlled experiment to test whether SciGym requires biological knowledge by removing domain context while keeping identical mathematical structure. Results show significant performance drops when biological reasoning is removed, confirming our hypothesis. Furthermore, SciGym explicitly tests the core reasoning skills required for scientists: hypothesis formation, experiment design and data analysis. These skills are essential for scientific discovery and go beyond what's required for standard ODE discovery tasks.
>
>
>
> ## Experiments: Biology reasoning is essential in SciGym
>
> ### Experiment Design
>
> If our benchmark only tests general ODE discovery, then domain context shouldn't matter, which means that models should perform equally well regardless of how we describe the task. To test this, we conducted a controlled experiment where we kept the exact same underlying SBML mathematical system but only changed the context in the system prompt. By describing the studied system as other subjects (such as Epidemiology and Ecology) instead of molecular biology, we prevented the model from accessing its molecular biology knowledge when reasoning. If biological reasoning doesn't matter, performance should be identical across all framings since the mathematical structure remains unchanged.
>
> ### Experiment results
>
> We tested this hypothesis using Gemini-2.5-Flash on SciGym-small:
>
> | Domain Framing | STE | RMS (w/ modifier) | RMS (w/o modifier) |
> | -------------- | --- | ----------------- | ------------------ |
> | Biology (Original) | 0.4181 | 0.1217 | 0.2005 |
> | Ecology | 0.4382** | 0.0623** | 0.1023** |
> | Epidemiology | 0.4417** | 0.0761** | 0.1241** |
>
> ### Key findings
>
> The result above shows that when the agent does not use biology reasoning (row 2 and 3), their performance drops significantly (p < 0.05 across all metrics). This confirms our hypothesis that biology reasoning is essential in our benchmark.
>
> ## Discussions
>
> **Biology and scientific priors are essential for efficient explorations.** As we have mentioned, SBML models in SciGym are designed by the systems biologists to represent the real biology process. While it might theoretically be possible to discover these systems through brute-force mathematical exploration, our benchmark limits agents to 20 interaction rounds. In order to succeed within a limited interaction round, we think agents must leverage biological priors like mass action kinetics, common regulatory patterns, and systematic perturbation strategies to efficiently navigate the vast space of hypotheses. Without this biological knowledge, agents would likely exhaust their interaction budget on implausible hypotheses, failing to discover the correct mechanisms in time.
>
> **Most importantly, SciGym test skills essential for scientific discovery.** The skills tested in SciGym are essential for scientific discovery and go beyond what's required for standard ODE discovery tasks: by requiring an agent to write code, make choices on how to analyze data, and select which experiments to execute, and in what order.
>
> ---
>
> # Q2: Scientific discovery is conflated with its ability to write code and mastering SBML
>
> Thank you for raising this important point about the confounding effects with coding and SBML skills. Below, we first provide a summary of our response, then detail our experiments and discussions.
>
> **Summary:** End-to-end scientific discovery inherently requires reasoning, engineering, and domain-specific skills to work in concert. We agree that coding is essential for success in our benchmark, but this reflects the reality of modern scientific research. Scientists routinely need to write code for data analysis, implement statistical tests, and use domain-specific software tools. Regarding SBML knowledge, we designed new experiments demonstrating that frontier models possess similar and sufficient SBML knowledge for SciGym. Therefore, performance bottlenecks stem primarily from limitations in scientific reasoning rather than technical knowledge of SBML.
>
> ## Experiments: Models have sufficient SBML knowledge for SciGym
>
> ### Experiment Motivation
>
> Knowing the syntax of SBML is a requirement to succeed in SciGym, but this challenge exists in virtually every benchmark. It's impossible to completely isolate abstract capabilities from concrete syntactic or domain-specific knowledge. **Moreover, we designed SciGym to minimize the SBML technical burden.** We filtered out advanced SBML features and retained only basic, standardized components (removing complex elements like `<events>` and `<rules>`). Our task primarily requires knowing how to add reactions, a simple operation that we also demonstrate with examples in the system prompt.
>
> ### Experiment Design
>
> We designed a simple zero-shot task to test whether the evaluated LLMs have the necessary SBML knowledge for SciGym (i.e., knowing how to add reactions using libsbml). In this experiment, the agent is given an incomplete SBML model and its corresponding complete SBML model. The agent is tasked to write a Python script using the libsbml package to add missing components to the incomplete model such that it matches the complete model. This set-up is similar to the one in scigym, except the agent is told what reactions are missing.
> Upon successful execution of the agent’s script, we extract the modified incomplete SBML model and systematically evaluate its equivalence to the complete version. We consider two models equivalent if they contain the exact same set of species, reactions, kinetic laws, parameters, values, and initial concentrations (this equivalence is sufficient to obtain perfect scores on the *NTS, RMS, STE* metrics). We randomly picked 50 models from the small SciGym split and evaluated the same six models in our paper.
>
> ### Experiment Results
>
> | Model                        | % Equiv | Avg. Iter. | Fail. Mode(s)                |
> |------------------------------|---------|------------|------------------------------|
> | claude-3-5-haiku-20241022    | 80      | 2.04       | wrong ID (2), equation (8)   |
> | claude-3-7-sonnet-20250219   | 100     | 1.16       | N/A                          |
> | gemini-2.5-flash-preview-05-20 | 98    | 1.16       | failed (1)                   |
> | gemini-2.5-pro-preview-03-25 | 100     | 1.71       | N/A                          |
> | gpt-4.1-2025-04-14           | 64      | 2.17       | wrong ID (9), failed (9)     |
> | gpt-4.1-mini-2025-04-14      | 66      | 2.44       | wrong eq. (1), failed (16)   |
> | o3-2025-04-16                | 96      | 1.47       | wrong ID (1), failed (1)     |
> | o4-mini-2025-04-16           | 44      | 2.73       | failed (28)                  |
>
>
> Notes on the failure modes:
>
> - Wrong ID → There was a mismatch in the IDs of at least one component
>
> - Wrong equation → There was a mismatch in the kinetic law equations
>
> - Failed → LLM submitted more than 4 responses that failed to parse or run
>
> ### Key Findings
>
> - From the results, **we first observe that most models possess sufficient SBML knowledge to succeed in our task**. If models lacked SBML knowledge, they should achieve near-zero success rates. Given that we used the same incomplete SBML models as those in our SciGym benchmark, these models demonstrate they can effectively manipulate SBML via libsbml for SciGym tasks.
>
> - **More importantly, all frontier models (Sonnet, Gemini, o3) achieve near-perfect performance on this task.** This implies that the performance bottleneck for frontier LLMs on SciGym stems primarily from scientific discovery aptitude rather than coding ability or SBML language mastery.
>
> ---
>
> # Q3: Not good at communicating at how the tasks are defined
>
> This appears to be a misunderstanding about what perturbations are allowed in our benchmark. The perturbations in our task refer only to changing initial concentrations of existing species, not manipulating reaction components (see Figure 4).
>
> We will improve the Figure 3 and 4 to help the readers better understand the design of our benchmark:
> - Fig 3: explicitly list what information the agent has access to (list of species in the system) and what parts of the reference system are hidden and must be discovered to complete the task (reactions, the relationships between species)
> - Fig 4: added legend which matches species shown in fig. 3 to illustrate how the agent may perturb concentrations of species.
>
> ---
>
> # Q4: Improve the code
>
> Thank you for your suggestions about the code! **We have refactored the codebase to reflect the following changes:**
>
> - We refactored the codebase to make it easier to evaluate new agents by requiring only the implementation of one abstract function: get_response(self, user_message: str) -> str.
>
> - We added detailed documentations about the SciGym task curation process and instructions for generating new SBML tasks.
>
> - We improved the configurability of the filtering and generation scripts to enable the creation of tasks with different difficulty levels. We also organized all processing scripts under the scigym/make folder.
>
> - The interface for adding more perturbations is difficult to modularize from a coding perspective because the space of possible perturbations can be quite flexible. Instead, we have updated our documentation to provide general guidelines on the files that need to be edited to support new perturbations.
>
> Since we are not permitted to submit or modify the codebase during the rebuttal period, we will publish these updates if the paper is accepted.

---

> > ### Comment · Reviewer_pjT8 · 2025-08-01
> >
> > Thank you for the additional experiments, the biological reasoning ablation is very elegant and convincing. I see an opportunity to highlight example trajectories in the appendix to directly highlight that this kind of reasoning is actually happening.
> >
> > The SBML experiment indeed confirms that frontier LLMs have a good handle on the language - but also that non-frontier LLMs (+o4, interestingly) can struggle on the formal language level already, kind of rendering it a benchmark that only becomes truly relevant once an LLM is able to pass this "first gate".

---

> > > ### Author Response · Authors · 2025-08-01
> > >
> > > Thank you for the positive comments and thoughtful feedback. We really appreciate it!

---

### Official Review · Reviewer_PzAn · 2025-07-01

**Rating:** 5
**Confidence:** 3

**Summary:**

The paper presents SciGym, a benchmark that measures an LLM agent’s ability to design experiments and infer mechanisms in biological systems encoded as Systems Biology Markup Language (SBML) models. Agents simulate a close-loop scientific discovery process by proposing perturbations, receiving simulated data, analyzing results with code, and iterating until they come up with a complete model. Three metrics (NTS, RMS, STE) are introduced to evaluate how well an agent recovers the underlying SBML model. 350 SBML models are included in this benchmark, although only 137 small SBML models are evaluated due to computational resource constraints. Experiments with six state-of-the-art LLMs on the 137 small SBML models show that larger models (1) perform better but still struggle as reaction networks grow, (2) overfit to experimental data and do not generalize well to unseen conditions and (3) rarely perform well on identifying modifier species.

**Dataset Code Accessibility:**

Yes

**Dataset Code Comments:**

I went through the github and huggingface repo. The code and data appear to be complete.

**Ethical Considerations:**

No, there are no or only very minor ethics concerns

**Final Justification:**

After reading the authors' rebuttal, I believe the two weaknesses I raised are not as serious as I initially thought. As the authors have adequately addressed my concerns, I will raise my rating to 5 (Accept).

**Limitations Weaknesses:**

1. A large portion of this benchmark (213 large SBML models with >10 reactions) remains unevaluated. As this paper does not introduce new SBML data (all data are curated from previous work BioModels), the absence of evaluation makes the contribution of large SBML models in this benchmark unclear. While the authors note that computational resource constraints, a detailed discussion and some preliminary results on large SBML models could help further clarify their contribution.
2. As mentioned in Sec. 5, SciGym restricts agents to setting initial concentrations (and a single species knockout in an appendix ablation). By doing so, it excludes many real experimental manipulations that biologists routinely use, such as time-varying stimuli or environmental changes. This narrow interface may encourage LLMs to find “shortcut” strategies rather than engage in richer exploration, as evidenced by this paper’s finding that LLMs tend to overfit to experimental data and do not generalize well.

**Strengths Contributions:**

1. The paper is well-structured and clearly written. Figures and tables are informative, properly labeled, and accompanied by concise captions that summarize key takeaways. Sec. 3 and the appendices provide clear description on the framework, evaluation metrics and benchmark curation process,
2. SciGym turns SBML models into an automated experimental sandbox, enabling AI agents to perform open-ended biological discovery research using experimental perturbations, which is not included by previous AI-scientist benchmarks.
3. SciGym is highly extensible. With the curation pipeline it provided, researchers can incorporate more SBML models and/or experimental configurations to continually expand and strengthen the benchmark.

---

> ### Author Rebuttal · Authors · 2025-07-30
>
> Thank you for your encouraging comments about our work! Below we provide some clarifications to address the questions raised in the weakness sections.
>
> ---
>
> # Q1: SciGym only evaluated small systems
>
> Thank you for this important question!
>
> **Frontier models already struggle significantly with the current complexity in SciGym.** As Figure 6 in the main paper shows, all models' predictive simulation error increases substantially as the number of reactions increases. We also conducted additional analysis on reaction matching accuracy (RMS) and found that all models achieve near-zero RMS for 9-reaction systems. Even the best-performing model, Gemini-pro, achieves only 0.0231 RMS (with modifier) on these moderately complex systems. Further increasing system complexity would likely result in uniformly poor performance across all models, making meaningful comparisons difficult. Therefore, we focused our computational resources on smaller systems where we can meaningfully differentiate between approaches and identify specific reasoning patterns.
>
>
> However, we think the large systems serve as important benchmarks for future AI capabilities. As scientific reasoning capabilities advance, these larger and more challenging systems will become increasingly valuable for evaluation.
>
> ---
>
> # Q2: SciGym restricts agents to setting initial concentrations
>
> Thank you for raising this question!
>
> Changing initial concentrations is a **fundamental experimental approach** across many fields of biology. For example, drug dosage studies systematically vary initial drug concentrations to establish dose-response relationships and metabolic studies alter substrate concentrations to understand pathway flux. While we could incorporate more specialized manipulations like time-varying stimuli via SBML events, we intentionally designed this initial version as **a general biology benchmark that works across diverse biological domains rather than optimizing for any single field.** Future work can develop domain-specific variants that include richer perturbation interfaces tailored to particular biological contexts.
>
> We do not think initial concentration changes provide shortcuts that bypass meaningful reasoning. On the contrary, the space of possible concentration combinations is vast, and **Carefully designed perturbations on the initial concentrations provide quite meaningful information.** For example, if changing the initial concentration of a species affects reaction speed but the species itself doesn't change, this indicates it's likely a modifier (catalyst or inhibitor) in the reaction, requiring the model to understand regulatory mechanisms. Similarly, systematically perturbing initial concentrations allows models to determine the functional forms of kinetic laws (whether reactions follow mass action, Michaelis-Menten, or other kinetics) by analyzing how reaction rates respond to concentration changes. These insights demand genuine scientific reasoning rather than simple pattern matching.

---

> > ### Comment · Reviewer_PzAn · 2025-08-02
> >
> > Thank you for your thoughtful response! After reading the authors' rebuttal, I believe the two weaknesses I raised are not as serious as I initially thought. As the authors have adequately addressed my concerns, I will raise my rating to 5 (Accept).

---

> > > ### Author Response · Authors · 2025-08-03
> > >
> > > Thank you for reconsidering our responses and updating your rating. We appreciate your thoughtful feedback.

---

### Official Review · Reviewer_6BtY · 2025-07-03

**Rating:** 5
**Confidence:** 4

**Summary:**

This paper introduces SciGym, a benchmark designed to evaluate the scientific reasoning capabilities of large language models (LLMs) through iterative experiment design and analysis tasks in simulated biology environments. Instead of relying on costly wet-lab experiments, SciGym leverages systems biology models encoded in SBML to create a “dry lab” setting, enabling efficient simulation of complex biological systems. The benchmark tests LLMs’ ability to uncover system mechanisms by performing interventions and analyzing outcomes, mimicking real scientific discovery workflows. Results from evaluating six state-of-the-art LLMs reveal that while more capable models outperform smaller ones, all models struggle as system complexity increases and particularly with capturing subtle modifier relationships.

**Dataset Code Accessibility:**

Yes

**Ethical Considerations:**

No, there are no or only very minor ethics concerns

**Final Justification:**

After the author's rebuttal, I think my original concern about this paper's generalizability to other disciplines/domains has been largely addressed. Thus, I raised my rating to 5 (originally 4).

**Limitations Weaknesses:**

As a first step towards benchmarking LLMs on end-to-end scientific discovery, this paper is valuable for providing the dataset and the empirical results. However, many aspects of this paper seem to be too specifically-designed for biology (for example, the SBML language, the perturbation strategy, and even the evaluation metrics). Thus, it might be too difficult or even impossible to adopt this work's pipeline onto other disciplines such as chemistry, geography, etc.

**Strengths Contributions:**

1. The paper presents SciGym, a first-of-its-kind benchmark that assesses LLMs’ scientific reasoning through open-ended, iterative experimentation and data analysis tasks. The authors creatively propose to leverage SBML models from the BioModels repository as dry lab simulators, making it possible to evaluate the whole lifecycle of scientific research.

2. The dataset was thoroughly curated from 1096 BioModels in the BioModels repository. The preprocessing/filtering steps, framework and evaluation metrics are carefully designed and well-demonstrated.

3. Evaluation on six state-of-the-art LLMs reveals that more capable models outperform smaller ones, but all show sharp performance drops with system complexity and struggle to recover modifier relationships. These findings are well-illustrated and clearly discussed, offering actionable insights into current limitations in LLM-based scientific reasoning.

---

> ### Author Rebuttal · Authors · 2025-07-30
>
> Thank you for your encouraging comments about our work! Below we provide some clarifications to address the questions raised in the weakness sections.
>
> ---
>
> # Q1: Extend SciGym to other disciplines
>
> Thanks for this great question!
>
> **SBML can describe processes well beyond molecular biology.** Although the majority of models in our benchmark come from biochemistry literature, SBML can describe dynamic systems with different contexts. In fact, our benchmark already spans multiple disciplines: we have chemistry reactions, gene regulation networks, and epidemiological models. One example in the BioModel repo (BIOMD0000000969) describes spatio-temporal COVID transmission dynamics and was published in Health & Place, a health geography journal.
>
> By drawing models from the most popular and widely used model repository, we expect that SciGym can improve over time with respect to coverage of a variety of disciplines. As the BioModels database continues to grow, our framework will flexibly allow models from different experimental modalities to be incorporated into an evaluation pipeline. SBML’s origins as an exchange format with high extensibility makes it promising that models originating from other disciplines could continue to be added to our benchmark.
>
> **We hope our framework will inspire more simulation-based AI scientist benchmarks in other domains.** The core approach of using computational experiments to discover system structure could be adapted to chemistry, ecology, economics, or any field where scientists study dynamical systems. We'd be excited to see extensions to other scientific disciplines.

---

> > ### Comment · Area_Chair_XwBx · 2025-08-06
> >
> > Reviewer 6BtY - can you please check the rebuttal and respond?

---

### Official Review · Reviewer_vZ9f · 2025-07-03

**Rating:** 5
**Confidence:** 3

**Summary:**

This paper introduces SciGym, a benchmark for evaluation of LLMs in scientific discovery pipelines, specifically in design of experiments to perturb mathematical models in order to uncover their underlying mechanisms. The authors utilize an open-source repository of ODE and agent-based models from systems biology to build out the dataset of 350 models. An extensive data cleaning process is applied to the models to ensure that their settings are reasonably evaluated by LLMs, and the authors develop metrics that measure correspondence to ground truth structures of the underlying models. Using these metrics, the authors benchmark several LLMs in both a zero-shot prompting strategy and ReAct-style agentic workflows, utilizing tools that allow the models to perturb species within the systems. From this, several conclusions are drawn about the performance of LLMs in this setting, including systematic performance degradations in more complex systems and the superiority of agentic approaches as well as larger, more powerful models.

**Additional Feedback:**

Please see "Limitations Weaknesses" for suggestions on how to improve the work.

**Dataset Code Accessibility:**

Yes

**Dataset Code Comments:**

Data is available on Huggingface and appears to be widely accessible.

**Ethical Considerations:**

No, there are no or only very minor ethics concerns

**Final Justification:**

The paper tackles an important problem in AI scientist development: active experimentation evaluation. It uses a scalable, robust system to do so that 1) has backing in biomedical literature, given the history of systems biology approaches such as these and 2) is scalable for use in high-throughput experiments. Their paper identifies some important challenges in AI scientist development and pave the way for future research in this area. Given the creativity of this benchmark and what seems to be a rigorous setup and methodology, I recommend this paper for acceptance. However, it does not meet the bar for an "outstanding paper" (i.e., 6 rating) mainly because of the domain-specific focus of the benchmark.

**Limitations Weaknesses:**

While exhibiting many strengths, this paper also has a few weaknesses:
- While the paper is overall very detailed and precise for application to SBML predictions and systems biology models, the application is admittedly limited. Mathematical models such as those presented in these papers can be quite contrived, and it could be argued that this doesn’t represent open-ended scientific discovery. However, I believe that the authors tackle this narrow application very strongly (see Strengths).
- In a similar vein, I worry about some results in the paper being confounded by the complexity of the SBML language and formatting. Did the authors conduct any study of zero-shot prompting for these types of formats, i.e., understanding how the model natively models or understands these large XML files? This might confound results such as the one around larger more premier-tier models such as Gemini-2.5-Pro vs. Gemini-Mini as these larger models tend to be better at adhering to structured output.
- The authors present many results on their custom-defined metrics, but no insights are given into the failure modes or behaviors of the models in these settings. For instance, it would be interesting to understand how the models fail to design experiments or the number of steps used to perturb the system. More systematic study of these properties might improve interpretation of results.
- While the paper is very well written and communicated, it could serve from some more concrete discussion of the tasks on which the models are evaluated. I can see a reader with very little biology background having a hard time understanding some of the concrete details around SBML and why this approach is important for evaluating LLM capabilities.
- While not necessary, I would appreciate evaluation of reasoning models for these tasks, such as o3 or Claude thinking. Gemini-2.5-Pro (a reasoning model) is evaluated here, but this is then compared directly to GPT 4.1 and Claude 3.7-Sonnet, which are both non-reasoning models; thus, this comparison is not entirely fair in this case.

**Strengths Contributions:**

This paper has a number of strengths:
- The motivation for the work is compelling. The authors convincingly motivate the evaluation of LLMs and agents in experimental design protocols, which is an open problem in the field of AI for scientific discovery. To my knowledge, this is the first application of mathematical models as realistic systems on which LLMs can design perturbations to understand underlying mechanisms; this formulation is novel and seems to provide a useful arena for testing LLM approaches.
- The authors underwent a rigorous process for filtering out useful models in the SciGym dataset. The filtration parameters are sensible for this use case, and there is extensive discussion of different parameters and statistics of the models filtered out and included in the Appendix of the paper. This transparency and thoroughness is appreciated in a dataset-forward paper.
- The authors undergo extensive benchmarking and present clear, concise results to back up their claims. Their results have robust error bars, and their conclusions seem to be grounded in realistic results that are supported through experimentation.
- The transparency and clear limitations outline in the conclusion of the work is appreciated. It is important to acknowledge the limitation of this approach and why it does not represent a holistic approach to evaluating LLMs and agents in scientific discovery settings.

---

> ### Author Rebuttal · Authors · 2025-07-30
>
> Thank you for your encouraging comments about our work! Below we provide some clarifications to address the questions raised in the weakness sections.
>
> ---
>
> # Q1: SBML can be quite contrived and may not represent open-ended scientific discovery.
>
> Thanks for this question! First, we want to point out that the key goal of creating an AI scientist is to test if the agent has similar **skills** as real scientists. We think SciGym achieves this goal: it requires the agent to propose informative experiments, analyze data from experiments, and do it in an iterative way.
> We acknowledge that providing a realistic environment is also important to evaluate these skills, and ours still has gaps compared to real-world scientific discovery (explained in the limitation section). However, compared with previous benchmarks where agents cannot conduct perturbation experiments (Lab-bench) or the data is purely synthetic (DiscoveryWorld), we believe SciGym represents significant progress toward more realistic scientific discovery evaluation, even if we haven't fully closed the gap to real-world complexity.
>
> ---
>
> # Q2: Evaluation may be confounded by the complexity of SBML
>
> Thank you for raising this important question! As the reviewer suggested, we conducted new zero-shot experiments showing that frontier models possess similar levels of SBML knowledge required for SciGym. Therefore, performance bottlenecks between frontier models stem primarily from limitations in scientific reasoning rather than technical knowledge of SBML.
>
> ## Experiments: Frontier models possess similar levels of SBML knowledge for SciGym
>
>
> ### Experiment Motivation
>
> Knowing the syntax of SBML is a requirement to succeed in SciGym, but this challenge exists in virtually every benchmark. It's impossible to completely isolate abstract capabilities from concrete syntactic or domain-specific knowledge. **Moreover, we designed SciGym to minimize the SBML technical burden.** We filtered out advanced SBML features and retained only basic, standardized components (removing complex elements like `<events>` and `<rules>`). Our task primarily requires knowing how to add reactions, a simple operation that we also demonstrate with examples in the system prompt.
>
> ### Experiment Design
>
> We designed a simple zero-shot task to test whether the evaluated LLMs have the necessary SBML knowledge for SciGym (i.e., knowing how to add reactions using libsbml). In this experiment, the agent is given an incomplete SBML model and its corresponding complete SBML model. The agent is tasked to write a Python script using the libsbml package to add missing components to the incomplete model such that it matches the complete model. This set-up is similar to the one in scigym, except the agent is told what reactions are missing.
> Upon successful execution of the agent’s script, we extract the modified incomplete SBML model and systematically evaluate its equivalence to the complete version. We consider two models equivalent if they contain the exact same set of species, reactions, kinetic laws, parameters, values, and initial concentrations (this equivalence is sufficient to obtain perfect scores on the *NTS, RMS, STE* metrics). We randomly picked 50 models from the small SciGym split and evaluated the same six models in our paper.
>
> ### Experiment Results
>
> | Model                        | % Equiv | Avg. Iter. | Fail. Mode(s)                |
> |------------------------------|---------|------------|------------------------------|
> | claude-3-5-haiku-20241022    | 80      | 2.04       | wrong ID (2), equation (8)   |
> | claude-3-7-sonnet-20250219   | 100     | 1.16       | N/A                          |
> | gemini-2.5-flash-preview-05-20 | 98    | 1.16       | failed (1)                   |
> | gemini-2.5-pro-preview-03-25 | 100     | 1.71       | N/A                          |
> | gpt-4.1-2025-04-14           | 64      | 2.17       | wrong ID (9), failed (9)     |
> | gpt-4.1-mini-2025-04-14      | 66      | 2.44       | wrong eq. (1), failed (16)   |
> | o3-2025-04-16                | 96      | 1.47       | wrong ID (1), failed (1)     |
> | o4-mini-2025-04-16           | 44      | 2.73       | failed (28)                  |
>
>
> Notes on the failure modes:
>
> - Wrong ID → There was a mismatch in the IDs of at least one component
>
> - Wrong equation → There was a mismatch in the kinetic law equations
>
> - Failed → LLM submitted more than 4 responses that failed to parse or run
>
> ### Key Findings
>
> - From the results, **we first observe that most models possess sufficient SBML knowledge to succeed in our task**. If models lacked SBML knowledge, they should achieve near-zero success rates. Given that we used the same incomplete SBML models as those in our SciGym benchmark, these models demonstrate they can effectively manipulate SBML via libsbml for SciGym tasks.
>
> - **More importantly, all frontier models (Sonnet, Gemini, o3) achieve near-perfect performance on this task.** This implies that the performance bottleneck for frontier LLMs on SciGym stems primarily from scientific discovery aptitude rather than coding ability or SBML language mastery.
>
> ---
>
> # Q3: Systematic study on the failure modes
>
> This is a great point! We conducted some systematic experiments to investigate failure modes and model behaviors. Below are what we find:
>
> - Successful models spend significantly more time analyzing experimental results. We measured the number of analysis steps the agent makes for each experiment they proposed:
>
> | Model 				| # of analysis steps / experiment	|
> | ----------------------------------------------- | ----------------- |
> | claude-3-5-haiku-20241022 		| 2.4		|
> | claude-3-7-sonnet-20250219	| 8.6		|
> | gemini-2.5-flash-preview-05-20	| 7.2		|
> | gemini-2.5-pro-preview-03-25	| 8.9		|
> | gpt-4.1-2025-04-14			| 3.2		|
> | gpt-4.1-mini-2025-04-14		| 3.8		|
> | o3-2025-04-16			| 6.9		|
> | o4-mini			| 3.1		|
>
>
>
> The result shows that models that perform well on SciGym (sonnet, gemini, o3) dedicate more reasoning steps to interpreting experimental outcomes, suggesting that thorough analysis is crucial for scientific discovery tasks.
>
> - Most models struggle with modifier-related relationships. As shown in Table 2 in the paper, identifying catalytic relationships (modifiers) proves particularly challenging compared to direct reactant-product relationships.
>
> We acknowledge these studies are only initial explorations rather than comprehensive analysis. Systematically studying model behavior in open-ended reasoning tasks is not easy. Therefore, we hope that our open-sourced reasoning traces will enable the community to conduct deeper investigations into model failure modes and behavioral patterns.
>
> ---
>
> # Q4: More concrete discussions on the task
>
> Rebuttal: Thanks for your suggestions! We will improve the clarity of Section 2  in the final version of the paper to make it more friendly to readers with limited biology backgrounds. Specifically, we plan to add a concrete walkthrough example showing how a simple biological system translates into SBML format.
>
> We will also improve the Figure 3 and 4 to help the readers better understand the design of our tasks.
>
> - Fig 3: explicitly list what information the agent has access to (list of species in the system) and what parts of the reference system are hidden and must be discovered to complete the task (reactions, the relationships between species)
>
> - Fig 4: added legend which matches species shown in fig. 3 to illustrate how the agent may perturb concentrations of species, and not reactions directly.
>
>
> ---
>
> # Q5: Evaluate more reasoning models
>
>  This is a good point! We have added more thinking models (O3 and O4 mini). The results are shown below:
>
> | **Model** | **STE ↓** | **With Modifiers** |  |  | **Without Modifiers** |  |  |
> |-----------|-----------|-------------------|----|----|---------------------|----|----|
> |           |           | **Precision** | **Recall** | **F1** | **Precision** | **Recall** | **F1** |
> | O3        | 0.3821    | 0.1627        | 0.1412     | 0.1511 | 0.2863        | 0.2923     | 0.2893 |
> | O4 mini        | 0.6224    | 0.1241        | 0.0923 |  0.1059   | 0.1872  | 0.1439        |  0.1627   |
>
> We will add these results to the main paper. We find that O3 is on par with the best models like sonnet and Gemini-pro, while O4-mini performs on par with the less-capable models. However, we want to point out that fair comparison among models is inherently difficult since model size, training data, and other capabilities are black-box to us. **Our main goal is to establish SciGym as a new framework to test end-to-end scientific discovery rather than definitive model rankings.**

---

> > ### Comment · Reviewer_vZ9f · 2025-08-01
> >
> > Thank you to the authors for addressing these points! While I cannot increase my score because I do not believe the paper quite warrants a 6, given mainly the limited application space, this response makes me more confident in my score of 5. I recommend this work for acceptance as it would be a valuable contribution to benchmarking AI scientist models for future development. Thank you again!

---

> > > ### Author Response · Authors · 2025-08-01
> > >
> > > Thank you very much for the encouraging comments!

---

### Author Response · Authors · 2025-08-01
**Summary of the response**

We sincerely thank all four reviewers for their thoughtful feedback, which has greatly helped us improve the quality of this work. We are pleased that reviewers found our work novel and well-motivated, noting that we "convincingly motivate the evaluation of LLMs and agents in experimental design protocols" (reviewer vZ9f) and that "using validated ODE models of biological systems to simulate scientific discovery processes is highly interesting and novel" (reviewer pjT8). We are also glad that reviewers appreciated our methods in designing the benchmark, with reviewer 6BtY observing that "the dataset was thoroughly curated" with "carefully designed" methodology, and reviewer PzAn noting that "SciGym is highly extensible" for future research.

We have conducted additional experiments and provided detailed responses to each reviewer's concerns. Below we provide a brief summary of each response.


> Is SciGym evaluating scientific discovery or simply ODE discovery? (Reviewer vZ9f and pjT8)

SciGym uses models studied by actual biologists to represent real biological processes, so reasoning like a biologist would be essential to recover these models in SciGym. We designed a controlled experiment to test whether SciGym requires biological knowledge by removing domain context while keeping identical mathematical structure. Results show significant performance drops when biological reasoning is removed, confirming our hypothesis. Furthermore, SciGym explicitly tests the core reasoning skills required for scientists: hypothesis formation, experiment design and data analysis. These skills are essential for scientific discovery and go beyond what's required for standard ODE discovery tasks.

> Do SBML knowledge and coding skills confound our scientific reasoning evaluation? (Reviewer vZ9f and pjT8)

End-to-end scientific discovery inherently requires reasoning, engineering, and domain-specific skills to work in concert. We agree that coding is essential for success in our benchmark, but this reflects the reality of modern scientific research. Regarding SBML knowledge, we designed new experiments demonstrating that frontier models possess similar levels of SBML knowledge related to SciGym. Therefore, performance bottlenecks stem primarily from limitations in scientific reasoning rather than technical knowledge of SBML.

> Why weren't the large SBML models (>10 reactions) evaluated? (Reviewer PzAn)

Our results show that frontier models already struggle with the current complexity in SciGym. The experiments on 10-reaction models show uniformly poor performance across all models, and even the best model achieves close to zero reaction matching accuracy on 10-reaction systems, Therefore, we decided to focus our computational resources on smaller systems where we can meaningfully differentiate between the current frontier models.


> What insights can be provided about model failure modes and behaviors in scientific reasoning tasks? (Reviewer vZ9f)

We conducted some systematic experiments to investigate failure modes in SciGym. We find that successful models spend significantly more time analyzing experimental results and that most models struggle specifically with modifier-related relationships compared to direct reactant-product relationships. However, we acknowledge these are initial explorations and have open-sourced our reasoning traces to enable deeper community investigation into model behavioral patterns.

> Does restricting agents to initial concentration changes exclude important experimental manipulations and encourage shortcuts? (Reviewer PzAn)

Changing initial concentrations is a fundamental experimental approach across biology. It also provides a rich exploration landscape that requires sophisticated reasoning (e.g., identifying modifiers vs. substrates, determining reaction kinetics forms). While we could incorporate specialized manipulations like time-varying stimuli via SBML events, SciGym is designed as a general biology benchmark that works across diverse domains rather than any single field.


> Can this work be adapted to fields other than biology? (Reviewer 6BtY)

Yes, SBML can describe any ODE-format dynamic system in theory. As long as domain experts curate such models to ensure they represent genuine mechanistic relationships, we can incorporate them in SciGym.


> Should we evaluate more reasoning models for fair comparison? (Reviewer vZ9f)

We expanded our evaluation to include additional reasoning models (O3 and O4 mini) alongside existing models. However, fair comparison among models is inherently difficult since model size, training data, and other capabilities are black-box to us. Our main goal is to establish SciGym as a new framework to test end-to-end scientific discovery rather than definitive model rankings.



We believe we have addressed all concerns raised by the reviewers. Please see our individual responses above for detailed explanations.

---

### Decision · Program_Chairs · 2025-09-18

**Decision:**

Accept (poster)

**Comment:**

(a) This paper introduces SciGym, a benchmark for the scientific reasoning capabilities of LLMs for Systems Biology. The authors claim that no existing benchmarks exist for testing experimental design and result interpretation capabilities, which is mainly down to the fact that wet-lab experiments are prohibitively expensive and/or require too much dedicated knowhow for most. SciGym uses a "dry lab" approach (aka in-silico) to address this, using a set of 350 systems biology models encoded in Systems Biology Markup Language (SBML). These models enable efficient (simulated) experiments (typically ODEs or stochastic systems) where an LLM can design experimental perturbations, analyse the results, and attempt to uncover the underlying biological mechanisms of the system being tested. 6 LLMs were tested: in general more powerful models performed better, but all models showed significant declines in performance as the complexity of the biological systems increased. This suggests that current LLMs still fall short as scientific reasoning agents.

(b) The main strengths are the novelty and design of the SciGym benchmark. The reviewers agreed that the approach of using validated ODE models of biological systems to simulate scientific discovery is both "highly interesting and novel" (reviewer pjT8) and "convincing" (reviewer vZ9f). The paper's contributions are particularly relevant to the DB Track CFP, as it introduces a new, thoroughly curated dataset and a well-designed framework for benchmarking AI systems. Specific strengths highlighted by the reviewers include novelty (first to use mathematical models as realistic systems for LLMs, reviewer vZ9f), rigorous curation (reviewer 6BtY), extensibility (reviewer PzAn) and clear and concise results (reviewer vZ9f).

(c) Some minor weaknesses were identified, but were addressed during the discussion period as follows. Reviewers vZ9f and pjT8 initially expressed concern that the evaluation might be confounded by the models' coding skills and knowledge of the SBML format, rather than their scientific reasoning abilities. Reviewer PzAn noted that restricting the agents to only changing initial concentrations excludes other important experimental manipulations and could encourage "shortcut" strategies. There was a question from reviewer 6BtY as to whether the benchmark could be applied to other scientific disciplines outside of biology. Reviewer PzAn pointed out that a large portion of the included SBML models (the larger, more complex ones) were not used in the evaluation.

(d) The recommendation to accept this paper is based on its novel and well-executed contribution to the field of benchmarking LLMs for scientific discovery. This represents a valuable resource for the community, and the paper is technically sound and well-written. The key reasons for acceptance are the high impact on the field of AI for science, the rigorous and well-documented curation of datasets, and the comprehensive evaluation of several state-of-the-art LLMs, which provide valuable insights into their current capabilities and limitations.

(e) In response to reviewers vZ9f and pjT8, the authors conducted new experiments to test whether frontier models possess sufficient SBML knowledge. The results showed that top models have near-perfect performance on tasks requiring SBML manipulation, suggesting that the performance bottleneck in SciGym is indeed scientific reasoning, not technical skill. To address reviewer pjT8's concern about whether the benchmark truly tests biological reasoning, the authors ran a controlled experiment where the biological context was removed from the task description. When doing this, a significant drop in performance was seen, that confirmed that biological reasoning is essential for success in SciGym. Reviewer 6BtY's concern about the benchmark's applicability to other domains was addressed by the authors pointing out that SBML is a flexible language capable of describing a wide range of dynamic systems, including those in chemistry and epidemiology, some of which are already present in the dataset. In response to reviewer PzAn, the authors argued that varying initial concentrations is a fundamental and rich experimental paradigm in biology. They also clarified that the current design is intended to be a general benchmark, with future work potentially incorporating more specialised perturbations. Regarding the larger models that weren’t evaluated, the authors explained that current models already struggle with the complexity of the smaller systems, and provided additional data showing near-zero performance on moderately complex system.

===== FINAL UPDATE FROM DB Track PCs ====

The final decision for this paper has been taken by the program chairs after consultation with the SACs. All Senior Area Chairs have ranked papers according to the feedback from the AC during the review process. We decided to leave the original meta-review to reflect the opinion of the AC in light of the initial discussions with reviewers and SAC.